# GENERALIZATION BELOW THE EDGE OF STABILITY: THE ROLE OF DATA GEOMETRY

**Tongtong Liang**[*]
UC San Diego

**Alexander Cloninger**
UC San Diego

**Rahul Parhi**
UC San Diego

**Yu-Xiang Wang**
UC San Diego

## ABSTRACT

Understanding generalization in overparameterized neural networks hinges on the interplay between the data geometry, neural architecture, and training dynamics. In this paper, we theoretically explore how data geometry controls this implicit bias. This paper presents theoretical results for overparametrized two-layer ReLU networks trained *below the edge of stability*. First, for data distributions supported on a mixture of low-dimensional balls, we derive generalization bounds that provably adapt to the intrinsic dimension. Second, for a family of isotropic distributions that vary in how strongly probability mass concentrates toward the unit sphere, we derive a spectrum of bounds showing that rates deteriorate as the mass concentrates toward the sphere. These results instantiate a unifying principle: When the data is harder to "shatter" with respect to the activation thresholds of the ReLU neurons, gradient descent tends to learn representations that capture shared patterns and thus finds solutions that generalize well. On the other hand, for data that is easily shattered (e.g., data supported on the sphere) gradient descent favors memorization. Our theoretical results consolidate disparate empirical findings that have appeared in the literature.

## 1 INTRODUCTION

How does gradient descent (GD) discover well-generalizing representations in overparameterized neural networks, when these models possess more than enough capacity to simply memorize the training data? Conventional wisdom in statistical learning attributes this to explicit capacity control via regularization such as weight decay. However, this view has been profoundly challenged by empirical findings that neural networks generalize remarkably even without explicit regularizers, yet can also fit randomly labeled data with ease, even with strong regularization (Zhang et al., 2017).

This paradox forces a critical re-evaluation of how we should characterize the effective capacity of neural networks, which appears to be implicitly constrained by the optimizer's preferences (Zhang et al., 2017; Arpit et al., 2017). A powerful lens for examining this *implicit regularization* is to inspect the properties of solutions to which GD can stably converge, since these stable points are the only solutions that the training dynamics can practically reach and maintain. This direction is strongly motivated by the empirical discovery of the "*Edge of Stability*" (EoS) regime, where GD with large learning rates operates in a critical regime where the step size is balanced by the local loss curvature (Cohen et al., 2020). This observation is further supported by theoretical analyses of GD's dynamical stability (Wu et al., 2018; Nar and Sastry, 2018; Mulayoff et al., 2021; Nacson et al., 2023; Damian et al., 2024), confirming that the curvature constraint imposed by stability provides a tractable proxy for this implicit regularization.

While the EoS regime offers a valuable proxy, a fundamental question remains: how precisely does this stability-induced regularization lead to generalization? Recent breakthroughs have established that for two-layer ReLU networks, this implicit regularization acts like a data-dependent penalty on the network's complexity. Technically, this is captured by a weighted path norm, where the weight function is determined by the training dataset itself (Liang et al., 2025; Qiao et al., 2024; Nacson et al., 2023; Mulayoff et al., 2021). This resulting *data-dependent regularity* provides an ideal theoretical microcosm to probe how data geometry governs effective capacity (Arpit et al., 2017). For example, for the uniform distribution on a ball, it implies generalization but also a curse of

---

[*]Correspondence: `ttliang@ucsd.edu`

dimensionality (Liang et al., 2025). However, this prediction of a curse is at odds with the empirical success of deep learning. This contradiction forces the question: how can we predict which data geometries will generalize well under implicit regularization, and which will not?

**Contributions.** We argue that the effectiveness of this data-dependent regularity is governed by a single geometric quantity, which we call *data shatterability*: qualitatively, how easily the data distribution can be partitioned into many disjoint small regions by ReLU half-spaces. Informally, *the less shatterable the data geometry, the stronger the implicit regularization at the EoS.* We make this principle precise and obtain the following results.

- **A Spectrum of Generalization on Isotropic Data.** For a one-parameter family of isotropic $\text{Beta}(\alpha)$-radial distributions, we derive generalization upper bounds and matching lower bounds that depend smoothly on the radial concentration parameter $\alpha$ (Theorems 3.4 and 3.5). As $\alpha$ decreases, the probability mass moves towards the boundary and the generalization guarantee degrades. In the limiting case where the support collapses to the unit sphere, we construct perfectly interpolating networks that still satisfy the BEoS stability condition (Theorem 3.6). In particular, the "neural shattering" phenomenon, identified by Liang et al. (2025) for the uniform ball distribution, represents one special point of the broader generalization spectrum we uncover.

- **Provable Adaptation to Low-dimensionality.** Under a mixture-of-subspaces assumption where inputs are supported on a union of $m$-dimensional balls in $\mathbb{R}^d$ with $m < d$, we prove that all BEoS-stable solutions enjoy a generalization rate $\tilde{O}\big(n^{-1/(2m+4)}\big)$ that depends on the *intrinsic* dimension $m$ rather than the ambient dimension $d$ (Theorem 3.10). The dependence on the number of mixture components is at most polynomial. Synthetic experiments confirm that gradient descent indeed behaves according to this intrinsic-dimension law.

Taken together, these results identify data shatterability as the key geometric quantity that controls the strength of implicit regularization for gradient descent below the edge of stability.

**Technical novelty.** Many classical uniform-convergence-based generalization bounds for overparameterized neural networks are distribution-agnostic. This approach is not available in our setting. The data-dependent regularity induced by the EoS condition defines a function class whose $L^\infty$ metric entropy is infinite, and our spherical interpolation result shows that nontrivial distribution-agnostic bounds cannot hold. The key observation is that stability-induced regularity is highly inhomogeneous over the input domain: there are regions where the effective regularization is strong ("good" regions) and regions where it is extremely weak ("bad" regions) such that metric entropy explodes.

Our main technical innovation is to bypass global metric entropy control via a *half-space-depth quantile partition* of the input space. This technique allows us to decouple the analysis based on the strength of regularization: In the "good region", the implicit regularization is effective, allowing us to enforce strict complexity control, while in the "bad region", where complexity explodes, we abandon function-space covering arguments. Instead, we control the generalization error by bounding the *probability mass* of these regions, tying the error directly to the data geometry.

Conceptually, this framework inverts the classical VC dimension perspective. While VC dimension characterizes a model's active capacity to shatter *arbitrary* data, our "data shatterability" principle characterizes the *feasibility* of a specific dataset being shattered by the GD-trained network.

**Related work.** We build upon a recent line of work (Wu and Su, 2023; Qiao et al., 2024; Liang et al., 2025) that theoretically study the generalization of neural networks in Edge-of-Stability regime (Cohen et al., 2020) from a function space perspective (Mulayoff et al., 2021; Nacson et al., 2023). We add to this literature by abstracting neural-shattering analysis on uniform-ball of Liang et al. (2025) into a depth-based data-shatterability framework and extending it to a broader class of distributions that capture both radial concentration and low-dimensional structure, yielding distribution-dependent upper and lower bounds that make the role of data geometry explicit.

More broadly, our work is inspired by the seminal work of Zhang et al. (2017) on "rethinking generalization". Our results provide new theoretical justification that rigorously explains several curious phenomena (such as why real data are harder to overfit than random Gaussian data) reported therein. Compared to other existing work inspired by Zhang et al. (2017), e.g., those that study the implicit bias of gradient descent from various alternative angles (dynamics (Arora et al., 2019; Mei

et al., 2019; Jin and Montúfar, 2023), algorithmic stability (Hardt et al., 2016), large-margin (Soudry et al., 2018), benign overfitting (Joshi et al., 2024; Kornowski et al., 2024)), our work has more end-to-end generalization bounds and requires (morally, since the settings are not all compatible) weaker assumptions. On the practical front, we provide new theoretical insight into how "mix-up" data augmentation (Zhang et al., 2018; 2021) and "activation-based pruning" (Hu et al., 2016; Ganguli and Chong, 2024) work. A more detailed discussion of the related work and the implications of our results can be found in Appendix B.1.

**Scope of analysis.** Our theoretical framework is situated in the feature-learning (or "rich") regime of overparameterized networks. Rather than tracking the full gradient dynamics[1], we focus on the regime in which gradient descent with a large step size operates for long stretches below around the edge-of-stability boundary. We therefore analyze the generalization behavior of all parameter vectors satisfying a Below-Edge-of-Stability (BEoS, see Definition 2.1), without assuming optimality or stationarity. Our bounds hold uniformly over this BEoS region and characterize a baseline form of implicit regularization that is enforced whenever training remains near the edge of stability.

## 2 PRELIMINARIES AND NOTATIONS

**Neural network, data, and loss.** We consider two-layer ReLU networks

$$f_{\boldsymbol{\theta}}(\boldsymbol{x}) = \sum_{k=1}^{K} v_k \, \phi(\boldsymbol{w}_k^{\mathsf{T}} \boldsymbol{x} - b_k) + \beta, \quad \phi(z) = \max\{z, 0\}, \tag{1}$$

with parameters $\boldsymbol{\theta} = \{(v_k, \boldsymbol{w}_k, b_k)\}_{k=1}^{K} \cup \{\beta\} \in \mathbb{R}^{(d+2)K+1}$. Let $\boldsymbol{\Theta}$ be the parameter set of such $\boldsymbol{\theta}$ for arbitrary $K \in \mathbb{N}$. We also assume $\boldsymbol{w}_k \neq \boldsymbol{0}$ for all $k$ in this form, otherwise we may absorb it into the output bias $\beta$. Given data $\mathcal{D} = \{(\boldsymbol{x}_i, y_i)\}_{i=1}^{n}$ with $\boldsymbol{x}_i$ in a bounded domain $\Omega \subset \mathbb{R}^d$ with $d > 1$, the training loss is $\mathcal{L}(\boldsymbol{\theta}) = \frac{1}{2n} \sum_{i=1}^{n} \big(f_{\boldsymbol{\theta}}(\boldsymbol{x}_i) - y_i\big)^2$. We assume $\|\boldsymbol{x}_i\| \leq R$ and $|y_i| \leq D$ for all $i$.

**"Edge of Stability" regime.** Empirical and theoretical research (Cohen et al., 2020; Damian et al., 2024) has established the critical role of the linear stability threshold in the dynamics of gradient descent. In GD's trajectory, there is an initial phase of "progressive sharpening" where $\lambda_{\max}(\nabla^2 \mathcal{L}(\boldsymbol{\theta}_t))$ increases. This continues until the GD process approaches the "Edge of Stability", a state where $\lambda_{\max}(\nabla^2 \mathcal{L}(\boldsymbol{\theta}_t)) \approx 2/\eta$, where $\eta$ is the learning rate. In this paper, all the GD refers to vanilla GD with learning rate $\eta$.

**Definition 2.1** (Below Edge of Stability (Qiao et al., 2024, Definition 2.3))**.** *We define the trajectory of parameters $\{\boldsymbol{\theta}_t\}_{t=1,2,\cdots}$ generated by gradient descent with a learning rate $\eta$ as* Below-Edge-of-Stability (BEoS) *if there exists a time $t^* > 0$ such that for all $t \geq t^*$, $\lambda_{\max}(\nabla^2 \mathcal{L}(\boldsymbol{\theta}_t)) \leq \frac{2}{\eta}$. Any parameter state $\boldsymbol{\theta}_t$ with $t \geq t^*$ is thereby referred to as a BEoS solution.*

This condition applies to any twice-differentiable solution found by GD, even when the optimization process does not converge to a local or global minimum. Moreover, BEoS is empirically verified to hold during both the "progressive sharpening" phase and the subsequent oscillatory phase.

Our work aims to analyze the generalization properties of any solutions that satisfy the BEoS condition (Definition 2.1). The set of solutions defined as:

$$\boldsymbol{\Theta}_{\mathrm{BEoS}}(\eta, \mathcal{D}) := \left\{ \boldsymbol{\theta} \,\middle|\, \lambda_{\max}(\nabla^2 \mathcal{L}(\boldsymbol{\theta})) \leq \frac{2}{\eta} \right\}. \tag{2}$$

**Data-dependent weighted path norm.** Given a weight function $g : \mathbb{S}^{d-1} \times \mathbb{R} \to \mathbb{R}$, where $\mathbb{S}^{d-1} := \{\boldsymbol{u} \in \mathbb{R}^d : \|\boldsymbol{u}\| = 1\}$, the $g$-*weighted path norm* of a neural network $f_{\boldsymbol{\theta}}(\boldsymbol{x}) = \sum_{k=1}^{K} v_k \phi(\boldsymbol{w}_k^{\mathsf{T}} \boldsymbol{x} - b_k) + \beta$ is defined to be

$$\|f_{\boldsymbol{\theta}}\|_{\mathrm{path},g} = \sum_{k=1}^{K} |v_k| \|\boldsymbol{w}_k\|_2 \cdot g\left(\frac{\boldsymbol{w}_k}{\|\boldsymbol{w}_k\|_2}, \frac{b_k}{\|\boldsymbol{w}_k\|_2}\right). \tag{3}$$

The link between the EoS regime and weighted path norm constraint is presented in the following data-dependent weight function (Liang et al., 2025; Nacson et al., 2023; Mulayoff et al., 2021). Fix

---

[1]Nevertheless, there is informal and heuristic discussion from the perspective of gradient dynamics in Appendix B.2.

a dataset $\mathcal{D} = \{(\boldsymbol{x}_i, y_i)\}_{i=1}^n \subset \mathbb{R}^d \times \mathbb{R}$, we consider a weight function $g_{\mathcal{D}} : \mathbb{S}^{d-1} \times \mathbb{R} \to \mathbb{R}$ defined by $g_{\mathcal{D}}(\boldsymbol{u}, t) := \min\{\tilde{g}_{\mathcal{D}}(\boldsymbol{u}, t), \tilde{g}_{\mathcal{D}}(-\boldsymbol{u}, -t)\}$, where

$$\tilde{g}_{\mathcal{D}}(\boldsymbol{u}, t) := \mathbb{P}(\boldsymbol{X}^\mathsf{T}\boldsymbol{u} > t)^2 \cdot \mathbb{E}[\boldsymbol{X}^\mathsf{T}\boldsymbol{u} - t \mid \boldsymbol{X}^\mathsf{T}\boldsymbol{u} > t] \cdot \sqrt{1 + \|\mathbb{E}[\boldsymbol{X} \mid \boldsymbol{X}^\mathsf{T}\boldsymbol{u} > t]\|^2}. \quad (4)$$

Here, $\boldsymbol{X}$ is a random vector drawn uniformly at random from the training examples $\{\boldsymbol{x}_i\}_{i=1}^n$. Specifically, we may also consider its population level $g_{\mathcal{P}}$ by viewing $\boldsymbol{X}$ as a random variable.

Informally, $g_{\mathcal{D}}(\boldsymbol{u}, t)$ measures how hard for GD to place a ReLU ridge with normalized $\boldsymbol{u}$ and threshold $t$ under the BEoS constraint. Large $g_{\mathcal{D}}$ corresponds to directions that sense that sense significant data activation, thereby yielding strong gradients and hence strong implicit regularization. A more detailed interpretation from the perspective of gradient dynamics is given in Appendix B.2.

**Proposition 2.2.** *For any* $\boldsymbol{\theta} \in \boldsymbol{\Theta}_{\mathrm{BEoS}}(\eta, \mathcal{D})$, $\|f_{\boldsymbol{\theta}}\|_{\mathrm{path}, g_{\mathcal{D}}} \leq \frac{1}{\eta} - \frac{1}{2} + (R+1)\sqrt{2\mathcal{L}(\boldsymbol{\theta})}$.

The proof of this proposition refers to (Liang et al., 2025, Corollary 3.3). The non-parametric version of the weighted path norm constrain can be found in (Liang et al., 2025; Nacson et al., 2023).

**Supervised statistical learning and generalization gap.** We consider a supervised learning problem where i.i.d. samples $\mathcal{D} = \{(\boldsymbol{x}_i, y_i)\}_{i=1}^n$ are drawn from an unknown distribution $\mathcal{P}$. In this paper, we assume the feature space is a compact subset of Euclidean space, $\Omega \subset \mathbb{R}^d$, the label space is $\mathbb{R}$, and the data is supported on $\Omega \times [-D, D]$. We use the squared loss, defined as $\ell(f, \boldsymbol{x}, y) = \frac{1}{2}(f(\boldsymbol{x}) - y)^2$. The performance of a predictor $f$ is measured by its population risk $R_{\mathcal{P}}(f) = \mathbb{E}_{(\boldsymbol{X}, Y) \sim \mathcal{P}} \ell(f, \boldsymbol{X}, Y)$, while we optimize the empirical risk $\widehat{R}_{\mathcal{D}}(f) = \frac{1}{|\mathcal{D}|} \sum_{(\boldsymbol{x}_i, y_i) \in \mathcal{D}} \ell(f, \boldsymbol{x}_i, y_i)$. The difference between these two quantities is the generalization gap $\mathrm{Gap}_{\mathcal{P}}(f; \mathcal{D}) = |R_{\mathcal{P}}(f) - \widehat{R}_{\mathcal{D}}(f)|$. Our work focuses on the hypothesis classes the BEoS class $\boldsymbol{\Theta}_{\mathrm{BEoS}}(\eta, \mathcal{D})$ and the bounded weighted-path norm class $\boldsymbol{\Theta}_g(\Omega; M, C)$,

$$\boldsymbol{\Theta}_g(\Omega; M, C) = \left\{ \boldsymbol{\theta} \in \boldsymbol{\Theta} \,\middle|\, \|f_{\boldsymbol{\theta}}|_\Omega\|_{L^\infty} \leq M, \, \|f_{\boldsymbol{\theta}}\|_{\mathrm{path}, g} \leq C \right\}. \quad (5)$$

where $g$ can be the weight function $g_{\mathcal{D}}$ associated to the empirical distribution $\mathcal{D}$ or the weight function $g_{\mathcal{P}}$ associated to the population distribution $\mathcal{P}$, see Section E for more details.

## 3  DATA SHATTERABILITY PRINCIPLE AND GENERALIZATION BOUNDS

This section introduces our notion of *data shatterability* and explains how it leads to a unified proof strategy for both the generalization upper bounds (Theorem 3.10, Theorem 3.4) and the lower bounds (Theorem 3.5, Theorem 3.6).

We start with the classical notion of **half-space depth** (also known as **Tukey depth**).

**Definition 3.1.** *Given a distribution* $\mathcal{P}_X$ *on* $\mathbb{R}^d$. *For any* $\boldsymbol{x} \in \mathbb{R}^d$, *its* **population half-space depth** $\mathrm{depth}(\boldsymbol{x}, \mathcal{P}_X) = \inf_{\boldsymbol{u} \in \mathbb{S}^{d-1}} \mathbb{P}_{\boldsymbol{X} \sim \mathcal{P}_X} (\boldsymbol{u}^\mathsf{T}(\boldsymbol{X} - \boldsymbol{x}) \geq 0)$. *On the other hand, given a point cloud* $\mathcal{X} = \{\boldsymbol{x}_1, \cdots, \boldsymbol{x}_n\} \subset \mathbb{R}^d$ *that is also represented by the empirical distribution* $\mathcal{P}(\mathcal{X})$ *that assigns mass* $1/n$ *to each point, the* **empirical half-space depth** $\mathrm{depth}(\boldsymbol{x}, \mathcal{X}) = \inf_{\boldsymbol{u} \in \mathbb{S}^{d-1}} \frac{1}{n} \sum_{i=1}^n \mathbb{1}\{\boldsymbol{u}^\mathsf{T}(\boldsymbol{x}_i - \boldsymbol{x}) \geq 0\}$.

Half-space depth measures the centrality of a point $\boldsymbol{x}$ by finding the minimum data mass (either population or empirical) on one side of any hyperplane passing through it (Tukey, 1975). For any $T \in [0, \frac{1}{2}]$, we define $T$-**deep region** $\Omega_T(\mathcal{P}_X) = \{\boldsymbol{x} \in \mathbb{R}^d \mid \mathrm{depth}(\boldsymbol{x}, \mathcal{P}_X) \geq T\}$ as an upper level set of the half-space depth function (we define $\Omega_T(\mathcal{X})$ similarly).

Now let $\mathcal{X} = \{\boldsymbol{x}_1, \ldots, \boldsymbol{x}_n\}$ be the point cloud of inputs in the dataset $\mathcal{D}$. The key geometric observation is that, for a ReLU neuron to create nonlinearity inside a $T$-deep region $\Omega_T(\mathcal{X})$, its activation boundary must intersect that region. Formally, let $f_{\boldsymbol{\theta}}$ be a network of the form (1) and define $N_T := \left\{ k \,\middle|\, \{\boldsymbol{w}_k^\mathsf{T}\boldsymbol{x} - b_k = 0\} \cap \Omega_T(\mathcal{X}) \neq \emptyset \right\}$. The neurons with $k \notin N_T$ are either always active or always inactive on $\Omega_T(\mathcal{X})$, hence contribute only an affine function on that region. Therefore there exists an affine function $\boldsymbol{x} \mapsto \boldsymbol{c}^\mathsf{T}\boldsymbol{x} + b$ (which absorbs all neurons with $k \notin N_T$) such that $f_{\boldsymbol{\theta}}(\boldsymbol{x}) = \sum_{k \in N_T} v_k \phi(\boldsymbol{w}_k^\mathsf{T}\boldsymbol{x} - b_k) + \boldsymbol{c}^\mathsf{T}\boldsymbol{x} + b, \forall \boldsymbol{x} \in \Omega_T(\mathcal{X})$.

By the definition of half-space depth, every hyperplane passing through a point in $\Omega_T(\mathcal{X})$ leaves at least a $T$-fraction of the data on each side. In particular, for any neuron whose boundary intersects

$\Omega_T(\mathcal{X})$, its activation event has probability at least $T$ (population or empirical, depending on the context). This gives a positive lower bound on the data-dependent weight function $g$. We denote

$$g_{\mathcal{D}}^{\min}(T) := \inf_{\{\boldsymbol{u}^\mathsf{T}\boldsymbol{x}-t=0\}\cap\Omega_T(\mathcal{X})\neq\emptyset} g_{\mathcal{D}}(\boldsymbol{u},t) > 0, \qquad g_{\mathcal{P}}^{\min}(T) := \inf_{\{\boldsymbol{u}^\mathsf{T}\boldsymbol{x}-t=0\}\cap\Omega_T(\mathcal{P}_X)\neq\emptyset} g_{\mathcal{P}}(\boldsymbol{u},t).$$

Thus, the neurons that generate nonlinearity on $\Omega_T(\mathcal{X})$ are effectively regularized by the BEoS condition, with strength controlled from below by $g_{\mathcal{D}}^{\min}(T)$ (and $g_{\mathcal{P}}^{\min}(T)$ at the population level).

On the $T$-deep region, only neurons in $N_T$ contribute nonlinearity, and for these we have $g_{\mathcal{D}}(\boldsymbol{u}_k,t_k) \geq g_{\mathcal{D}}^{\min}(T)$. Hence their unweighted path norm satisfies $\sum_{k\in N_T} |v_k|\,\|\boldsymbol{w}_k\|_2 \leq (g_{\mathcal{D}}^{\min}(T))^{-1}\|\|f_{\boldsymbol{\theta}}\|\|_{\mathrm{path},g_{\mathcal{D}}}$, so the restriction of $f_{\boldsymbol{\theta}}$ to $\Omega_T(\mathcal{X})$ lies in a standard (unweighted) path-norm ball with radius proportional to $(g_{\mathcal{D}}^{\min}(T))^{-1}$. Generalization bounds for such unweighted path-norm classes on bounded inputs are known (Parhi and Nowak, 2023; Neyshabur et al., 2015) and yield the $T$-deep term in (6). Outside the $T$-deep region, BEoS provides only weak control, so we bound the contribution by the worst-case $L^\infty$ amplitude times the probability mass of the shallow region. Finally, since $g_{\mathcal{D}}^{\min}(T)$ and $\Omega_T(\mathcal{X})$ are random, we replace them by their population counterparts $g_{\mathcal{P}}^{\min}(T)$ and $\Omega_T(\mathcal{P}_X)$ and control the discrepancy using empirical process tools (Appendix E.2). This yields the following generic decomposition, holding with high probability

$$\sup_{\boldsymbol{\theta}\in\boldsymbol{\Theta}_{\mathrm{BEoS}}(\eta,\mathcal{D})} \mathrm{Gap}(f_{\boldsymbol{\theta}},\mathcal{D}) \leq \underbrace{\tilde{O}\Big(\mathbb{P}_{\boldsymbol{X}}(\boldsymbol{X}\notin\Omega_T(\mathcal{P}_X))\Big)}_{\text{shallow region}} + \underbrace{\tilde{O}\Big(g_{\mathcal{P}}^{\min}(T)^{-\frac{d}{2d+3}}\,n^{-\frac{d+3}{4d+6}}\Big)}_{T\text{-deep region}}, \quad (6)$$

where $\tilde{O}$ hides logarithmic factors and universal constants. This inequality is the main technical bridge between data geometry and the behavior of BEoS solutions. Now we instantiate it for isotropic and anisotropic distributions to establish the corresponding generalization bounds.

### 3.1 A SPECTRUM OF GENERALIZATION ON ISOTROPIC DISTRIBUTIONS

We now specialize to isotropic distributions with compact support and assume that the maximal support radius is 1. In this case, the depth of a point depends only on its radius. More precisely, we consider distributions of the form $\boldsymbol{X} = h(R)\boldsymbol{U}$, $\boldsymbol{U} \sim \mathrm{Uniform}(\mathbb{S}^{d-1})$, where $h$ is a radial profile and $R$ is a scalar random variable. After rescaling, we may assume the support of $\mathcal{P}_X$ is contained in $\mathbb{B}_1^d$. Writing $r = \|\boldsymbol{x}\|_2$ and setting $\varepsilon := 1 - r$, we decompose $\mathbb{B}_1^d$ into **(1) $\varepsilon$-annulus** $\mathbb{A}_\varepsilon^d := \{\boldsymbol{x}\in\mathbb{B}_1^d \mid \|\boldsymbol{x}\|_2 \geq 1-\varepsilon\}$, **(2) $\varepsilon$-strict interior** $\mathbb{I}_\varepsilon^d := \mathbb{B}_{1-\varepsilon}^d = \overline{\mathbb{B}_1^d \setminus \mathbb{A}_\varepsilon^d}$.

For isotropic $\mathcal{P}_X$, the weight function $g_{\mathcal{P}}(\boldsymbol{u},t)$ depends only on $t$, so we write $g_{\mathcal{P}}(t)$. In this setting, $g_{\mathcal{P}}(1-\varepsilon)$ is a lower bound on the regularization strength for ReLUs whose activation boundaries lie at offsets $t \leq 1-\varepsilon$. For any fixed $\varepsilon\in(0,1)$, plugging this into (6) and rewriting in radial form, we obtain

$$\sup_{\boldsymbol{\theta}\in\boldsymbol{\Theta}_{\mathrm{BEoS}}(\eta,\mathcal{D})} \mathrm{Gap}(f_{\boldsymbol{\theta}},\mathcal{D}) \leq \tilde{O}\Big(\mathcal{P}_X(\mathbb{A}_\varepsilon^d)\Big) + \tilde{O}\Big(g_{\mathcal{P}}(1-\varepsilon)^{-\frac{d}{2d+3}}\,n^{-\frac{d+3}{4d+6}}\Big), \quad \text{w.h.p.} \quad (7)$$

We instantiate this decomposition on a flexible family of radial profiles that interpolate between a "thick" center-concentration ball and a thin spherical shell.

**Definition 3.2** (Isotropic Beta-radial distributions). *Let $\boldsymbol{X}$ be a $d$-dimensional random vector in $\mathbb{R}^d$. For any $\alpha\in(0,\infty)$, the isotropic $\mathrm{Beta}(\alpha)$-radial distribution is defined by*

$$\boldsymbol{X} = h(R)\boldsymbol{U} \sim \mathcal{P}_X(\alpha), \quad (8)$$

*where $R \sim \mathrm{Uniform}[0,1]$, $\boldsymbol{U} \sim \mathrm{Uniform}(\mathbb{S}^{d-1})$, and $h(r) = 1 - (1-r)^{1/\alpha}$ is a radial profile.*

**Assumption 3.3.** *Fix $\alpha\in(0,\infty)$. Let $\mathcal{P}(\alpha)$ be a joint distribution over $\mathbb{R}^d\times\mathbb{R}$ whose marginal over $\boldsymbol{x}$ is $\mathcal{P}_{\boldsymbol{X}}(\alpha)$. The corresponding labels $y$ are generated from a conditional distribution $\mathcal{P}(y|\boldsymbol{x})$ and are bounded: $|y| \leq D$ for some constant $D > 0$.*

This framework enables a generalization upper bound that depends explicitly on the parameter $\alpha$.

**Theorem 3.4** (Spectrum of generalization on isotropic Beta-radial distributions). *Fix a dataset $\mathcal{D} = \{(\boldsymbol{x}_i,y_i)\}_{i=1}^n$, where each $(\boldsymbol{x}_i,y_i)$ is drawn i.i.d. from $\mathcal{P}(\alpha)$ defined in Assumption 3.3. Then, with probability at least $1-\delta$, for any $\boldsymbol{\theta}\in\boldsymbol{\Theta}_{\mathrm{BEoS}}(\eta,\mathcal{D})$,*

$$\mathrm{Gap}_{\mathcal{P}}(f_{\boldsymbol{\theta}};\mathcal{D}) \lesssim_d \begin{cases} \left(\frac{1}{\eta}-\frac{1}{2}+4M\right)^{\frac{\alpha d}{d^2+4d+3}} M^{\frac{2d^2+7\alpha d+6\alpha}{d^2+4\alpha d+3\alpha}}\,n^{-\frac{\alpha(d+3)}{2(d^2+4\alpha d+3\alpha)}}, & \alpha \geq \frac{3d}{2d-3}, \\[2mm] \left(\frac{1}{\eta}-\frac{1}{2}+4M\right)^{\frac{\alpha d}{d^2+4d+3}} M^{\frac{2d^2+7\alpha d+6\alpha}{d^2+4\alpha d+3\alpha}}\,n^{-\frac{\alpha}{2d+4\alpha}}, & \alpha < \frac{3d}{2d-3}, \end{cases} \quad (9)$$

*where $M := \max\{D, \|f_{\boldsymbol{\theta}}|_{\mathbb{B}_1^d}\|_{L^\infty}, 1\}$ and $\lesssim_d$ hides constants (which may depend on d) and logarithmic factors in n and $(1/\delta)$.*

**Sketch proof of Theorem 3.4.** Suppose $\mathcal{P}_X$ is a $\mathrm{Beta}(\alpha)$-radial distribution $\mathcal{P}_X(\alpha)$ with radial profile $h(r) = 1 - (1-r)^{1/\alpha}$ (Definition 3.2). Then Lemma G.2 gives $\mathbb{P}_{\boldsymbol{X}}(\mathbb{A}_\varepsilon^d) \asymp \varepsilon^\alpha$ and Proposition G.6 gives $g_{\mathcal{P}}(1-\varepsilon) \asymp \varepsilon^{d+2\alpha}$. Substituting into (7), the right-hand side becomes $\tilde{O}(\varepsilon^\alpha) + \tilde{O}\left(\varepsilon^{-\frac{d(d+2\alpha)}{2d+3}} n^{-\frac{d+3}{4d+6}}\right)$. Morally[2], optimizing over $\varepsilon$ yields the upper bounds in Theorem 3.4, see Appendix G for details.

In the proof of the upper bound using (7), we sacrifice the shallow region $\mathbb{A}_\varepsilon^d$, but this is not just a proof artifact. Our lower bound constructions place localized ReLU atoms on disjoint spherical caps in the shallow shell where $g_{\mathcal{P}}$ is very small, so these neurons can develop large path norm at small cost. Combinations of these boundary-supported ReLUs form a family of hard-to-distinguish networks (see Construction H.4), yielding Theorem 3.5.

**Theorem 3.5** (Generalization gap lower bound). *Let $\mathcal{P}$ be any joint distribution of $(\boldsymbol{x}, y)$ where the marginal distribution of $\boldsymbol{x}$ is $\mathcal{P}_X(\alpha)$ and $y$ is supported on $[-1, 1]$. Let $\mathcal{D}_n = \{(\boldsymbol{x}_j, y_j)\}_{j=1}^n$ be a dataset of $n$ i.i.d. samples from $\mathcal{P}$. Let $\widehat{R}_{\mathcal{D}_n}(f)$ be any empirical risk estimator for the true risk $R_{\mathcal{P}}(f) := \mathbb{E}_{(\boldsymbol{x},y)\sim\mathcal{P}}[(f(\boldsymbol{x}) - y)^2]$. Then,*

$$\inf_{\widehat{R}} \sup_{\mathcal{P}} \mathbb{E}_{\mathcal{D}_n}\left[\sup_{\boldsymbol{\theta}\in\boldsymbol{\Theta}_g(\mathbb{B}_1^d;1,1)} \left|R_{\mathcal{P}}(f_{\boldsymbol{\theta}}) - \widehat{R}_{\mathcal{D}_n}(f_{\boldsymbol{\theta}})\right|\right] \gtrsim_{d,\alpha} n^{-\frac{2\alpha}{d-1+2\alpha}}.$$

The smaller $\alpha$ is, the more mass concentrates in $\mathbb{A}_\varepsilon^d$, the more disjoint spherical caps that each carry comparable probability mass (see Figure 1(b)&(c)), the more memorization capacity the network sustain. In the extreme case where $\alpha \to 0$ and the law of $\boldsymbol{X}$ approaches $\mathrm{Uniform}(\mathbb{S}^{d-1})$, each datapoint can be isolated by a network and fitted perfectly within the BEoS regime, see Appendix I.

**Theorem 3.6** (Flat interpolation with width $\leq n$). *Assume that $\{(\boldsymbol{x}_i, y_i)\}_{i=1}^n$ is a dataset with $\boldsymbol{x}_i \in \mathbb{S}^{d-1}$ and pairwise distinct inputs. Then there exists a width $K \leq n$ network of the form (1) that interpolates the dataset and whose Hessian operator norm satisfies*

$$\lambda_{\max}\left(\nabla_{\boldsymbol{\theta}}^2 \mathcal{L}\right) \leq 1 + \frac{D^2 + 2}{n}. \tag{10}$$

*If we remove the output bias parameter $\beta$ in (1), then $\lambda_{\max}\left(\nabla_{\boldsymbol{\theta}}^2 \mathcal{L}\right) \leq \frac{D^2+2}{n}$.*

**Remark 3.7.** *If we remove the hidden bias term $b$ and assume a uniform spherical distribution, the model gets harder to shatter the data, and thus yields a generalization bound of approximately $\tilde{O}(n^{-1/4})$, which is compatible with the rates in (Wu and Su, 2023).*

## 3.2 A UNIFIED FRAMEWORK: DATA SHATTERABILITY PRINCIPLE

The trade-off in (6) and (7) shows that generalization is governed by the probability mass located in the $T$-deep region. Intuitively, the more mass lies in deeper regions, the stronger the implicit regularization. To quantify this geometric distribution, we introduce the following scalar summary.

**Definition 3.8.** *Given a distribution $\mathcal{P}_X$ on $\mathbb{R}^d$, its **half-space-depth quantile function** is $\Psi_{\mathcal{P}_X}(T) := \mathbb{P}_{\boldsymbol{X}\sim\mathcal{P}_X}(\mathrm{depth}(\boldsymbol{X}, \mathcal{P}_X) \geq T)$, for $T \in [0, 1/2]$. We define the **half-space-depth concentration index** $\mathsf{S}_{\mathrm{DQ}}(\mathcal{P}_X) := \left(\int_0^{1/2} \Psi_{\mathcal{P}_X}(T) \, dT\right)^{-1}$, the reciprocal of the area under $\Psi_{\mathcal{P}_X}$.*

We propose using this concentration index as a proxy for data shatterability, the feasibility of the data being partitioned into disjoint regions. In the isotropic case, the spherical symmetry ensures that one can always arrange a large number of disjoint activation regions (spherical caps), a quantity governed by the packing number of the high-dimensional sphere (see Lemma H.3). Consequently, the feasibility of shattering is strictly determined by the radial concentration: does each such potential cap contain sufficient probability mass? As illustrated in Figure 1(b)&(c), if we require every cap to contain a fixed amount of data, the total number of disjoint caps we can pack becomes a

---

[2]Since the empirical process only guarantee $\sup|g_{\mathcal{D}}(\boldsymbol{u}, t) - g_{\mathcal{P}}(\boldsymbol{u}, t)| \leq \tilde{O}(n^{-1/2})$ with $n = |\mathcal{D}|$, we cannot let $g_{\mathcal{P}}(1-\varepsilon^\star)$ less than $\tilde{O}(n^{-1/2})$, so there is a two-case discussion in Theorem 3.4.

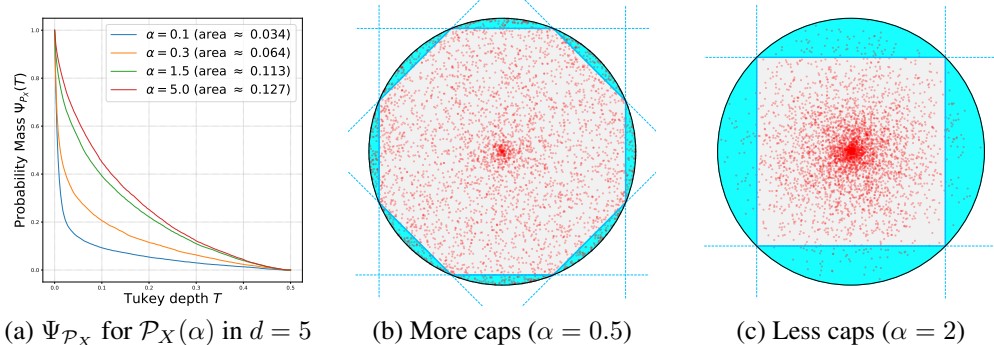

(a) $\Psi_{\mathcal{P}_X}$ for $\mathcal{P}_X(\alpha)$ in $d=5$     (b) More caps ($\alpha = 0.5$)     (c) Less caps ($\alpha = 2$)

Figure 1: **Visualization of isotropic data shatterability. (a)** For isotropic $\mathrm{Beta}(\alpha)$-radial distributions, larger $\alpha$ implies mass is more concentrated centrally (larger area, smaller index). **(b & c)** Red dots depict data points and dashed lines represent neuron boundaries. We visualize the partitioning feasibility by fixing the probability mass within each cap and comparing the number of disjoint caps that can be packed. A distribution with mass near the boundary (small $\alpha$) populates significantly more such disjoint regions than one concentrated at the center ($\alpha = 2$).

measure of shatterability. A large index (small $\alpha$) implies mass accumulates at the boundary, populating these pre-existing packing slots and facilitating the construction of shattering functions for our lower bounds, see Construction H.4. Conversely, a small index (large $\alpha$) starves these regions of data, limiting the noise-fitting feasibility.

A further consistency check comes from the spherical limit: for distributions on a sphere, the depth is zero everywhere. The quantile function collapses and the concentration index diverges, correctly matching the flat interpolation behavior in Theorem 3.6.

**Remark 3.9** (Rectangle vs. Area). *This picture also clarifies why our upper bounds are not tight. On the $T$-deep region, we bound the entire contribution of the network by the worst-case variation over $\Omega_T(\mathcal{P}_X)$, and on the shallow region we use a crude worst-case $L^\infty$ bound times $\mathbb{P}_X(X \notin \Omega_T(\mathcal{P}_X))$, ignoring any additional regularization. Our analysis only exploits the area of a single inscribed rectangle under $T \mapsto \Psi_{\mathcal{P}_X}(T)$, and choosing the optimal $T^\star$ in (6) is equivalent to picking the largest rectangle under the curve, which discards the rest of the underlying area. This gap is exactly where potential improvements to the bounds would have to come from.*

### 3.3 PROVABLE ADAPTATION TO INTRINSIC LOW-DIMENSIONALITY

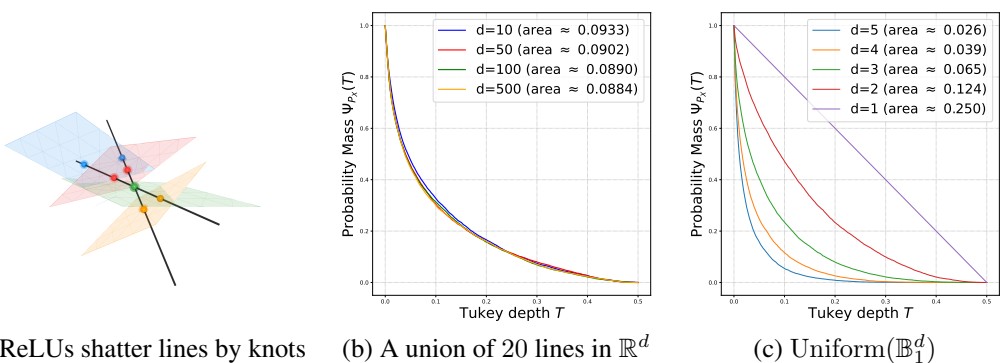

(a) ReLUs shatter lines by knots     (b) A union of 20 lines in $\mathbb{R}^d$     (c) $\mathrm{Uniform}(\mathbb{B}_1^d)$

Figure 2: **Visualization of anisotropic data shatterability. (a)** When data lies on a mixture of lines (black lines), the complex activation boundaries of ReLUs (translucent planes) reduce to a finite set of **knots** (colored dots). This rigidity makes the data harder to shatter than the ambient depth suggests. **(b)** For a union of 20 embedded lines, the curves remain nearly identical across ambient dimensions, matching the prediction by Theorem 3.10. **(c)** As a comparion, the curves for $\mathrm{Uniform}(\mathbb{B}_1^d)$ degenerates and reveal the curse of dimensionality predicted in (Liang et al., 2025).

We now return to anisotropic data. Here, the connection between the concentration index and generalization becomes heuristic because the geometric structure restricts the available partitioning directions. Unlike the isotropic case, where we can leverage the packing number of the ambient sphere to form disjoint regions, low-dimensional structures constrain the effective directions. For instance, on a 1D line, the high-dimensional spherical caps effectively degenerate into simple intervals or knots (Figure 2a). This collapse drastically reduces the number of possible disjoint partitions, even if the data has low ambient depth (high concentration index). Consequently, for structured data, the index serves as a conservative estimate of difficulty. Nevertheless, the principle of depth adaptation still holds: the implicit regularization adapts to the *intrinsic* structure rather than the ambient dimension.

Formalizing this, suppose the feature marginal is a finite mixture $\mathcal{P}_X = \sum_{j=1}^{J} \pi_j \mathcal{P}_{X,j}$. Once the depth-quantile curve $\Psi_{\mathcal{P}_{X,j}}(T)$ of each component is understood, the law of total probability yields

$$\Psi_{\mathcal{P}_X}(T) = \mathbb{P}_{\boldsymbol{X}}\big(\text{depth}(\boldsymbol{X}, \mathcal{P}_X) \geq T\big) = \sum_{j=1}^{J} \pi_j \, \mathbb{P}_{\boldsymbol{X}}\big(\text{depth}(\boldsymbol{X}, \mathcal{P}_X) \geq T \mid Z = j\big). \quad (11)$$

For any fixed $j$, the mixture cannot reduce all halfspace probabilities contributed by component $j$ by more than a factor $\pi_j$ (i.e., $\text{depth}(\boldsymbol{x}, \mathcal{P}_X) \geq \pi_j \, \text{depth}(\boldsymbol{x}, \mathcal{P}_{X,j})$). Hence, at the level of depth-quantile curves, this shows that $\Psi_{\mathcal{P}_X}$ is controlled by the collection $\{\Psi_{\mathcal{P}_{X,j}}\}_{j=1}^{J}$, and the area under $\Psi_{\mathcal{P}_X}$ behaves like a mixture-weighted average of the areas under the $\Psi_{\mathcal{P}_{X,j}}$.

In this heuristic picture, if each component is essentially low-dimensional, then the concentration index of $\Psi_{\mathcal{P}_X}$ is governed by the embedded low-dimensional components and hence by the intrinsic dimension rather than the ambient one. Formally, we have the following theorem.

**Theorem 3.10** (Generalization bound for mixture models). *Given a data distribution $\mathcal{P}$ on $\mathbb{R}^d \times \mathbb{R}$ whose feature marginal satisfies $\mathcal{P}_X = \sum_{j=1}^{J} \pi_j \mathcal{P}_{X,j}$, where each $\mathcal{P}_{X,j}$ is the uniform distribution on the unit ball in an $m$-dimensional affine subspace $V_j \subset \mathbb{R}^d$, let $\mathcal{D} = \{(\boldsymbol{x}_i, y_i)\}_{i=1}^{n}$ be a dataset of $n$ i.i.d. samples drawn from $\mathcal{P}$. Then, with probability at least $1 - \delta$,*

$$\sup_{\boldsymbol{\theta} \in \boldsymbol{\Theta}_{\text{BEoS}}(\eta, \mathcal{D})} \text{Gap}_{\mathcal{P}}(f_{\boldsymbol{\theta}}; \mathcal{D}) \lesssim_d \left(\frac{1}{\eta} - \frac{1}{2} + 4M\right)^{\frac{m}{m^2 + 4m + 3}} M^2 \, J^{\frac{4}{m}} \, n^{-\frac{1}{2m+4}} + M^2 J \sqrt{\frac{1}{2n}}, \quad (12)$$

*where $M := \max\{D, \|f_{\boldsymbol{\theta}}|_{\mathbb{B}_1^Y}\|_{L^\infty}, 1\}$ and $\lesssim_d$ hides constants (which may depend on d) and logarithmic factors in $J/\delta$ and $n$.*

**Sketch proof of Theorem 3.10.** The detailed proof is in Appendix F. The argument instantiates the above shatterability picture at the level of the BEoS-induced regularization. For each component $j$, we define a local weight function $g_j$ using only samples with $\boldsymbol{X} \in V_j$, and Lemma F.4 formalizes this as a uniform domination $g \gtrsim g_j$. Consequently, the restriction of $f_{\boldsymbol{\theta}}$ to $V_j$ has controlled $g_j$-weighted variation with the same order of bound. Crucially, if we restrict the network to a single $m$-dimensional subspace $V_j$, a neuron's activation is governed not by its full weight vector $\boldsymbol{w}_k$, but solely by $\text{proj}_{V_j} \boldsymbol{w}_k$, since the component of $\boldsymbol{w}_k$ that is orthogonal to $V_j$ is "invisible" to the data on $V_j$. This projection mechanism mathematically formalizes how linearly low-dimensional structures limit the model's shatterability (as visualized by the knots in Figure 2), and thus refine the characterization of network's *effective capacity*, see Theorem F.3. Moreover, some insight of this low-dimensional adaptation from the perspective of gradient dynamics is stated in Appendix B.2.

## 4 EXPERIMENTS

In this section, we present empirical verification of both our theoretical claims and *proof strategies*.

### 4.1 EMPIRICAL VERIFICATION OF THE GENERALIZATION UPPER BOUNDS

We test two predictions of our theory using synthetic data and two-layer ReLU networks of width 1000 trained with MSE loss and vanilla GD with learning rate 0.4 for 20000 epochs. The synthetic training data is produced by fixing a ground-truth function $f$ (ReLU networks or quadratic functions) to noisy labels $y_i = f(\boldsymbol{x}_i) + \xi_i$, where $\xi_i$ is an i.i.d Gaussian noise. Generalization gap is measured by the *true MSE* $\mathbb{E}_{\mathcal{D}}[(\hat{f}(\boldsymbol{X}) - f(\boldsymbol{X}))^2]$ on the training set. In other words, this measures the resistance to memorize noise. Theory predicts Error $\lesssim n^{-c}$ with a geometry-dependent exponent

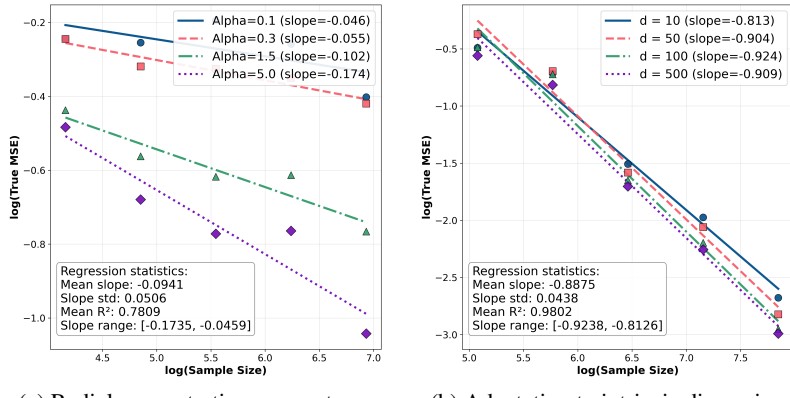

(a) Radial concentration parameter $\alpha$      (b) Adaptation to intrinsic dimension

Figure 3: **How data geometry controls generalization.** **(a)** Fixed ambient dimension $d = 5$ with isotropic Beta-radial distributions (Definition 3.2) for $\alpha \in \{0.1, 0.3, 1.5, 5.0\}$. Larger $\alpha$ yields steeper slopes in the log–log error curve, consistent with improved rates as probability mass concentrates away from the boundary. **(b)** Union of $J = 20$ lines ($m = 1$) embedded in $\mathbb{R}^d$ with $d \in \{10, 50, 100, 500\}$. The regression slopes remain nearly constant across different $d$, showing that generalization adapts to intrinsic rather than ambient dimension.

$c$, so we plot $\log(\text{clean MSE})$ against $\log n$ and estimate the slope by OLS. For each sample size $n$, we train on $n$ i.i.d. examples and report their true MSE. For each setup, we ran experiments with 6 random seeds and averaged the results. The results are summarized in Figure 3.

## 4.2 How Data Geometry Affects Representation Learning

We study how data geometry shapes the *representation* selected by GD at the BEoS regime through *data activation rate* of neurons. Given a neuron $v_k \phi(\boldsymbol{w}_k^\top \boldsymbol{x} - b_k)$ in the neural network, its data activation rate is defined as $\frac{1}{n} \sum_{i=1}^{n} \mathbb{1}\{\boldsymbol{w}_k^\top \boldsymbol{x}_i > b_k\}$, which is exactly the probability term in the definition of the weight function $g$ in (28). Low data activation rate means the neuron fires on a small portion of the data.

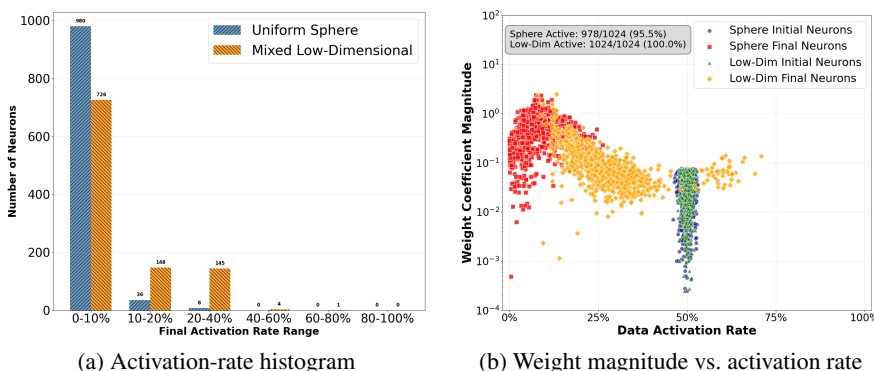

(a) Activation-rate histogram      (b) Weight magnitude vs. activation rate

Figure 4: **Neuron activation statistics under different geometries.** **(a)** On the uniform sphere, most neurons fire on less than $10\%$ of the data, indicating highly specialized ReLUs as we predict in Theorem 3.5 and Theorem 3.6. On the low-dimensional mixture, many neurons fire on $10\text{-}40\%$ of the data, reflecting broader feature reuse. **(b)** Scatter of weight coefficient magnitude versus activation rate. On the sphere, GD produces many low-activation neurons with large coefficients. On the low-dimensional mixture, neurons spread to medium activation rates with moderate coefficients.

To verify it empirically, we compare two input distribution in $\mathbb{R}^{50}$: (i) uniform distribution on a sphere and (ii) a union of 20 lines by training ReLU networks with the same recipe and initialization. As a result, the ReLU network trained on the sphere interpolates the noisy label quickly with final true MSE $1.0249 \approx$ noise level ($\sigma^2 = 1$), while the the ReLU network trained on a union of lines resist to overfitting with final true MSE $0.07 \approx 0$ (more details appear in Appendix C). Notably, the trained representations are presented in Figure 4. In particular, GD empirically finds our lower bound construction below the edge of stability.

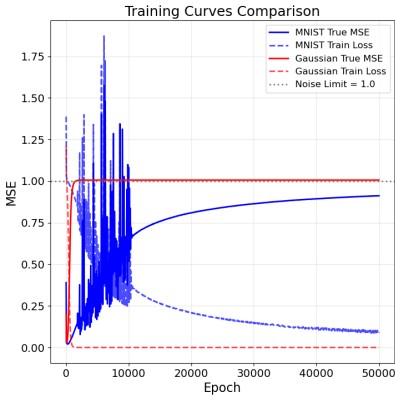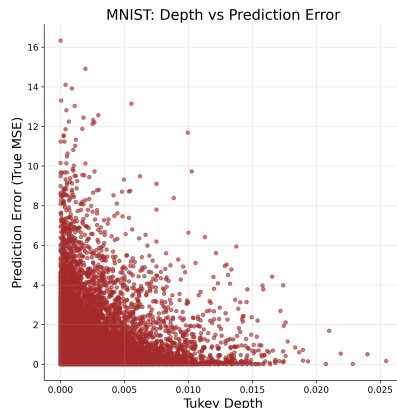

Figure 5: **Data geometry and memorization on MNIST. Left panel:** Comparison of training curves under the same ground-truth predictor with Gaussian inputs versus MNIST inputs ($n = 30000$). GD on the Gaussian data set quickly interpolates, while MNIST resists overfitting for tens of thousands of steps. **Right panel:** Prediction error against half-space depth for MNIST samples. Shallow points (low depth) exhibit larger errors. This region refers to "highly shatterable region".

### 4.3 EMPIRICAL EVIDENCE FOR THE DATA SHATTERABILITY PRINCIPLE

Our theory assumes data supported exactly on a mixture of low-dimensional subspaces. In practice, real datasets are only approximately low-dimensional, as highlighted in the literature on subspace clustering (Vidal et al., 2016; Elhamifar and Vidal, 2013). For instance, MNIST images do not perfectly lie on a union of lines or planes, but still exhibit strong correlations that concentrate them near such structures. Our experiments (Figure 5, more details in Appendix C.2) show that even this approximate structure has a pronounced effect: compared to Gaussian data of the same size, GD on MNIST requires orders of magnitude more iterations before mildly overfitting solutions emerge. This demonstrates that our theoretical prediction is not fragile: generalization benefits from low-dimensional structure across a spectrum.

## 5 DISCUSSION AND FURTHER QUESTIONS

In this work, we present a mechanism explaining *how* data geometry governs the implicit bias of neural networks trained below the Edge of Stability. We introduce the principle of "data shatterability," demonstrating that geometries resistant to shattering guide gradient descent towards discovering shared, generalizable representations.

**Limitations.** While our theoretical framework instantiates the principle of data shatterability, there are limitations to our current analysis. First, the proposed *half-space-depth concentration index* ($S_{DQ}$) serves as a proxy primarily for isotropic distributions and extending this scalar metric to quantify the shatterability of arbitrary, anisotropic, structured data remains a non-trivial challenge. Second, our results are derived for two-layer ReLU networks to maintain tractability in the function-space analysis. Generalizing these bounds to deep and complicated networks presents substantial theoretical hurdles, particularly in characterizing how the EoS-induced regularity constraint propagates through hierarchical layers without becoming ensuring the bounds remain non-vacuous.

**Further Questions.** Our framework opens several promising avenues for future research. A central question is the connection between shatterability and optimization. The observation that a flip side of being prone to overfitting is often faster optimization leads to a natural hypothesis: are high-shatterability distributions easier to optimize? This, in turn, raises further questions about the role of normalization techniques. For instance, do normalization techniques like Batch Norm accelerate training precisely by enforcing more isotropic, and thus more shatterable, representations at each layer? This line of inquiry extends naturally to deep networks, where hidden layers not only sense the initial data geometry but actively create a new "representation geometry". Can our principles be translated to the understanding of representation geometry? Finally, this framework may offer a new lens to understand architectural inductive biases. For example, do CNNs generalize well precisely because their local receptive fields impose an architectural constraint that inherently reduces the model's ability to shatter the data, forcing it to learn local, reusable features? Answering such questions, alongside developing a quantifiable metric for shatterability, remains a key direction.

## 6 ACKNOWLEDGMENTS

The research was partially supported by NSF Award # 2134214. Tongtong Liang thanks Dan Qiao and Zihan Shao for providing helpful suggestions.

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

## Supplementary Materials

## A  The Use of Large Language Models (LLMs)

In accordance with the ICLR 2026 policy on the responsible use of LLMs, we disclose the following. We employed commercial LLM services during manuscript preparation. Specifically, we used Gemini 2.5 Pro, ChatGPT 5, and DeepSeek to assist with language polishing, literature search, and consistency checks of theoretical derivations. We further used Claude 4 and Cursor to help generate experimental code templates. Importantly, all research ideas, theoretical results, and proof strategies

originated entirely from the authors. The LLMs were used solely as productivity aids and did not contribute novel scientific content.

# B  MORE RELATED WORKS AND DISCUSSIONS

## B.1  MORE RELATED WORKS

**How we "rethink" generalization.** Our shatterability principle provides a theoretical account of the discrepancy noted by Zhang et al. (2017): networks fit Gaussian noise much faster than real images with random labels. Gaussian inputs concentrate on a thin spherical shell and are highly shatterable, while CIFAR-10 exhibits unknown low-dimensional structure that resists shattering. Strong generalization arises in practice because gradient descent implicitly exploits this non-shatterable geometry of the real world data. We conduct a similar experiment from the perspective of generalization in Section 4.3 (see Figure 5).

**Revisit data augmentation.** Mixup forms convex combinations of inputs and labels and encourages approximately linear predictions along these segments (Zhang et al., 2018). The added in-between samples penalize solutions that memorize isolated points with sharply varying piecewise-linear behavior. For example, on spherical-like data that ReLU units can easily shatter, such memorization incurs high loss on the mixed samples, which suppresses shattering-type separators. Prior work mostly views Mixup as a data-dependent regularizer that improves generalization and robustness (Zhang et al., 2021). Our analysis complements this view by tracing the effect to the implicit bias of gradient descent near the edge of stability and by linking the gains to a reduction in data shatterability induced by interpolation in low-density regions.

**Activation-based network pruning.** Empirical works have shown that pruning strategies based on neuron activation frequency, such as removing neurons with low activation counts, can even improve the test performance after retraining (Hu et al., 2016; Ganguli and Chong, 2024). This coincide with our theory: such rare-firing neurons may be harmful to generalization and pruning these neurons help models to learn more generalizable features.

**Subspace and manifold hypothesis.** A common modeling assumption in high-dimensional learning is that data lies on or near one or several low-dimensional subspaces embedded in the ambient space, especially in image datasets where pixel values are constrained by geometric structure and are well-approximated by local subspaces or unions of subspaces (Vidal et al., 2016). In particular, results in sparse representation and subspace clustering demonstrate that such structures enable efficient recovery and segmentation of high-dimensional data into their intrinsic subspaces (Elhamifar and Vidal, 2013). This also extends to a more general framework of the manifold hypothesis (Fefferman et al., 2016).

**Capacity of neural networks.** The subspace and manifold hypotheses have important implications for the capacity and generalization of neural networks. When data lies near low-dimensional subspaces and manifolds, networks can achieve expressive power with significantly fewer parameters, as the complexity of the function to be learned is effectively constrained by the subspace dimension rather than the ambient dimension (Poggio and Liao, 2017; Cloninger and Klock, 2021; Kohler et al., 2022). However, these results focus only on expressivity and the existence of neural networks to learn efficiently on this data.

**Interpolation, Benign overfitting and data geometry.** Benign-overfitting (Bartlett et al., 2020) studies the curious phenomenon that one can interpolate noisy labels (i.e., 0 training loss) while consistently learn (excess risk $\to 0$ as $n$ gets larger). Joshi et al. (2024) establishes that overfitting in ReLU Networks is not benign in general, but it could become more benign as the input dimension grows (Kornowski et al., 2024) in the isotropic Gaussian data case. Our results suggest that such conclusion may be fragile under *low-dimensional or structured* input distributions. On a positive note, our results suggest that in these cases, generalization may follow from edge-of-stability, which applies without requiring interpolation.

**Implicit bias of gradient descent.** A rich line of work analyzes the implicit bias of (stochastic) gradient descent (GD), typically through optimization dynamics or limiting kernels (Arora et al., 2019; Mei et al., 2019; Jin and Montúfar, 2023). In contrast, we do not analyze the time evolution per se; we characterize the *function spaces* that GD tends to realize at solutions. Our results highlight

a strong dependence on the *input distribution*: even for the same architecture and loss, the induced hypothesis class (and thus generalization) changes as the data geometry changes, complementing prior dynamics-centric views.

**Edge of Stability (EoS) and minima stability.** The EoS literature primarily seeks to explain when and why training operates near instability and how optimization proceeds there (Cohen et al., 2020; Kong and Tao, 2020; Arora et al., 2022; Ahn et al., 2022; Damian et al., 2024). Central flows offer an alternative viewpoint on optimization trajectories that also emphasizes near-instability behavior (Cohen et al., 2025). Closest to our work is the line on *minima stability* (Ma and Ying, 2021; Mulayoff et al., 2021; Nacson et al., 2023), which links Hessian spectra and training noise to the geometry of solutions but largely leaves generalization out of scope. We leverage the EoS/minima-stability phenomena to *define* and analyze a data-distribution-aware notion of stability, showing adaptivity to low-dimensional structure and making explicit how distributional geometry shapes which stable minima GD selects.

**Nonparametric function estimation with neural networks.** The notion of generalization gap is closely related to the estimation error. It is well known that neural networks are minimax optimal estimators for a wide variety of functions (Suzuki, 2018; Schmidt-Hieber, 2020; Kohler and Langer, 2021; Parhi and Nowak, 2023; Zhang and Wang, 2023; Wu and Su, 2023; Yang and Zhou, 2024; Qiao et al., 2024). Outside of the univariate work of Qiao et al. (2024), all prior works construct their estimators via empirical risk minimization problems. Thus, they do not incorporate the training dynamics that arise when training neural networks in practice. In contrast, Qiao et al. (2024); Liang et al. (2025) derive nonparametric guarantees for solutions selected directly by gradient descent dynamics, without assuming even stationary condition. Our work further develops in this direction by modeling how data geometry affect generalization for the gradient trajectories below the edga-of-stability.

**Flatness vs. generalization.** Whether (and which notion of) flatness predicts generalization remains debated. Several works argue sharp minima can still generalize (Dinh et al., 2017), propose information-geometric or Fisher-Rao-based notions (Liang et al., 2019), or develop relative/scale-invariant flatness measures (Petzka et al., 2021). We focus on the *largest* curvature direction (i.e., $\lambda_{\max}$) motivated by EoS/minima-stability. Our results rigorously prove that flatness in this notion does imply generalization (note that there is no contradiction with Dinh et al. (2017)), but it depends on data distribution.

**Linear regions of neural networks.** Our research connects to a significant body of work that investigates the shattering capability of neural networks by quantifying their linear activation regions (Hanin and Rolnick, 2019a;b; Hanin et al., 2021; Montúfar et al., 2014; Serra et al., 2017). Other empirical work has meticulously characterized the geometric properties of linear regions shaped by different optimizers (Zhang and Wu, 2020). Particularly, (Tiwari and Konidaris, 2022) consider the how these linear regions intersect with data manifolds. These analyses primarily leverage the number of regions to characterize the expressive power of deep networks, while our work shifts the focus on the generalization performance of shallow networks at the EoS regime.

## B.2 A GRADIENT-DYNAMICS PERSPECTIVE ON STABILITY AND DATA GEOMETRY

We now discuss our results from the viewpoint of *gradient dynamics*.

**Why Shattering Neural Networks May Be Dynamically Stable.** Here we paraphrase and adapt the discussion from (Liang et al., 2025, Appendix A.4). The BEoS condition defines a set of dynamically stable parameter states, and the data-dependent weight function $g_{\mathcal{D}}(\boldsymbol{u}, t)$ provides a static summary of how expensive it is to place a ReLU ridge at orientation $\boldsymbol{u}$ and threshold $t$. Small values of $g_{\mathcal{D}}(\boldsymbol{u}, t)$ indicate weak stability constraints in that region of parameter space, so a neuron aligned with $(\boldsymbol{u}, t)$ can carry a large coefficient while still satisfying the BEoS curvature bound. In highly shatterable geometries, the shallow shell contains many such directions with tiny $g_{\mathcal{D}}$, creating ample room inside the stable set for high-magnitude, sparsely activating neurons supported on disjoint caps.

This static picture is closely tied to the actual gradient dynamics. If the dataset is highly shatterable in the sense of the half-space-depth concentration index, a neuron's activation boundary can easily drift toward regions where it fires on only a few data points. Once those few points are already well-fitted,

the gradient contributions in (13) become small and localized, so the neuron experiences almost no force pulling it back toward more central regions of the data cloud. Its parameters become effectively "stuck" near the boundary, and the corresponding directions in the loss landscape remain flat enough to satisfy the BEoS condition. Our lower bound constructions exploit exactly this mechanism: by arranging many such trapped, boundary-supported neurons on disjoint caps, we obtain shattering networks that interpolate, remain dynamically stable, and yet are statistically hard to learn. Although our analysis is carried out for ReLU, where hard sparsity makes this effect particularly transparent, the underlying "weak-gradient trapping" mechanism suggests that similar phenomena may persist for other activations with rapidly decaying gradients away from their transition region.

**How GD Adapts to Low-dimensionality Dynamically.** Recall our notations: for a two-layer ReLU network

$$f_{\boldsymbol{\theta}}(\boldsymbol{x}) = \sum_{k=1}^{K} v_k \, \phi(\boldsymbol{w}_k^{\mathsf{T}} \boldsymbol{x} - b_k) + \beta$$

trained on data $\{(\boldsymbol{x}_i, y_i)\}_{i=1}^{n}$ with empirical loss $\mathcal{L}(\boldsymbol{\theta}) = \frac{1}{2n} \sum_{i=1}^{n} \ell(f_{\boldsymbol{\theta}}(\boldsymbol{x}_i), y_i)$, the gradient with respect to a hidden weight $\boldsymbol{w}_k$ has the form

$$\nabla_{\boldsymbol{w}_k} \mathcal{L}(\boldsymbol{\theta}) = \frac{1}{n} \sum_{i=1}^{n} \ell'(f_{\boldsymbol{\theta}}(\boldsymbol{x}_i), y_i) \, v_k \, \phi'(\boldsymbol{w}_k^{\mathsf{T}} \boldsymbol{x}_i - b_k) \, \boldsymbol{x}_i = \sum_{i=1}^{n} \alpha_{k,i}(\boldsymbol{\theta}) \, \boldsymbol{x}_i. \tag{13}$$

Thus, at every step of gradient descent, the update of $\boldsymbol{w}_k$ is a linear combination of input vectors. In informal terms, the *shape* of the gradient field is inherited from the shape of the data cloud: the dynamics cannot move in arbitrary directions in parameter space, but only along directions induced by the training inputs and the neuron-specific activation pattern.

In the idealized case where all $\boldsymbol{x}_i$ lie in a linear subspace $V \subset \mathbb{R}^d$, (13) already suggests a form of intrinsic-dimension adaptation: the trajectory of each hidden weight lives in an affine translate of $V$, so any meaningful notion of effective complexity should depend on $\dim V$ rather than on the ambient dimension. However, as soon as we move to more realistic geometries—affine subspaces $a + V$ or mixtures of $m$-dimensional components—the global span of the data can easily be full-dimensional. In those settings, the observation that gradients lie in $\mathrm{span}\{\boldsymbol{x}_i\}$ is nearly vacuous: it no longer encodes the *structured* way in which the data constrain the dynamics, and by itself it does not yield intrinsic-dimension generalization bounds.

Our contribution is to make this stability-based picture interact explicitly with the *geometry of the data*. For structured distributions such as mixtures of $m$-dimensional balls, the data-dependent weight function $g_{\mathcal{D}}(\boldsymbol{u}, t)$ inherits a corresponding structure. A hyperplane that cuts through the thick interior of an $m$-dimensional component activates on many of its points. Neurons aligned with such hyperplanes receive large gradients and are strongly constrained by the stability condition, which is reflected in large values of $g_{\mathcal{D}}(\boldsymbol{u}, t)$. In contrast, hyperplanes that only touch shallow caps or skim the boundary fire on very few points; neurons aligned with these directions see much weaker "gradient pressure" and correspondingly small values of $g_{\mathcal{D}}(\boldsymbol{u}, t)$, allowing them to carry high norm and implement localized features.

In this way, the gradient-dynamics perspective can be summarized as

$$\text{data geometry} \implies \text{gradient geometry} \implies \text{stability-induced regularization},$$

with $g_{\mathcal{D}}$ acting as the bridge between dynamics and capacity control. Stability does not merely say that gradients live in the span of the data; through $g_{\mathcal{D}}$, it tells us *which parts* of the data geometry are expensive to fit and *which parts* can host high-norm, sparsely supported neurons. This refinement allows our analysis to turn the qualitative statement that "gradient trajectories are shaped by the data geometry" into quantitative, distribution-dependent generalization bounds that scale with the intrinsic dimension $m$ rather than the ambient dimension $d$.

## C  DETAILS OF EXPERIMENTS

### C.1  EXPERIMENTAL DETAILS FOR SECTION 4.2

Here we provide the full experimental details of the discussion in Section 4.2.

We worked in ambient dimension $d = 50$ with $n = 2000$ training examples. For the *Sphere* condition, samples were drawn uniformly from the unit sphere. For the *Low-dimensional mixture*, we generated data from a mixture of 20 randomly oriented 1-dimensional subspaces uniformly. Labels were produced by a fixed quadratic teacher function with added Gaussian noise of variance 1.

We trained a two-layer ReLU network with hidden width 1024. All models were trained with GD for 10000 epochs using learning rate 0.4 and gradient clipping at 50. The loss function was the squared error against noisy labels, while generalization performance was evaluated by the *true MSE* against the noiseless teacher. For comparability, both datasets shared the same initialization of parameters.

We monitored (i) training loss and true MSE, (ii) Hessian spectral norm estimated by power iteration on random minibatches, and (iii) neuron-level statistics such as activation rate and coefficient magnitude. The training curves are shown in Figure 6 and $\lambda_{\max}(\nabla_{\boldsymbol{\theta}}\mathcal{L})$-curves are shown in Figure 7.

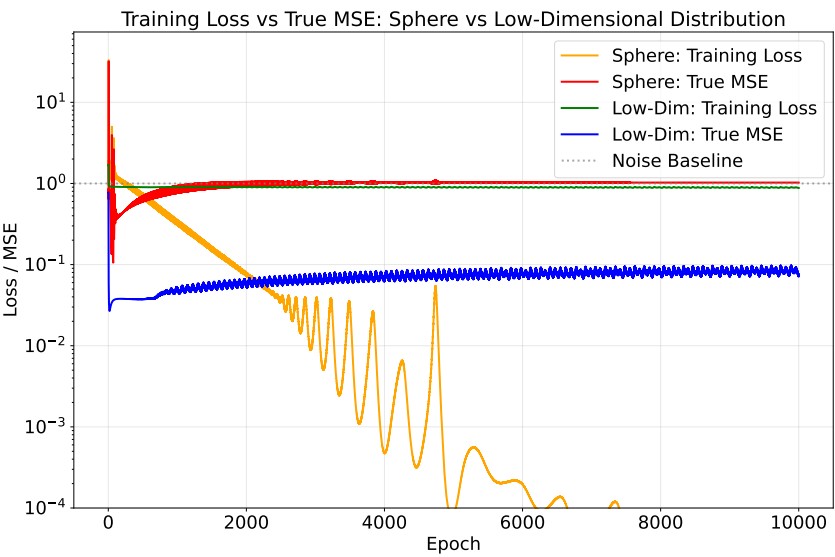

Figure 6: **Training curves on different geometries.** Training loss and clean MSE on Sphere vs. Low-dimensional mixture. We can see GD on sphere interpolate very quickly (before the 2000-th epoch) while the mixed low-dimensional data resist to overfitting.

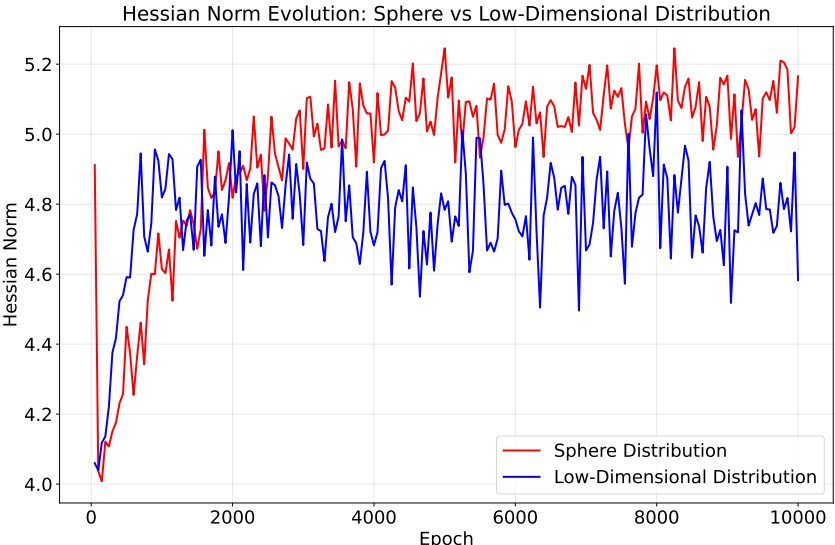

Figure 7: $\lambda_{\max}(\nabla_{\boldsymbol{\theta}}\mathcal{L})$-**curves.** Both of the curves oscillates around $2/\eta = 5$, signaling the edge of stability regime.

## C.2 EXPERIMENTAL DETAILS OF SECTION 4.3

We complement the main experiments with a controlled comparison between real data (MNIST) and synthetic Gaussian noise under the same ground-truth function. The goal is to illustrate how the geometry of real-world data affects the speed and nature of memorization by GD.

We fix a ground-truth predictor $f$ (a two-layer ReLU network) and generate noisy labels

$$y_i = f(\boldsymbol{x}_i) + \xi_i, \qquad \xi_i \sim \mathcal{N}(0,1).$$

We then compare two input distributions of size $n = 30000$:

(i) Gaussian inputs $\boldsymbol{x}_i \sim \mathcal{N}(0, I_d)$ with $d = 784$, and

(ii) MNIST images $\boldsymbol{x}_i \in [0,1]^{784}$ after normalization by $1/255$.

Both datasets are trained with identical architecture (two-layer ReLU neuron network of 512 neurons), initialization, learning rate $\eta = 0.2$, gradient clip threshold 50.

We track both the empirical training loss and the *true MSE* $\frac{1}{n}\sum_{i=1}^{n}(\hat{f}(\boldsymbol{x}_i) - f(\boldsymbol{x}_i))^2$, which measures generalization. The horizontal dotted line at $y = 1$ corresponds to the noise variance and represents the interpolation limit.

Figure 8 shows training curves over the first 5000 epochs. On Gaussian inputs, GD rapidly interpolates: the training loss vanishes and the clean MSE rises to the noise limit within a few hundred steps. On MNIST inputs, GD initially decreases both training loss and clean MSE, entering a prolonged BEoS regime where interpolation is resisted. Only after thousands of epochs does the clean MSE start to increase, suggesting that memorization occurs at a much slower rate.

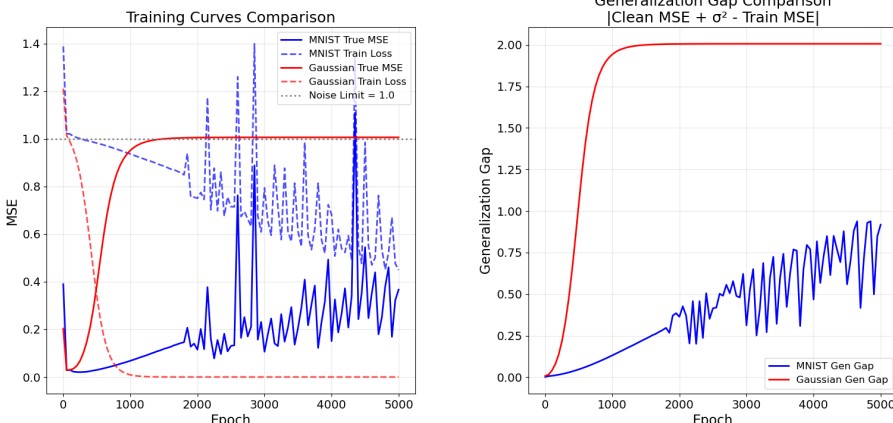

Figure 8: Training curves for Gaussian noise vs. MNIST over the first 5000 epochs. Gaussian quickly interpolates, while MNIST remains in a BEoS regime where clean MSE stays well below the noise level.

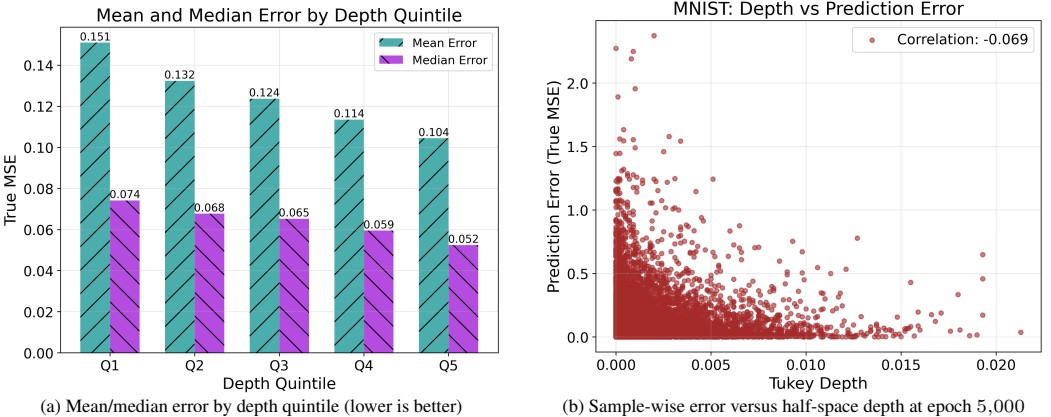

(a) Mean/median error by depth quintile (lower is better)

(b) Sample-wise error versus half-space depth at epoch 5,000

Figure 9: MNIST at 5,000 epochs: deeper points have smaller error. The shallow region produces a long upper tail of errors, consistent with the annulus-interior decomposition used in our upper bounds.

## D  FUNCTIONAL ANALYSIS OF SHALLOW ReLU NETWORKS

### D.1  PATH-NORM AND VARIATION SEMI-NORM OF ReLU NETWORKS

In this section, we summarize some result in (Parhi and Nowak, 2023) and (Siegel and Xu, 2023).

**Definition D.1.** *Let $f_{\boldsymbol{\theta}}(\boldsymbol{x}) = \sum_{k=1}^{K} v_k\, \phi(\boldsymbol{w}_k^{\mathsf{T}}\boldsymbol{x} - b_k) + \beta$ be a two-layer neural network. The (unweighted) path-norm of $f_{\boldsymbol{\theta}}$ is defined to be*

$$\|f_{\boldsymbol{\theta}}\|_{\mathrm{path}} := \sum_{k=1}^{K} |v_k|\, \|\boldsymbol{w}_k\|_2. \tag{14}$$

**Dictionary representation of ReLU networks.**  By the positive 1-homogeneity of ReLU, each neuron can be rescaled without changing the realized function:

$$v_k\, \phi(\boldsymbol{w}_k^{\mathsf{T}}\boldsymbol{x} - b_k) = a_k\, \phi(\boldsymbol{u}_k^{\mathsf{T}}\boldsymbol{x} - t_k), \quad \boldsymbol{u}_k := \frac{\boldsymbol{w}_k}{\|\boldsymbol{w}_k\|_2} \in \mathbb{S}^{d-1},\ t_k := \frac{b_k}{\|\boldsymbol{w}_k\|_2},\ a_k := v_k\, \|\boldsymbol{w}_k\|_2.$$

Hence $f_{\boldsymbol{\theta}}$ admits the normalized finite-sum form

$$f(\boldsymbol{x}) = \sum_{k=1}^{K'} a_k\, \phi(\boldsymbol{u}_k^{\mathsf{T}}\boldsymbol{x} - t_k) + \boldsymbol{c}^{\mathsf{T}}\boldsymbol{x} + c_0. \tag{15}$$

Let the (ReLU) ridge dictionary be $\mathscr{D}_\phi := \left\{ \phi(\boldsymbol{u}^{\mathsf{T}} \cdot -t) :\ \boldsymbol{u} \in \mathbb{S}^{d-1},\ t \in \mathbb{R} \right\}$. We study the *over-parameterized, width-agnostic* class given by the *union over all finite widths*

$$\mathcal{F}_{\mathrm{fin}} := \bigcup_{K \geq 1} \left\{ \sum_{k=1}^{K} a_k\, \phi(\boldsymbol{u}_k^{\mathsf{T}} \cdot -t_k) + \boldsymbol{c}^{\mathsf{T}}(\cdot) + c_0 \right\}, \tag{16}$$

and measure complexity by the minimal path-norm needed to realize $f$:

$$\|f\|_{\mathrm{path,min}} := \inf \left\{ \|f_{\boldsymbol{\theta}}\|_{\mathrm{path}} :\ f_{\boldsymbol{\theta}} \equiv f \text{ of the form } (15) \right\}.$$

**From finite sums to a *width-agnostic* integral representation.**  To analyze $\mathcal{F}_{\mathrm{fin}}$ without committing to a fixed width $K$, we pass to a convex, measure-based description that *represents the closure/convex hull of* (16). Specifically, let $\nu$ be a finite signed Radon measure on $\mathbb{S}^{d-1} \times [-R, R]$ and consider

$$f(\boldsymbol{x}) = \int_{\mathbb{S}^{d-1} \times [-R,R]} \phi(\boldsymbol{u}^{\mathsf{T}}\boldsymbol{x} - t)\, \mathrm{d}\nu(\boldsymbol{u}, t) + \boldsymbol{c}^{\mathsf{T}}\boldsymbol{x} + c_0. \tag{17}$$

Any finite network (15) corresponds to the *sparse* measure $\nu = \sum_{k=1}^{K} a_k\, \delta_{(\boldsymbol{u}_k, t_k)}$, and conversely sparse measures yield finite networks. Thus, (17) is a *width-agnostic relaxation* of (14), not an assumption of an infinite-width limit.

**Definition D.2.** *The (unweighted) variation (semi)norm*

$$|f|_{\mathrm{V}} := \inf \left\{ \|\nu\|_{\mathcal{M}} :\ f \text{ admits } (15) \text{ for some } (\nu, c, c_0) \right\}, \tag{18}$$

*where $\|\nu\|_{\mathcal{M}}$ is the total variation of $\nu$.*

*For the compact region $\Omega = \mathbb{B}_R^d$, we define the bounded variation function class as*

$$\mathrm{V}_C(\Omega) := \left\{ f : \Omega \to \mathbb{R} \mid f = \int_{\mathbb{S}^{d-1} \times [-R,R]} \phi(\boldsymbol{u}^{\mathsf{T}}\boldsymbol{x} - t)\, \mathrm{d}\nu(\boldsymbol{u}, t) + \boldsymbol{c}^{\mathsf{T}}\boldsymbol{x} + b,\ |f|_{\mathrm{V}} \leq C \right\}. \tag{19}$$

Specifically, by identifying (15) with the atomic measure $\nu = \sum_k a_k \delta_{(\boldsymbol{u}_k, t_k)}$, we have

$$|f|_{\mathrm{V}} \leq \sum_k |a_k| = \|f_{\boldsymbol{\theta}}\|_{\mathrm{path}}, \quad \text{hence} \quad |f|_{\mathrm{V}} \leq \|f\|_{\mathrm{path,min}}.$$

Conversely, the smallest variation needed to represent $f$ equals the smallest path-norm across all finite decompositions,

$$\|f\|_{\mathrm{path,min}} = |f|_{\mathrm{V}}. \tag{20}$$

Thus, the variation seminorm (18) is the *nonparametric* counterpart of the path-norm, which captures the same notion of complexity but *without fixing the width $K$.*

**Remark D.3** ("Arbitrary width" $\neq$ "infinite width"). *Our analysis concerns $\mathcal{F}_{\mathrm{fin}}$ in (16), i.e., the union over all finite widths. The integral model (17) is a convexification/closure of this union that facilitates analysis and regularization; it does not assume an infinite-width limit. In variational training with a total-variation penalty on $\nu$, first-order optimality ensures sparse solutions (finite support of $\nu$), which correspond to finite-width networks. Thus, all results in this paper apply to arbitrary (but finite) width, and the continuum measure is only a device to characterize and control $\|f\|_{\mathrm{path,min}}$.*

## D.2 Total Variation Semi-norm on Radon Domain

We now connect the (unweighted) variation semi-norm of shallow ReLU networks to an analytic description on the *Radon domain*. Our presentation follows (Parhi and Nowak, 2021; 2022; 2023; 2026; Parhi and Unser, 2024).

**Definition D.4.** *For a function $f : \mathbb{R}^d \to \mathbb{R}$ and $(\boldsymbol{u}, t) \in \mathbb{S}^{d-1} \times \mathbb{R} := \mathbb{S}^{d-1} \times \mathbb{R}$, the Radon transform and its dual are defined by*

$$\mathscr{R}f(\boldsymbol{u}, t) = \int_{\{\boldsymbol{x}: \boldsymbol{u}^\mathsf{T}\boldsymbol{x}=t\}} f(\boldsymbol{x}) \, \mathrm{d}s(\boldsymbol{x})$$

$$\mathscr{R}^* \{\Phi\} (\boldsymbol{x}) = \int_{\mathbb{S}^{d-1}} \Phi(\boldsymbol{u}, \boldsymbol{u}^\mathsf{T}\boldsymbol{x}) \, \mathrm{d}\sigma(\boldsymbol{u}).$$

**Proposition D.5** (Filtered backprojection (Radon inversion)). *There exists $c_d > 0$ such that*

$$c_d \, f \;=\; \mathscr{R}^* \left\{ \Lambda^{d-1} \mathscr{R}f \right\},$$

*where $\Lambda^{d-1}$ acts in the $t$-variable with Fourier symbol $\widehat{\Lambda^{d-1}\Phi}(\boldsymbol{u}, \omega) \propto |\omega|^{d-1} \hat{\Phi}(\boldsymbol{u}, \omega)$.*

**Definition D.6** (Second-order Radon total variation (ReLU case)). *The (second-order) Radon total-variation seminorm is*

$$\mathscr{R}\mathrm{TV}^2(f) := \left\| \mathscr{R} \left\{ (-\Delta)^{\frac{d+1}{2}} f \right\} \right\|_{\mathcal{M}(\mathbb{S}^{d-1} \times \mathbb{R})},$$

*where the fractional power is understood via the Fourier transform.*

**Proposition D.7** (Equivalence of seminorms on bounded domains (Parhi and Nowak, 2023)). *Let $\mathcal{B} = \mathbb{B}_R^d$. For any $f : \mathcal{B} \to \mathbb{R}$ with finite variation seminorm, its canonical extension $f_{\mathrm{ext}}$ to $\mathbb{R}^d$ satisfies*

$$|f|_\mathrm{V} \;=\; \mathscr{R}\mathrm{TV}^2(f_{\mathrm{ext}}),$$

*and, in particular, for any finite two-layer ReLU network in reduced form $f_\theta(\boldsymbol{x}) = \sum_{k=1}^K v_k \, \phi(\boldsymbol{w}_k^\mathsf{T}\boldsymbol{x} - b_k) + \boldsymbol{c}^\mathsf{T}\boldsymbol{x} + c_0$,*

$$\mathscr{R}\mathrm{TV}^2(f_{\boldsymbol{\theta}}) \;=\; \sum_{k=1}^K |v_k| \, \|\boldsymbol{w}_k\|_2 \,,$$

*which equals the minimal (unweighted) path-norm needed to realize $f_\theta$ on $\mathbb{B}_R^d$.*

**Remark D.8.** *For ReLU networks on bounded domains, the three viewpoints:*

*path-norm $\|f\|_{\mathrm{path}}$ $\longleftrightarrow$ unweighted variation $|f|_\mathrm{V}$ $\longleftrightarrow$ second-order Radon-TV $\mathscr{R}\mathrm{TV}^2(f)$*

*are equivalent up to affine functions.*

## D.3 The Metric Entropy of Variation Spaces

Metric entropy quantifies the compactness of a set $A$ in a metric space $(X, \rho_X)$. Below we introduce the definition of covering numbers and metric entropy.

**Definition D.9** (Covering Number and Entropy). *Let $A$ be a compact subset of a metric space $(X, \rho_X)$. For $t > 0$, the covering number $N(A, t, \rho_X)$ is the minimum number of closed balls of radius $t$ needed to cover $A$:*

$$N(t, A, \rho_X) := \min \left\{ N \in \mathbb{N} : \exists \, x_1, \ldots, x_N \in X \text{ s.t. } A \subset \bigcup_{i=1}^N \mathbb{B}(x_i, t) \right\}, \tag{21}$$

*where $\mathbb{B}(x_i, t) = \{y \in X : \rho_X(y, x_i) \leq t\}$. The metric entropy of $A$ at scale $t$ is defined as:*

$$H_t(A)_X := \log N(t, A, \rho_X). \tag{22}$$

The metric entropy of the bounded variation function class has been studied in previous works. More specifically, we will directly use the one below in future analysis.

**Proposition D.10** (Parhi and Nowak 2023, Appendix D). *The metric entropy of* $\mathrm{V}_C(\mathbb{B}_R^d)$ *(see Definition D.2) with respect to the* $L^\infty(\mathbb{B}_R^d)$*-distance* $\| \cdot \|_\infty$ *satisfies*

$$\log N(t, \mathrm{V}_C(\mathbb{B}_R^d), \| \cdot \|_\infty) \lesssim_d \left( \frac{C}{t} \right)^{\frac{2d}{d+3}}. \tag{23}$$

*where* $\lesssim_d$ *hides constants (which could depend on d) and logarithmic factors.*

### D.4 GENERALIZATION GAP OF UNWEIGHTED VARIATION FUNCTION CLASS

As a middle step towards bounding the generalization gap of the weighted variation function class, we first bound the generalization gap of the unweighted variation function class according to a metric entropy analysis.

**Lemma D.11.** *Let* $\mathcal{F}_{M,C} = \{f \in \mathrm{V}_C(\mathbb{B}_R^d) \mid \|f\|_\infty \leq M\}$ *with* $M \geq D$*. Then let* $\mathcal{D} \sim \mathcal{P}^{\otimes n}$ *be a sampled data set of size* $n$*, with probability at least* $1 - \delta$*,*

$$\sup_{f \in \mathcal{F}_{M,C}} \left| R(f) - \widehat{R}_\mathcal{D}(f) \right| \lesssim_d C^{\frac{d}{2d+3}} M^{\frac{3(d+2)}{2d+3}} n^{-\frac{d+3}{4d+6}} + M^2 \left( \frac{\log(4/\delta)}{n} \right)^{-\frac{1}{2}}. \tag{24}$$

*Proof.* According to Proposition D.10, one just needs $N(t)$ balls to cover $\mathcal{F}$ in $\| \cdot \|_\infty$ with radius $t > 0$ such that where

$$\log N(t) \lesssim_d \left( \frac{C}{t} \right)^{\frac{2d}{d+3}}.$$

Then for any $f, g \in \mathcal{F}_{M,C}$ and any $(\boldsymbol{x}, y)$,

$$\left| (f(\boldsymbol{x}) - y)^2 - (g(\boldsymbol{x}) - y)^2 \right| = |f(\boldsymbol{x}) - g(\boldsymbol{x})| \, |f(\boldsymbol{x}) + g(\boldsymbol{x}) - 2y| \leq 4M \|f - g\|_\infty.$$

Hence replacing $f$ by a centre $f_i$ within $t$ changes both the empirical and true risks by at most $4Mt$.

For any fixed centre $\bar{f}$ in the covering, Hoeffding's inequality implies that with probability at least $\geq 1 - \delta$, we have

$$|R(\bar{f}) - \widehat{R}_\mathcal{D}(\bar{f})| \leq 4M^2 \sqrt{\frac{\log(2/\delta)}{n}} \tag{25}$$

because each squared error lies in $[0, 4M^2]$. Then we take all the centers with union bound to deduce that with probability at least $1 - \delta/2$, for any center $\bar{f}$ in the set of covering index, we have

$$\begin{aligned} |R(\bar{f}) - \widehat{R}_\mathcal{D}(\bar{f})| &\leq 4M^2 \sqrt{\frac{\log(4N(t)/\delta)}{n}} \\ &\lesssim M^2 \cdot \left( \frac{C}{t} \right)^{\frac{d}{d+3}} \left( \frac{1}{n} \right)^{-\frac{1}{2}} + M^2 \left( \frac{\log(4/\delta)}{n} \right)^{-\frac{1}{2}} \\ &\lesssim_d M^2 \cdot \left( \frac{C}{t} \right)^{\frac{d}{d+3}} \left( \frac{1}{n} \right)^{-\frac{1}{2}}, \end{aligned} \tag{26}$$

where $\lesssim_d$ hides the logarithmic factors about $1/\delta$ and constants.

According to the definition of covering sets, for any $f \in \mathcal{F}_{M,C}$, we have that $\|f - \bar{f}\|_\infty \leq t$ for some center $\bar{f}$. Then we have

$$\begin{aligned} &|R(f) - \widehat{R}_\mathcal{D}(f)| \\ &\lesssim_d |R(\bar{f}) - \widehat{R}_\mathcal{D}(\bar{f})| + O(Mt) \\ &\lesssim_d M^2 \cdot \left( \frac{C}{t} \right)^{\frac{d}{d+3}} n^{-\frac{1}{2}} + O(Mt). \end{aligned} \tag{27}$$

After tuning $t$ to be the optimal choice, we deduce that (24). $\qquad \square$

# E DATA-DEPENDENT REGULARITY FROM EDGE-OF-STABILITY

This section summarizes the *data-dependent regularity* induced by minima stability for two-layer ReLU networks.

## E.1 FUNCTION SPACE VIEWPOINT OF NEURAL NETWORKS BELOW THE EDGE OF STABILITY

Recall the notations: given a dataset $\mathcal{D} = \{(\boldsymbol{x}_i, y_i)\}_{i=1}^n \subset \mathbb{R}^d \times \mathbb{R}$, we define the data-dependent weight function $g_{\mathcal{D}} : \mathbb{S}^{d-1} \times \mathbb{R} \to \mathbb{R}$ by

$$g_{\mathcal{D}}(\boldsymbol{u}, t) := \min\{\tilde{g}_{\mathcal{D}}(\boldsymbol{u}, t), \tilde{g}_{\mathcal{D}}(-\boldsymbol{u}, -t)\},$$

where

$$\tilde{g}_{\mathcal{D}}(\boldsymbol{u}, t) := \mathbb{P}_{\mathcal{D}}(\boldsymbol{X}^{\mathsf{T}}\boldsymbol{u} > t)^2 \cdot \mathbb{E}_{\mathcal{D}}[\boldsymbol{X}^{\mathsf{T}}\boldsymbol{u} - t \mid \boldsymbol{X}^{\mathsf{T}}\boldsymbol{u} > t] \cdot \sqrt{1 + \|\mathbb{E}_{\mathcal{D}}[\boldsymbol{X} \mid \boldsymbol{X}^{\mathsf{T}}\boldsymbol{u} > t]\|^2}. \quad (28)$$

Here, $\boldsymbol{X}$ denotes a random draw uniformly sampled from $\{\boldsymbol{x}_i\}_{i=1}^n$, so that $\mathbb{P}_{\mathcal{D}}, \mathbb{E}_{\mathcal{D}}$ refer to probability and expectation under the empirical distribution $\frac{1}{n}\sum_{i=1}^n \delta_{\boldsymbol{x}_i}$. When the dataset $\mathcal{D}$ is fixed and clear from context, we will simply write $g$ in place of $g_{\mathcal{D}}$.

Then the curvature constrain on the loss landscape of $\mathcal{L}$ is converted into a weighted path norm constrain in the following sense.

**Proposition E.1** (Finite-sum version of Theorem 3.2 in (Liang et al., 2025)). *Suppose that $f_{\boldsymbol{\theta}}(\boldsymbol{x}) = \sum_{k=1}^K v_k \phi(\boldsymbol{w}_k^{\mathsf{T}}\boldsymbol{x} - b_k) + \beta$ is two-layer neural network such that the loss $\mathcal{L}$ is twice differentiable at $\boldsymbol{\theta}$. Then*

$$\sum_{k=1}^K |v_k| \|\boldsymbol{w}_k\| \cdot g\left(\frac{\boldsymbol{w}_k}{\|\boldsymbol{w}_k\|}, \frac{b_k}{\|\boldsymbol{w}_k\|}\right) \leq \frac{\lambda_{\max}(\nabla_{\boldsymbol{\theta}}^2 \mathcal{L}(\boldsymbol{\theta}))}{2} - \frac{1}{2} + (R+1)\sqrt{2\mathcal{L}(\boldsymbol{\theta})}. \quad (29)$$

*If we write $f_{\boldsymbol{\theta}}$ into a reduced form in (15), then we have*

$$\sum_{k=1}^{K'} a_k \cdot g\left(\boldsymbol{u}_k, t_k\right) \leq \frac{\lambda_{\max}(\nabla_{\boldsymbol{\theta}}^2 \mathcal{L}(\boldsymbol{\theta}))}{2} - \frac{1}{2} + (R+1)\sqrt{2\mathcal{L}(\boldsymbol{\theta})}. \quad (30)$$

Therefore, we bring up the definition the $g$-weighted path norm and variation norm are introduced as prior work introduced (Liang et al., 2025; Nacson et al., 2023).

**Definition E.2.** *Let $f_{\boldsymbol{\theta}}(\boldsymbol{x}) = \sum_{k=1}^K v_k \phi(\boldsymbol{w}_k^{\mathsf{T}}\boldsymbol{x} - b_k) + \beta$ be a two-layer neural network. The (g-)weighted path-norm of $f_{\boldsymbol{\theta}}$ is defined to be*

$$\|f_{\boldsymbol{\theta}}\|_{\mathrm{path},g} := \sum_{k=1}^K |v_k| \|\boldsymbol{w}_k\|_2 \cdot g\left(\frac{\boldsymbol{w}_k}{\|\boldsymbol{w}_k\|}, \frac{b_k}{\|\boldsymbol{w}_k\|}\right). \quad (31)$$

*Similarly, for functions of the form*

$$f_{\nu,\boldsymbol{c},c_0}(\boldsymbol{x}) = \int_{\mathbb{S}^{d-1} \times [-R,R]} \phi(\boldsymbol{u}^{\mathsf{T}}\boldsymbol{x} - t)\, \mathrm{d}\nu(\boldsymbol{u}, t) + \boldsymbol{c}^{\mathsf{T}}\boldsymbol{x} + c_0, \quad \boldsymbol{x} \in \mathbb{R}^d, \quad (32)$$

*where $R > 0$, $\boldsymbol{c} \in \mathbb{R}^d$, and $c_0 \in \mathbb{R}$, we define the $g$-weighted variation (semi)norm as*

$$|f|_{\mathrm{V}_g} := \inf_{\substack{\nu \in \mathcal{M}(\mathbb{S}^{d-1} \times [-R,R]) \\ \boldsymbol{c} \in \mathbb{R}^d, c_0 \in \mathbb{R}}} \|g \cdot \nu\|_{\mathcal{M}} \quad \text{s.t.} \quad f = f_{\nu,\boldsymbol{c},c_0}, \quad (33)$$

*where, if there does not exist a representation of $f$ in the form of (32), then the seminorm is understood to take the value $+\infty$. Here, $\mathcal{M}(\mathbb{S}^{d-1} \times [-R,R])$ denotes the Banach space of (Radon) measures and, for $\mu \in \mathcal{M}(\mathbb{S}^{d-1} \times [-R,R])$, $\|\mu\|_{\mathcal{M}} := \int_{\mathbb{S}^{d-1} \times [-R,R]} \mathrm{d}|\mu|(\boldsymbol{u}, t)$ is the measure-theoretic total-variation norm.*

*With this seminorm, we define the Banach space of functions $\mathrm{V}_g(\mathbb{B}_R^d)$ on the ball $\mathbb{B}_R^d := \{\boldsymbol{x} \in \mathbb{R}^d : \|\boldsymbol{x}\|_2 \leq R\}$ as the set of all functions $f$ such that $|f|_{\mathrm{V}_g}$ is finite. When $g \equiv 1$, $|\cdot|_{\mathrm{V}_g}$ and $\mathrm{V}_g(\mathbb{B}_R^d)$ coincide with the variation (semi)norm and variation norm space of Bach (2017).*

*For convenience, we introduce the notation of bounded weighted variation class*

$$\mathcal{F}_g(\Omega; M, C) := \left\{ f \colon \Omega \to \mathbb{R} \,\middle|\, |f|_{\mathrm{V}_g} \leq C, \, \|f|_\Omega\|_{L^\infty} \leq M \right\}. \tag{34}$$

*In particular, for any $\boldsymbol{\theta} \in \boldsymbol{\Theta}_g(\Omega; M, C)$, we have $f_{\boldsymbol{\theta}} \in \mathcal{F}_g(\Omega; M, C)$.*

**Remark E.3.** $\mathrm{V}_g$ *is a* weighted variation space *in the sense of DeVore et al. (2025).*

Within this framework together with the connection between $|\cdot|_{\mathrm{V}}$ and $\mathscr{R}\mathrm{TV}^2(\cdot)$ as summarized in Section D.2, we show the functional characterization of stable minima.

**Theorem E.4.** *For any $f_{\boldsymbol{\theta}} \in \boldsymbol{\Theta}_{\mathrm{BEoS}}(\eta, \mathcal{D})$, $|f_{\boldsymbol{\theta}}|_{\mathrm{V}_g} = \|g \cdot \mathscr{R}(-\Delta)^{\frac{d+1}{2}} f_{\boldsymbol{\theta}}\|_{\mathcal{M}} \leq \frac{1}{\eta} - \frac{1}{2} + (R + 1)\sqrt{2\mathcal{L}(\boldsymbol{\theta})}.$*

The detailed explanation and proof can be found in (Liang et al., 2025, Theorem 3.2, Corollary 3.3, Theorem 3.4, Appendix C, D).

### E.2 EMPIRICAL PROCESS FOR THE WEIGHT FUNCTION $g$

The implicit regularization of Edge-of-Stability induces a *data-dependent* regularity weight on the cylinder $\mathbb{S}^{d-1} \times \mathbb{R} := \mathbb{S}^{d-1} \times [-1, 1]$. Denote this empirical weight by $g_{\mathcal{D}}$ for a dataset $\mathcal{D} = \{x_i\}_{i=1}^n$. Directly analyzing generalization through the random, data-dependent class weighted by $g_{\mathcal{D}}$ is conceptually delicate, since the hypothesis class itself depends on the sample. To separate *statistical* from *algorithmic* randomness, we adopt the following paradigm.

(l) Fix an underlying distribution $\mathcal{P}$ for $X$ with only the support assumption $\mathrm{supp}(\mathcal{P}) \subseteq \mathbb{B}_R^d := \{x \in \mathbb{R}^d : \|x\| \leq R\}$. Define a *population* reference weight $g_{\mathcal{P}}$ on $\mathbb{S}^{d-1} \times \mathbb{R}$ (see below). This anchors a distribution-level notion of regularity independent of the particular sample.

(ii) For a realized dataset $\mathcal{D} \sim \mathcal{P}^{\otimes n}$, form the empirical plug-ins that define $g_{\mathcal{D}}$ on the same index set $\mathbb{S}^{d-1} \times \mathbb{R}$.

(iii) Use empirical-process theory to control the uniform deviation $\|g_{\mathcal{D}} - g_{\mathcal{P}}\|_\infty$ with high probability over the draw of $\mathcal{D}$. After this step, we can *condition on the high-probability event* and regard $\mathcal{D}$ as fixed in any subsequent analysis.

Let $X \sim \mathcal{P}$ with $\mathrm{supp}(\mathcal{P}) \subseteq \mathbb{B}_R^d$. For $(\boldsymbol{u}, t) \in \mathbb{S}^{d-1} \times \mathbb{R}$ define

$$p_{\mathcal{P}}(\boldsymbol{u}, t) := \mathcal{P}(X^\mathsf{T}\boldsymbol{u} > t), \qquad s_{\mathcal{P}}(\boldsymbol{u}, t) := \mathbb{E}_{X \sim \mathcal{P}}\big[(X^\mathsf{T}\boldsymbol{u} - t)_+\big].$$

On the unit ball we have $0 \leq (X^\mathsf{T}\boldsymbol{u} - t)_+ \leq 2$ and $\|\mathbb{E}_{\mathcal{P}}[X \mid X^\mathsf{T}\boldsymbol{u} > t]\| \leq 1$, which yields the *pointwise equivalence*

$$g_{\mathcal{P}}(\boldsymbol{u}, t) \asymp p_{\mathcal{P}}(\boldsymbol{u}, t)\, s_{\mathcal{P}}(\boldsymbol{u}, t) \quad \text{(with absolute constants).} \tag{35}$$

Given a dataset $\mathcal{D} = \{x_i\}_{i=1}^n$, let $\mathbb{P}_{\mathcal{D}}, \mathbb{E}_{\mathcal{D}}$ denote probability and expectation under the empirical distribution $\frac{1}{n}\sum_{i=1}^n \delta_{x_i}$. Define

$$p_{\mathcal{D}}(\boldsymbol{u}, t) := \mathbb{P}_{\mathcal{D}}(X^\mathsf{T}\boldsymbol{u} > t) = \frac{1}{n}\sum_{i=1}^n \mathbb{1}\{x_i^\mathsf{T}\boldsymbol{u} > t\}, \quad s_{\mathcal{D}}(\boldsymbol{u}, t) := \mathbb{E}_{\mathcal{D}}\big[(X^\mathsf{T}\boldsymbol{u} - t)_+\big] = \frac{1}{n}\sum_{i=1}^n (x_i^\mathsf{T}\boldsymbol{u} - t)_+,$$

and the empirical weight

$$g_{\mathcal{D}}(\boldsymbol{u}, t) \asymp p_{\mathcal{D}}(\boldsymbol{u}, t)\, s_{\mathcal{D}}(\boldsymbol{u}, t). \tag{36}$$

**Lemma E.5** (Uniform deviation for halfspaces)**.** *There exists a universal constant $C > 0$ such that, for every $\delta \in (0, 1)$,*

$$\mathbb{P}\left( \sup_{u \in \mathbb{S}^{d-1},\, t \in [-1,1]} \big| p_{\mathcal{D}}(\boldsymbol{u}, t) - p_{\mathcal{P}}(\boldsymbol{u}, t) \big| > C\sqrt{\frac{d + \log(1/\delta)}{n}} \right) \leq \delta.$$

*Proof.* The class $\{(\boldsymbol{x} \mapsto \mathbb{1}\{x^\mathsf{T}\boldsymbol{u} > t\}) : \boldsymbol{u} \in \mathbb{S}^{d-1},\, t \in \mathbb{R}\}$ has VC-dimension $d + 1$. Apply the VC-uniform convergence inequality for $\{0, 1\}$-valued classes (e.g., Vapnik (1998)) to the index set $\mathbb{S}^{d-1} \times [-1, 1]$ to obtain the stated bound. $\square$

**Lemma E.6** (Uniform deviation for ReLU). *There exists a universal constant $C > 0$ such that, for every $\delta \in (0, 1)$,*

$$\mathbb{P}\left(\sup_{u \in \mathbb{S}^{d-1},\, t \in [-1,1]} \left|s_{\mathcal{D}}(\boldsymbol{u}, t) - s_{\mathcal{P}}(\boldsymbol{u}, t)\right| > C\sqrt{\frac{d + \log(1/\delta)}{n}}\right) \leq \delta.$$

*Proof.* Let $\mathcal{F} := \{f_{\boldsymbol{u}, t}(\boldsymbol{x}) = (\boldsymbol{u}^{\mathsf{T}}\boldsymbol{x} - t)_+ : \boldsymbol{u} \in \mathbb{S}^{d-1}, t \in [-1, 1]\}$. Since $\|\boldsymbol{x}\| \leq 1$ and $t \in [-1, 1]$, every $f \in \mathcal{F}$ takes values in $[0, 2]$. Consider the subgraph class

$$\mathsf{subG}(\mathcal{F}) = \left\{ (\boldsymbol{x}, y) \in \mathbb{R}^d \times \mathbb{R} : y \leq (\boldsymbol{u}^{\mathsf{T}}\boldsymbol{x} - t)_+ \right\}.$$

For any $(\boldsymbol{x}, y)$ with $y \leq 0$, membership in $\mathsf{subG}(\mathcal{F})$ holds for all parameters, hence such points do not contribute to shattering. For points with $y > 0$, the condition $y \leq (\boldsymbol{u}^{\mathsf{T}}\boldsymbol{x} - t)_+$ is equivalent to $\boldsymbol{u}^{\mathsf{T}}\boldsymbol{x} - t - y \geq 0$, i.e., an affine halfspace in $\mathbb{R}^{d+1}$ with variables $(\boldsymbol{x}, y)$. Therefore the family $\mathsf{subG}(\mathcal{F})$ is (up to the immaterial fixed set $\{y \leq 0\}$) parametrized by affine halfspaces in $\mathbb{R}^{d+1}$, whose VC-dimension is at most $d + 2$. By the standard equivalence $\mathrm{Pdim}(\mathcal{F}) = \mathrm{VCdim}(\mathsf{subG}(\mathcal{F}))$, we obtain

$$\mathrm{Pdim}(\mathcal{F}) \leq d + 2.$$

Then by (Haussler, 1992, Theorem 3, Theorem 6, Theorem 7), we

$$\sup_{(\boldsymbol{u}, t)} \left|s_{\mathcal{D}}(\boldsymbol{u}, t) - s_{\mathcal{P}}(\boldsymbol{u}, t)\right| \leq C\sqrt{\frac{d + \log(1/\delta)}{n}}$$

with probability at least $1 - \delta$ for some universal constant $C$, which is the claimed bound. $\square$

**Theorem E.7** (Distribution-free uniform deviation for $\hat{g}_n$). *There exists a universal constant $C > 0$ such that, for every $\delta \in (0, 1)$,*

$$\mathbb{P}\left(\sup_{\boldsymbol{u} \in \mathbb{S}^{d-1},\, t \in [-1,1]} \left|g_{\mathcal{D}}(\boldsymbol{u}, t) - g_{\mathcal{P}}(\boldsymbol{u}, t)\right| > C\sqrt{\frac{d + \log(1/\delta)}{n}}\right) \leq 2\delta.$$

*Proof.* By (35) and (36), it suffices (up to a universal factor) to control $\left|p_{\mathcal{D}}s_{\mathcal{D}} - p_{\mathcal{P}}s_{\mathcal{P}}\right|$. Using $0 \leq s_{\mathcal{D}}, s_{\mathcal{P}} \leq 2$ and $0 \leq p_{\mathcal{D}}, p_{\mathcal{P}} \leq 1$,

$$\left|p_{\mathcal{D}}s_{\mathcal{D}} - p_{\mathcal{P}}s_{\mathcal{P}}\right| \leq |p_{\mathcal{D}} - p_{\mathcal{P}}|\, s_{\mathcal{P}} + |s_{\mathcal{D}} - s_{\mathcal{P}}|\, p_{\mathcal{P}} + |p_{\mathcal{D}} - p_{\mathcal{P}}|\, |s_{\mathcal{D}} - s_{\mathcal{P}}|$$

Taking the supremum over $(\boldsymbol{u}, t) \in \mathbb{S}^{d-1} \times [-1, 1]$ and applying Lemmas E.5 and E.6 with a union bound yields

$$\mathbb{P}\left(\sup_{\boldsymbol{u}, t} \left|p_{\mathcal{D}}s_{\mathcal{D}} - p_{\mathcal{P}}s_{\mathcal{P}}\right| \gtrsim \sqrt{\frac{d + \log(1/\delta)}{n}}\right) \leq 2\delta.$$

Recall that for $Q \in \{\mathcal{D}, \mathcal{P}\}$,

$$g_Q(\boldsymbol{u}, t) = \min\{\tilde{g}_Q(\boldsymbol{u}, t), \tilde{g}_Q(-\boldsymbol{u}, -t)\}.$$

For any real numbers $a, b, c, d$,

$$\left|\min\{a, b\} - \min\{c, d\}\right| \leq \max\{|a - c|, |b - d|\}.$$

Hence

$$\sup_{\boldsymbol{u} \in \mathbb{S}^{d-1},\, t \in [-1,1]} |g_{\mathcal{D}}(\boldsymbol{u}, t) - g_{\mathcal{P}}(\boldsymbol{u}, t)| \leq \sup_{\boldsymbol{u} \in \mathbb{S}^{d-1},\, t \in [-1,1]} |\tilde{g}_{\mathcal{D}}(\boldsymbol{u}, t) - \tilde{g}_{\mathcal{P}}(\boldsymbol{u}, t)|.$$

Since $(\boldsymbol{u}, t) \mapsto (-\boldsymbol{u}, -t)$ preserves $\mathbb{S}^{d-1} \times [-1, 1]$, the previously established uniform deviation bound for $\tilde{g}_{\mathcal{D}} - \tilde{g}_{\mathcal{P}}$ applies to both terms inside the minimum, and therefore transfers to $g_{\mathcal{D}} - g_{\mathcal{P}}$ with the same rate and failure probability. $\square$

## F  GENERALIZATION UPPER BOUND: MIXTURE OF LOW-DIMENSIONAL BALLS

In this section, we present the proof of Theorem 3.10. We formalize the mixture model by the following setting.

**Assumption F.1** (Mixture of low-dimensional balls). *Let $\{V_j\}_{j=1}^J$ be a finite collection of $J$ distinct $m$-dimensional (affine) linear subspaces within $\mathbb{R}^d$. Let $\mathcal{P}$ be a joint distribution over $\mathbb{R}^d \times \mathbb{R}$. The marginal distribution of the features $\boldsymbol{x}$ under $\mathcal{P}$, denoted $\mathcal{P}_{\boldsymbol{X}}$, is a mixture distribution given by*

$$\mathcal{P}_{\boldsymbol{X}}(\boldsymbol{x}) = \sum_{j=1}^J p_j \mathcal{P}_{\boldsymbol{X},j}(\boldsymbol{x}), \quad \mathcal{P}_{\boldsymbol{X},j}(\boldsymbol{x}) = \mathcal{P}_{\boldsymbol{X}}(\boldsymbol{x} \mid \boldsymbol{x} \in V_j), \tag{37}$$

*where $p_j > 0$ are the mixture probabilities $\mathbb{P}(\boldsymbol{x} \in V_j)$ satisfying $\sum_{j=1}^J p_j = 1$. Each component distribution $\mathcal{P}_j$ is the uniform distribution on the unit ball $\mathbb{B}_1^{V_j} := \{\boldsymbol{x} \in V_j : \|\boldsymbol{x}\|_2 \le 1\}$. The corresponding labels $y$ are generated from a conditional distribution $\mathcal{P}(y|\boldsymbol{x})$ and are assumed to be bounded, i.e., $|y| \le D$ for some constant $D > 0$. Similarly, we define $\mathcal{P}_j(\boldsymbol{x},y) = \mathcal{P}(\boldsymbol{x},y \mid \boldsymbol{x} \in V_j)$.*

First, we prove the simple case of singe-subspace assumption ($J = 1$) via Theorem F.3.

### F.1  CASE: UNIFORM DISTRIBUTION ON UNIT DISC OF A LINEAR SUBSPACE

Fix an $m$-dimensional subspace $V \subset \mathbb{R}^d$ and write $\mathbb{B}_1^V := \{\boldsymbol{x} \in V : \|\boldsymbol{x}\|_2 \le 1\}$, the canonical linear projection $\mathrm{proj}_V : \mathbb{R}^d \to V$. Recall the notations in (1): the parameters $\boldsymbol{\theta} := \{(v_k, \boldsymbol{w}_k, b_k)_{k=1}^K, \beta\}$ with $\boldsymbol{w}_k \ne \boldsymbol{0}$, define a two-layer neural network

$$f_{\boldsymbol{\theta}}(\boldsymbol{x}) = \sum_{k=1}^K v_k \phi(\boldsymbol{w}_k^\mathsf{T} \boldsymbol{x} - b_k) + \beta, \qquad \bar{\boldsymbol{w}}_k := \frac{\boldsymbol{w}_k}{\|\boldsymbol{w}_k\|_2}, \quad \bar{b}_k := \frac{b_k}{\|\boldsymbol{w}_k\|_2}.$$

Then we define neuronwise projection operator from neural networks to neural networks

$$\mathrm{proj}_V^* : \ f_{\boldsymbol{\theta}}(\boldsymbol{x}) \mapsto \sum_{k=1}^K v_k \phi\big((\mathrm{proj}_V \boldsymbol{w}_k)^\mathsf{T} \boldsymbol{x} - b_k\big) + \beta. \tag{38}$$

**Lemma F.2** (Projection reduction). *Fix $\mathcal{F}$ a hyothesis class of two-layer neural networks. Let $\mathcal{P}$ be a joint distribution on $(\boldsymbol{x}, y)$ supported on $\mathbb{R}^d \times [-D, D]$ such that the marginal distribution $\mathcal{P}_{\boldsymbol{X}}$ of $\boldsymbol{x}$ supports on $V$. For any dataset $\mathcal{D} := \{(\boldsymbol{x}_i, y_i)\}_{i=1}^n$ drawn i.i.d. from $\mathcal{P}$,*

$$\sup_{f \in \mathcal{F}} \mathrm{Gap}_{\mathcal{P}}(f; \mathcal{D}) = \sup_{f \in \mathcal{F}} \mathrm{Gap}_{\mathcal{P}}(\mathrm{proj}_V^* f; \mathcal{D}). \tag{39}$$

*Proof.* Because $\boldsymbol{x} \in V$ almost surely and in the sample, we have $f(\boldsymbol{x}) = (f \circ \mathrm{proj}_V)(\boldsymbol{x})$ for every $f$ and every $\boldsymbol{x} \in \mathbb{B}_1^V$. Using the identity $\boldsymbol{w}_k^\mathsf{T}(\mathrm{proj}_V \boldsymbol{x}) = (\mathrm{proj}_V \boldsymbol{w}_k)^\mathsf{T} \boldsymbol{x}$, we obtain $f \circ \mathrm{proj}_V = \mathrm{proj}_V^* f$ pointwise on $\mathbb{B}_1^V$. Hence for any $f \in \mathcal{F}$, $\mathrm{Gap}_{\mathcal{P}}(f; \mathcal{D}) = \mathrm{Gap}_{\mathcal{P}}(\mathrm{proj}_V^* f; \mathcal{D})$. $\square$

**Theorem F.3.** *Let $\mathcal{P}$ denote the joint distribution of $(\boldsymbol{x}, y)$. Assume that $\mathcal{P}$ is supported on $\mathbb{B}_1^d \times [-D, D]$ for some $D > 0$ and that the marginal distribution of $\boldsymbol{x}$ is $\mathrm{Uniform}(\mathbb{B}_1^V)$. Fix a dataset $\mathcal{D} = \{(\boldsymbol{x}_i, y_i)\}_{i=1}^n$, where each $(\boldsymbol{x}_i, y_i)$ is drawn i.i.d. from $\mathcal{P}$. Then, with probability $\ge 1 - \delta$,*

$$\sup_{f_{\boldsymbol{\theta}} \in \boldsymbol{\Theta}_{g_{\mathcal{D}}}(\mathbb{B}_1^{V_j}; M, C)} \mathrm{Gap}_{\mathcal{P}}(f_{\boldsymbol{\theta}}; \mathcal{D}) \lesssim_d C^{\frac{m}{m^2+4m+3}} M^2 n^{-\frac{1}{2m+4}} + M^2 \left(\frac{\log(4/\delta)}{n}\right)^{-\frac{1}{2}},$$

*where $M := \max\{D, \|f_{\boldsymbol{\theta}}\|_{L^\infty(\mathbb{B}_1^V)}, 1\}$ and $\lesssim_d$ hides constants (which could depend on $d$).*

*Proof.* By Lemma F.2, it remains to consider $\mathrm{proj}_V^* f_{\boldsymbol{\theta}}$. Let $\mathcal{D} \subset V$ and let $\boldsymbol{X}$ be supported on $V$. For any $\boldsymbol{w} \in \mathbb{R}^d$ with $\mathrm{proj}_V(\boldsymbol{w}) \ne 0$, set

$$\boldsymbol{u} := \frac{\boldsymbol{w}}{\|\boldsymbol{w}\|}, \qquad \boldsymbol{u}_V := \frac{\mathrm{proj}_V(\boldsymbol{w})}{\|\mathrm{proj}_V(\boldsymbol{w})\|}.$$

Since for all $\boldsymbol{x} \in V$,
$$\boldsymbol{w}^\top \boldsymbol{x} = \mathrm{proj}_V(\boldsymbol{w})^\top \boldsymbol{x},$$

we have
$$\boldsymbol{w}^\top \boldsymbol{x} > b \quad \Longleftrightarrow \quad \mathrm{proj}_V(\boldsymbol{w})^\top \boldsymbol{x} > b.$$

After normalization,
$$\boldsymbol{u}^\top \boldsymbol{x} > \frac{b}{\|\boldsymbol{w}\|} \quad \Longleftrightarrow \quad \boldsymbol{u}_V^\top \boldsymbol{x} > \frac{b}{\|\mathrm{proj}_V(\boldsymbol{w})\|}.$$

Hence
$$g\left(\frac{\boldsymbol{w}}{\|\boldsymbol{w}\|}, \frac{b}{\|\boldsymbol{w}\|}\right) = g\left(\frac{\mathrm{proj}_V(\boldsymbol{w})}{\|\mathrm{proj}_V(\boldsymbol{w})\|}, \frac{b}{\|\mathrm{proj}_V(\boldsymbol{w})\|}\right).$$

Therefore, when the data are supported on $V$, the weight function depends only on directions in $V$. It suffices to consider the class
$$\boldsymbol{\Theta}_{g_\mathcal{D}}^V(\mathbb{B}_1^{V_j}; M, C) = \left\{\mathrm{proj}_V^* f_{\boldsymbol{\theta}} : f_{\boldsymbol{\theta}} \in \boldsymbol{\Theta}_{g_\mathcal{D}}(\mathbb{B}_1^{V_j}; M, C)\right\}.$$

Therefore, we just need consider the case where the whole algorithm with any dataset sample from $V$ operates in $V$ and we get the result from (Liang et al., 2025, Theorem F.8) by replacing $\mathbb{R}^d$ with $V \cong \mathbb{R}^m$. $\qquad\square$

## F.2    PROOF OF THEOREM 3.10

In this section, we extend the generalization analysis from a single low-dimensional subspace to a more complex and practical scenario where the data is supported on a finite union of such subspaces. This setting is crucial for modeling multi-modal data, where distinct clusters can each be approximated by a low-dimensional linear structure. Our main result demonstrates that the sample complexity of stable minima adapts to the low intrinsic dimension of the individual subspaces, rather than the high ambient dimension of the data space.

### F.2.1    ANALYSIS OF THE GLOBAL WEIGHT FUNCTION

A critical step in our proof is to understand the relationship between the global weight function $g(\boldsymbol{u}, t)$, which is induced by the mixture distribution $\mathcal{P}$, and the local weight functions $g_j(\boldsymbol{u}, t)$, each induced by a single component distribution $\mathcal{P}_j$ defined on $V_j$, which should be understood as the distribution conditioned to $\boldsymbol{x} \in V_j$. Fix a dataset $\mathcal{D}$, the function class $\boldsymbol{\Theta}_{\mathrm{BEoS}}(\eta; \mathcal{D})$ is defined by the properties of the global function $g$. To analyze the performance on a specific subspace $V_j$, we must ensure that the global regularity constraint is sufficiently strong when viewed locally. The following lemma provides this crucial guarantee.

**Lemma F.4** (Global-to-Local Weight Domination)**.** *For any mixed distribution* $\mathcal{P}_X = \sum_{j=1}^J p_j \mathcal{P}_{X,j}$ *with* $\mathrm{supp}(\mathcal{P}_{X,j}) = V_j$. *Let* $g$ *be the global weight induced by the mixture* $\mathcal{P}_X$, *and* $g_j$ *the weight induced by* $\mathcal{P}_{X,j}$. *For every* $j \in \{1, \dots, J\}$,

$$g(\boldsymbol{u}, t) \geq \frac{p_j^2}{\sqrt{2}} g_j(\boldsymbol{u}, t), \quad \text{for all } (\boldsymbol{u}, t) \in \mathbb{S}^{d-1} \times \mathbb{R}. \tag{40}$$

*Consequently, for any* $M, C > 0$,

$$\mathcal{F}_g(\mathbb{B}_1^{V_j}; M, C) \subseteq \mathcal{F}_{g_j}\left(\mathbb{B}_1^{V_j}; M, \sqrt{2}\,C/2p_j^2\right). \tag{41}$$

*Proof.* Fix $j$ and the activation event $A := \{\boldsymbol{x} : \boldsymbol{u}^\top \boldsymbol{x} > t\}$. By definition of $g$ (global) and $g_j$ (local) we can write

$$g(\boldsymbol{u}, t) = \mathcal{P}_X(A)^2 \cdot \mathop{\mathbb{E}}_{\boldsymbol{x} \sim \mathcal{P}_X}[\boldsymbol{X}^\top \boldsymbol{u} - t \mid A] \cdot \sqrt{1 + \|\mathop{\mathbb{E}}_{\boldsymbol{x} \sim \mathcal{P}_X}[\boldsymbol{X} \mid A]\|_2^2}$$

$$g_j(\boldsymbol{u}, t) = \mathcal{P}_X(A \mid \boldsymbol{x} \in V_j)^2 \cdot \mathop{\mathbb{E}}_{\boldsymbol{x} \sim \mathcal{P}_X}[\boldsymbol{X}^\top \boldsymbol{u} - t \mid A, \boldsymbol{x} \in V_j] \cdot \sqrt{1 + \|\mathop{\mathbb{E}}_{\boldsymbol{x} \sim \mathcal{P}_X}[\boldsymbol{X} \mid A, \boldsymbol{x} \in V_j]\|_2^2}$$

$$= \mathcal{P}_{X,j}(A)^2 \cdot \mathop{\mathbb{E}}_{\boldsymbol{x} \sim \mathcal{P}_{X,j}}[\boldsymbol{X}^\top \boldsymbol{u} - t \mid A] \cdot \sqrt{1 + \|\mathop{\mathbb{E}}_{\boldsymbol{x} \sim \mathcal{P}_{X,j}}[\boldsymbol{X} \mid A]\|_2^2}$$

Using the law of total probability and total expectation for the mixture distribution $\mathcal{P}_X = \sum_{i=1}^{J} p_i \mathcal{P}_{\boldsymbol{X},i}$, and the non-negativity of $(\boldsymbol{X}^\mathsf{T}\boldsymbol{u} - t)\mathbb{1}_A$, we get

$$\mathcal{P}_X(A) \geq p_j \, \mathcal{P}_{\boldsymbol{X},j}(A), \qquad \mathbb{E}_{\boldsymbol{x}\sim\mathcal{P}_X}[(\boldsymbol{X}^\mathsf{T}\boldsymbol{u} - t)\mathbb{1}_A] \geq p_j \, \mathbb{E}_{\boldsymbol{x}\sim\mathcal{P}_{\boldsymbol{X},j}}[(\boldsymbol{X}^\mathsf{T}\boldsymbol{u} - t)\mathbb{1}_A].$$

Hence, by combining the first two terms of $g(\boldsymbol{u}, t)$ as $\mathcal{P}_X(A)\, \mathbb{E}_{\boldsymbol{x}\sim\mathcal{P}_X}[(\boldsymbol{X}^\mathsf{T}\boldsymbol{u} - t)\mathbb{1}_A]$, we have:

$$g(\boldsymbol{u}, t) \geq \big(p_j \mathcal{P}_{\boldsymbol{X},j}(A)\big) \cdot \big(p_j \, \mathbb{E}_{\boldsymbol{x}\sim\mathcal{P}_{\boldsymbol{X},j}}[(\boldsymbol{X}^\mathsf{T}\boldsymbol{u} - t)\mathbb{1}_A]\big) \cdot 1 = p_j^2\, \mathcal{P}_{\boldsymbol{X},j}(A) \, \mathbb{E}_{\boldsymbol{x}\sim\mathcal{P}_{\boldsymbol{X},j}}[(\boldsymbol{X}^\mathsf{T}\boldsymbol{u} - t)\mathbb{1}_A].$$

For the local weight function $g_j$, the same algebra gives

$$g_j(\boldsymbol{u}, t) = \mathcal{P}_{\boldsymbol{X},j}(A) \, \mathbb{E}_{\boldsymbol{x}\sim\mathcal{P}_{\boldsymbol{X},j}}[(\boldsymbol{X}^\mathsf{T}\boldsymbol{u} - t)\mathbb{1}_A] \cdot \sqrt{1 + \|\, \mathbb{E}_{\boldsymbol{x}\sim\mathcal{P}_{\boldsymbol{X},j}}[\boldsymbol{X} \mid A]\|_2^2}\,.$$

Since the support of $\mathcal{P}_{\boldsymbol{X},j}$ is $\mathbb{B}_1^{V_j}$, we have $\|\boldsymbol{X}\|_2 \leq 1$ almost surely under $\mathcal{P}_{\boldsymbol{X},j}$. This implies $\|\, \mathbb{E}_{\boldsymbol{x}\sim\mathcal{P}_{\boldsymbol{X},j}}[\boldsymbol{X} \mid A]\|_2 \leq 1$, and therefore $\sqrt{1 + \|\, \mathbb{E}_{\boldsymbol{x}\sim\mathcal{P}_{\boldsymbol{X},j}}[\boldsymbol{X} \mid A]\|_2^2} \leq \sqrt{2}$.

Combining these results, we establish the lower bound:

$$g(\boldsymbol{u}, t) \geq \frac{p_j^2}{\sqrt{2}}\left(\mathcal{P}_{\boldsymbol{X},j}(A) \, \mathbb{E}_{\boldsymbol{x}\sim\mathcal{P}_{\boldsymbol{X},j}}[(\boldsymbol{X}^\mathsf{T}\boldsymbol{u} - t)\mathbb{1}_A] \cdot \sqrt{1 + \|\, \mathbb{E}_{\boldsymbol{x}\sim\mathcal{P}_{\boldsymbol{X},j}}[\boldsymbol{X} \mid A]\|_2^2}\right) = \frac{p_j^2}{\sqrt{2}} \, g_j(\boldsymbol{u}, t),$$

which proves (40). The class embedding (41) follows directly from the definition of the weighted variation seminorm. $\qquad\square$

**Proposition F.5.** *Let $\mathcal{P}$ be a distribution defined in Assumption F.1 and recall that $\mathcal{P}_j$ is $\mathcal{P}$ conditional to $\boldsymbol{x} \in V_j$. Fix $j \in \{1, \dots, J\}$ and a data set $\mathcal{D} \sim \mathcal{P}^{\otimes n}$. Let $\mathcal{D}_j := \mathcal{D} \cap V_j$ and $n_j := |\mathcal{D}_j|$. Then with probability $1 - \delta$,*

$$\sup_{f_{\boldsymbol{\theta}}\in\boldsymbol{\Theta}_{\mathrm{BEoS}}(\eta,\mathcal{D})} \mathrm{Gap}_{\mathcal{P}_j}(f_{\boldsymbol{\theta}}; \mathcal{D}_j) \lesssim_d \left(\frac{\frac{1}{\eta} - \frac{1}{2} + 4M}{p_j^2}\right)^{\frac{m}{m^2+4m+3}} M^2 \, n_j^{-\frac{1}{2m+4}} + M^2 \left(\frac{\log(4/\delta)}{n}\right)^{-\frac{1}{2}}. \tag{42}$$

*where $M := \max\{D, \|f_{\boldsymbol{\theta}}\|_{L^\infty(\mathbb{B}_1^{V_j})}, 1\}$ and $\lesssim_d$ hides constants (which could depend on $d$).*

*Proof.* Note that the notation $\mathrm{Gap}_{\mathcal{P}_j}(f_{\boldsymbol{\theta}}; \mathcal{D}_j)$ can be expanded into

$$\mathrm{Gap}_{\mathcal{P}_j}(f_{\boldsymbol{\theta}}; \mathcal{D}_j) = \left| R_{\mathcal{P}_j}(f_{\boldsymbol{\theta}}) - \widehat{R}_{\mathcal{D}_j}(f_{\boldsymbol{\theta}}) \right|$$

$$= \left| \mathbb{E}_{(\boldsymbol{x},y)\sim\mathcal{P}_j}\left[(f_{\boldsymbol{\theta}}(\boldsymbol{x}) - y)^2\right] - \widehat{R}_{\mathcal{D}_j}(f_{\boldsymbol{\theta}}) \right|$$

$$= \left| \mathbb{E}_{(\boldsymbol{x},y)\sim\mathcal{P}}\left[(f_{\boldsymbol{\theta}}(\boldsymbol{x}) - y)^2 \mid \boldsymbol{x} \in V_j\right] - \widehat{R}_{\mathcal{D}_j}(f_{\boldsymbol{\theta}}) \right|$$

Let $C = \frac{1}{\eta} - \frac{1}{2} + 4M$. According to (Liang et al., 2025, Corollary 3.3), we have that

$$f_{\boldsymbol{\theta}} \in \boldsymbol{\Theta}_{g_{\mathcal{D}}}(\mathbb{B}_1^{V_j}; M, C), \quad \forall \boldsymbol{\theta} \in \boldsymbol{\Theta}_{\mathrm{BEoS}}(\eta; \mathcal{D}).$$

Then by Lemma F.4, we conclude that

$$\boldsymbol{\Theta}_g(\mathbb{B}_1^{V_j}; M, C) \subseteq \boldsymbol{\Theta}_{g_j}(\mathbb{B}_1^{V_j}; M, \sqrt{2}C/p_j^2),$$

where the weight functions $g$ and $g_j$ can be either empirical or population.

Therefore,

$$\sup_{\boldsymbol{\theta}\in\boldsymbol{\Theta}_{\mathrm{BEoS}}(\eta;\mathcal{D})} \mathrm{Gap}_{\mathcal{P}_j}(f_{\boldsymbol{\theta}}; \mathcal{D}_j) \leq \sup_{f\in\boldsymbol{\Theta}_{g_j}(\mathbb{B}_1^{V_j};M,\sqrt{2}C/p_j^2)} \mathrm{Gap}_{\mathcal{P}_j}(f; \mathcal{D}_j)$$

Then by Theorem F.3, we may conclude that

$$\sup_{f_{\boldsymbol{\theta}} \in \mathcal{F}_{g_j}(\mathbb{B}_1^{V_j}; M, \sqrt{2}C/p_j^2)} \mathrm{Gap}_{\mathcal{P}_j}(f_{\boldsymbol{\theta}}; \mathcal{D}_j) \lesssim_d \left( \frac{\frac{1}{\eta} - \frac{1}{2} + 4M}{p_j^2} \right)^{\frac{m}{m^2+4m+3}} M^2 \, n_j^{-\frac{1}{2m+4}}$$

$\square$

**Theorem F.6** (Generalization Bound for Mixture Models). *Let the data distribution $\mathcal{P}$ be as defined in Assumption 1. Let $\mathcal{D} = \{(\boldsymbol{x}_i, y_i)\}_{i=1}^n$ be a dataset of $n$ i.i.d. samples drawn from $\mathcal{P}$. Then, with probability at least $1 - 2\delta$,*

$$\sup_{\boldsymbol{\theta} \in \Theta_{\mathrm{BEoS}}(\eta, \mathcal{D})} \mathrm{Gap}_{\mathcal{P}}(f_{\boldsymbol{\theta}}; \mathcal{D}) \lesssim_d \left( \frac{1}{\eta} - \frac{1}{2} + 4M \right)^{\frac{m}{m^2+4m+3}} M^2 \, J^{\frac{4}{m}} \, n^{-\frac{1}{2m+4}} + M^2 \, J \sqrt{\frac{\log(4J/\delta)}{2n}}. \tag{43}$$

*where $M := \max\{D, \|f_{\boldsymbol{\theta}}\|_{\mathbb{B}_1^V}\|_{L^\infty}, 1\}$ and $\lesssim_d$ hides constants (which could depend on $d$).*

The proof proceeds in several steps. First, we establish a high-probability event where the number of samples drawn from each subspace is close to its expected value. Second, we decompose the total generalization gap into several terms. Finally, we bound each of these terms, showing that the dominant term is determined by the generalization performance on the individual subspaces, which scales with the intrinsic dimension $m$.

*Proof.* Let $n_j = \sum_{i=1}^n \mathbb{1}_{\{x_i \in V_j\}}$ be the number of samples from the dataset $\mathcal{D}$ that fall into the subspace $V_j$. Each $n_j$ is a random variable following a Binomial distribution, $n_j \sim \mathrm{Bin}(n, p_j)$. We need to ensure that for all subspaces simultaneously, the empirical proportion $n_j/n$ is close to the true probability $p_j$.

We use Hoeffding's inequality for each $j \in \{1, \ldots, J\}$. For any $\epsilon > 0$, $\mathbb{P}\left(\left|\frac{n_j}{n} - p_j\right| \geq \epsilon\right) \leq 2e^{-2n\epsilon^2}$. To ensure this holds for all $J$ subspaces at once, we apply a union bound. Let $\delta_j$ be the failure probability allocated to the $j$-th subspace. The total failure probability is at most $\sum_{j=1}^J \delta_j = \delta$, so we set $\delta_j = \delta/J$ and yields $\epsilon = \sqrt{\frac{\log(2J/\delta)}{2n}}$.

Let $\mathcal{E}$ be the event that $\left|\frac{n_j}{n} - p_j\right| \leq \epsilon$ holds for all $j = 1, \ldots, J$. We have shown that $\mathbb{P}(\mathcal{E}) \geq 1 - \delta$. The remainder of our proof is conditioned on this event $\mathcal{E}$. A direct consequence of this event is a lower bound on each $n_j$

$$n_j \geq np_j - n\epsilon = np_j - \sqrt{\frac{n}{2} \log \frac{2J}{\delta}}. \tag{44}$$

Now we decompose the generalization gap using the law of total expectation for the true risk and by partitioning the empirical sum for the empirical risk.

Let $\mathcal{P}_j$ denote the distribution $\mathcal{P}$ conditioned on $x \in V_j$, and let $\mathcal{D}_j = \mathcal{D} \cap V_j\}$.

$$
\begin{aligned}
\mathrm{Gap}_{\mathcal{P}}(f_{\boldsymbol{\theta}}; \mathcal{D}) &= \left| R(f_{\boldsymbol{\theta}}) - \widehat{R}_{\mathcal{D}}(f_{\boldsymbol{\theta}}) \right| \\
&= \left| \sum_{j=1}^{J} p_j \mathop{\mathbb{E}}_{\boldsymbol{x} \sim \mathcal{P}} [(f(\boldsymbol{x}) - y)^2 \mid \boldsymbol{x} \in V_j] - \sum_{j=1}^{J} \frac{n_j}{n} \frac{1}{n_j} \sum_{(\boldsymbol{x}_i, y_i) \in \mathcal{D}_j} (f(\boldsymbol{x}_i) - y_i)^2 \right| \\
&\le \left| \sum_{j=1}^{J} p_j \mathop{\mathbb{E}}_{\boldsymbol{x} \sim \mathcal{P}_j} [(f_{\boldsymbol{\theta}}(\boldsymbol{x}) - y)^2] - \sum_{j=1}^{J} p_j \frac{1}{n_j} \sum_{(\boldsymbol{x}_i, y_i) \in \mathcal{D}_j} (f_{\boldsymbol{\theta}}(\boldsymbol{x}_i) - y_i)^2 \right| \\
&\quad + \left| \sum_{j=1}^{J} p_j \frac{1}{n_j} \sum_{(x_i, y_i) \in \mathcal{D}_j} (f_{\boldsymbol{\theta}}(\boldsymbol{x}_i) - y_i)^2 - \sum_{j=1}^{J} \frac{n_j}{n} \frac{1}{n_j} \sum_{(\boldsymbol{x}_i, y_i) \in \mathcal{D}_j} (f_{\boldsymbol{\theta}}(\boldsymbol{x}_i) - y_i)^2 \right| \\
&\le \sum_{j=1}^{J} p_j \left| R_{\mathcal{P}_j}(f_{\boldsymbol{\theta}}) - \widehat{R}_{\mathcal{D}_j}(f_{\boldsymbol{\theta}}) \right| + \sum_{j=1}^{J} \left| p_j - \frac{n_j}{n} \right| \widehat{R}_{\mathcal{D}_j}(f_{\boldsymbol{\theta}}) \\
&= \underbrace{\sum_{j=1}^{J} p_j \mathrm{Gap}_{\mathcal{P}_j}(f_{\boldsymbol{\theta}}; \mathcal{D}_j)}_{\text{Term A}} + \underbrace{\sum_{j=1}^{J} \left| p_j - \frac{n_j}{n} \right| \widehat{R}_{\mathcal{D}_j}(f_{\boldsymbol{\theta}})}_{\text{Term B}}
\end{aligned}
$$

where $\widehat{R}_{\mathcal{D}_j}(f) = \frac{1}{n_j} \sum_{(\boldsymbol{x}_i, y_i) \in \mathcal{D}_j} (f(\boldsymbol{x}_i) - y_i)^2$.

- **Bounding the Weighted Sum of Conditional Gaps (Term A):** According to Proposition F.5, with probability at least $1 - \delta$, for each $j$,

$$
\mathrm{Gap}_{\mathcal{P}_j}(f_{\boldsymbol{\theta}}; \mathcal{D}_j) \lesssim_d \left( \frac{\frac{1}{\eta} - \frac{1}{2} + 4M}{p_j^2} \right)^{\frac{m}{m^2 + 4m + 3}} M^2 n_j^{-\frac{1}{2m+4}} + M^2 \left( \frac{\log(4J/\delta)}{n} \right)^{-\frac{1}{2}}.
$$

Conditioned on $\mathcal{E}$, we use the lower bound on $n_j$ from (44) , $n_j \le n p_j (1 - \epsilon/p_j)$.

$$
\begin{aligned}
\text{Term A} &= \sum_{j=1}^{J} p_j \mathrm{Gap}_{\mathcal{P}_j}(f_{\boldsymbol{\theta}}; \mathcal{D}_j) \\
&\lesssim_d \sum_{j=1}^{J} p_j \left( \frac{\frac{1}{\eta} - \frac{1}{2} + 4M}{p_j^2} \right)^{\frac{m}{m^2 + 4m + 3}} M^2 (n p_j (1 - \epsilon/p_j))^{-\frac{1}{2m+4}} \\
&= \left( \frac{1}{\eta} - \frac{1}{2} + 4M \right)^{\frac{m}{m^2 + 4m + 3}} M^2 n^{-\frac{1}{2m+4}} \sum_{j=1}^{J} p_j \cdot (p_j^{-2})^{\frac{m}{m^2 + 4m + 3}} \cdot \left( p_j - \sqrt{\frac{\log(2J/\delta)}{2n}} \right)^{-\frac{1}{2m+4}} \\
&\lesssim_d \left( \frac{1}{\eta} - \frac{1}{2} + 4M \right)^{\frac{m}{m^2 + 4m + 3}} M^2 n^{-\frac{1}{2m+4}} \sum_{j=1}^{J} p_j^{1 - \frac{2m}{m^2 + 4m + 3} - \frac{1}{2m+4}}.
\end{aligned}
$$

The exponent of $p_j$ simplifies to

$$
1 - \frac{2m}{(m+1)(m+3)} - \frac{1}{2m+4} = \frac{2m^3 + 7m^2 + 10m + 9}{2(m+1)(m+2)(m+3)}. \tag{45}
$$

For positive integers $m$, (45) is strictly increasing and bounded above by 1. In particular, when $m = 1$, (45) $= \frac{7}{12}$. Therefore, a brute-force upper bound is

$$
\sum_{j=1}^{J} p_j^{\frac{2m^3 + 7m^2 + 10m + 9}{2(m+1)(m+2)(m+3)}} \le J
$$

and thus

$$\text{Term A} \lesssim_d \left(\frac{1}{\eta} - \frac{1}{2} + 4M\right)^{\frac{m}{m^2+4m+3}} M^2 n^{-\frac{1}{2m+4}} \sum_{j=1}^{J} p_j^{1-\frac{2m}{m^2+4m+3}-\frac{1}{2m+4}}$$

$$\lesssim_d \left(\frac{1}{\eta} - \frac{1}{2} + 4M\right)^{\frac{m}{m^2+4m+3}} M^2 \, J \, n^{-\frac{1}{2m+4}}.$$

Note that the dependence of Term A on $J$ is very mild. Indeed, if we denote

$$\alpha(m) = 1 - \frac{2m}{m^2+4m+3} - \frac{1}{2m+4},$$

then

$$\sum_{j=1}^{J} p_j^{\alpha(m)} \leq J^{1-\alpha(m)} \leq J^{\frac{2m}{m^2+4m+3}+\frac{1}{2m+4}} \leq J^{\frac{4}{m}},$$

since $\sum_j p_j = 1$. For large $m$, the exponent $\alpha(m)$ is close to 1, hence $\sum_j p_j^{\alpha(m)}$ remains essentially of order one. Consequently, the bound on Term A grows at most linearly with $J$, and in practice the $J$-dependence is negligible in high $m$. Here we use the power $4/m$ upper for clean format.

- **Bounding the Sampling Deviation Error (Term B):** Conditioned on the event $\mathcal{E}$, we have $|p_j - n_j/n| \leq \epsilon$ for all $j$. The empirical risk term is bounded because $\max\{|f(\boldsymbol{x})|, |y|\} \leq M$, which implies $|\frac{1}{n_j} \sum_{(\boldsymbol{x}_i, y_i) \in \mathcal{D}_j} (f_{\boldsymbol{\theta}}(\boldsymbol{x}_i) - y_i)^2| \leq 4M^2$. Thus, Term B is bounded by:

$$\text{Term B} \leq \sum_{j=1}^{J} \epsilon 4M^2 = 4M^2\epsilon = 4JM^2 \sqrt{\frac{\log(4J/\delta)}{2n}}. \tag{46}$$

The total generalization gap is bounded by the sum of the bounds for Term A and Term B.

$$\text{Gap}_{\mathcal{P}}(f_{\boldsymbol{\theta}}; \mathcal{D}) \lesssim_d \left(\frac{1}{\eta} - \frac{1}{2} + 4M\right)^{\frac{m}{m^2+4m+3}} M^2 \, J^{\frac{4}{m}} \, n^{-\frac{1}{2m+4}} + M^2 \, J \sqrt{\frac{\log(4J/\delta)}{2n}}.$$

This completes the proof. □

## G  GENERALIZATION UPPER BOUNDS: ISOTROPIC BETA FAMILY

In this section, the data generalization process is considered to be a family of isotropic Beta-radial distributions.

**Definition G.1** (Isotropic Beta-radial distributions). *Let $\boldsymbol{X}$ be a $d$-dimensional random vector in $\mathbb{R}^d$. For any $\alpha \in (0, \infty)$, the isotropic $\text{Beta}(\alpha)$-radial distribution is defined by the generation process*

$$\boldsymbol{X} = h(R)\boldsymbol{U} \sim \mathcal{P}_X(\alpha), \tag{47}$$

*where $R \sim \text{Uniform}[0, 1]$ is a random variable drawn from a continuous uniform distribution on the interval $[0, 1]$, $\boldsymbol{U} \sim \text{Uniform}(\mathbb{S}^{d-1})$ is a random vector drawn uniformly from the unit sphere $\mathbb{S}^{d-1}$ in $\mathbb{R}^d$ and $h(r) = 1 - (1-r)^{1/\alpha}$ is a radial profile.*

**Lemma G.2.** *Let $\mathcal{P}_X(\alpha)$ be the isotropic $\text{Beta}(\alpha)$-radial distribution in Definition 3.2. For $\boldsymbol{X} \sim \mathcal{P}_X(\alpha)$ any $t \in [0, 1]$, $\mathbb{P}(\|\boldsymbol{X}\| > 1 - t) = t^\alpha$. In particular, $\|\boldsymbol{X}\|_2$ is a $\text{Beta}(1, \alpha)$ distribution.*

*Proof.* The proof follows from a direct calculation based on the properties of the data-generating process.

First, the norm simplifies to: $\|\boldsymbol{X}\| = \|h(R)\boldsymbol{U}\| = h(R)$. Next, it is equivalent to calculating the probability that the scalar random variable $h(R)$ is greater than $1 - t$:

$$\mathbb{P}(\|\boldsymbol{X}\| > 1 - t) = \mathbb{P}(h(R) > 1 - t).$$

To proceed, we need to apply the inverse of the function $h$ to both sides of the inequality. The function $h(r)$ is monotonically increasing for $r \in [0, 1]$, so applying its inverse preserves the direction of the inequality. Note that the inverse function is $h^{-1}(y) = 1 - (1 - y)^\alpha$.

Applying the inverse function $h^{-1}$ to the inequality $h(R) > 1 - t$, we get

$$R > h^{-1}(1 - t).$$

Substituting the expression for $h^{-1}$:

$$R > 1 - (1 - (1 - t))^\alpha = 1 - t^\alpha.$$

Finally, we compute the probability of this event for the random variable $R$. By our initial assumption, $R$ is uniformly distributed on the interval $[0, 1]$, i.e., $R \sim \mathrm{Uniform}[0, 1]$. The cumulative distribution function (CDF) of $R$ is $F_R(x) = x$ for $x \in [0, 1]$. The tail probability is therefore,

$$\mathbb{P}(R > x) = 1 - F_R(x) = 1 - x.$$

Applying this to our inequality $R > 1 - t^\alpha$:

$$\mathbb{P}\left(R > 1 - t^\alpha\right) = 1 - (1 - t^\alpha) = t^\alpha.$$

Combining all steps, we have rigorously shown that

$$\mathbb{P}\left(\|\boldsymbol{X}\| > 1 - t\right) = t^\alpha.$$

To show that this implies $\|\boldsymbol{X}\|$ is a $\mathrm{Beta}(1, \alpha)$ distribution, we can examine its cumulative distribution function (CDF). Let $Y = \|\boldsymbol{X}\|$. The CDF is $F_Y(y) = \mathbb{P}(Y \leq y)$. Substituting $y = 1 - t$, we have $t = 1 - y$. Then the tail probability becomes:

$$\mathbb{P}(\|\boldsymbol{X}\| > y) = (1 - y)^\alpha.$$

From this, the CDF can be derived as

$$F_Y(y) = \mathbb{P}(\|\boldsymbol{X}\| \leq y) = 1 - \mathbb{P}(\|\boldsymbol{X}\| > y) = 1 - (1 - y)^\alpha.$$

This is the characteristic CDF of a $\mathrm{Beta}(1, \alpha)$ distribution, thus completing the proof. $\qquad\square$

**Assumption G.3.** *Fix $\alpha \in (0, \infty)$. Let $\mathcal{P}(\alpha)$ be a joint distribution over $\mathbb{R}^d \times \mathbb{R}$ such that the marginal distribution of the features $x$ is $\mathcal{P}_{\boldsymbol{X}}(\alpha)$. The corresponding labels $y$ are generated from a conditional distribution $\mathcal{P}(y|\boldsymbol{x})$ and are assumed to be bounded, i.e., $|y| \leq D$ for some constant $D > 0$. Similarly, we define $\mathcal{P}_j(\boldsymbol{x}, y) = \mathcal{P}(\boldsymbol{x}, y \mid \boldsymbol{x} \in V_j)$.*

### G.1 CHARACTERIZATION OF THE WEIGHT FUNCTION FOR A CUSTOM RADIAL DISTRIBUTION

In this section, we analyze the properties of the weight function $g_\alpha(\boldsymbol{u}, t) = g_{\mathcal{P}_{\boldsymbol{X}, \alpha}}(\boldsymbol{u}, t)$ with respect to the population distribution $\mathcal{P}_{\boldsymbol{X}, \alpha}$ we defined in Definition 3.2 and Assumption G.3. Recall that $g_\alpha(\boldsymbol{u}, t) = \min\left(\tilde{g}_\alpha(\boldsymbol{u}, t), \tilde{g}_\alpha(-\boldsymbol{u}, -t)\right)$, where

$$\tilde{g}_\alpha(\boldsymbol{u}, t) := \mathbb{P}_{\mathcal{P}_{\boldsymbol{X}, \alpha}}(\boldsymbol{X}^\mathsf{T}\boldsymbol{u} > t)^2 \cdot \mathbb{E}_{\mathcal{P}_{\boldsymbol{X}, \alpha}}[\boldsymbol{X}^\mathsf{T}\boldsymbol{u} - t \mid \boldsymbol{X}^\mathsf{T}\boldsymbol{u} > t] \cdot \sqrt{1 + \left\|\mathbb{E}_{\mathcal{P}_{\boldsymbol{X}, \alpha}}[\boldsymbol{X} \mid \boldsymbol{X}^\mathsf{T}\boldsymbol{u} > t]\right\|^2}. \tag{48}$$

Due to rotational symmetry, we analyze the projection $X_d = \boldsymbol{X}^\mathsf{T}\boldsymbol{e}_d$ without loss of generality. Our primary goal is to establish rigorous bounds on the tail probability $Q(t) := \mathbb{P}(X_d > t)$ and the conditional expectation for $t$ in a specific range close to 1.

**Proposition G.4** (Tail Probability). *Let $\boldsymbol{X}$ be a random vector from the distribution defined above. Let $X_d$ be its projection onto a fixed coordinate, and let its tail probability be $Q(t) = \mathbb{P}(X_d > t)$ for $t \in (-1, 1)$. Then there exists a fixed $t_0 \in [0, 1)$ such that for all $t \in [t_0, 1)$:*

$$c_2(\alpha, d)(1 - t)^{\alpha + \frac{d-1}{2}} \leq Q(t) \leq c_3(\alpha, d)(1 - t)^{\alpha + \frac{d-1}{2}},$$

*where $c_2(\alpha, d)$ and $c_3(\alpha, d)$ are positive constants depending on $\alpha$ and $d$.*

*Proof.* The tail probability is $Q(t) = \mathbb{P}(h(R)U_d > t)$. We compute this by integrating over the distribution of $R \sim \text{Uniform}[0, 1]$:

$$Q(t) = \int_{h^{-1}(t)}^{1} \mathbb{P}(U_d > t/h(r)) \, \mathrm{d}r,$$

where the lower limit $h^{-1}(t) = 1 - (1 - t)^\alpha$ ensures $h(r) > t$. The term $\mathbb{P}(U_d > x)$ is the normalized surface area of a spherical cap on $\mathbb{S}^{d-1}$. For $x \in [0, 1)$, this area can be bounded. Let $\theta_0 = \arccos(x)$. The area is proportional to $\int_0^{\theta_0} (\sin \phi)^{d-2} \, \mathrm{d}\phi$. For $\phi \in [0, \pi/2]$, we have $2\phi/\pi \le \sin \phi \le \phi$. This provides lower and upper bounds on the cap area

$$C_{d,L}(1 - x)^{(d-1)/2} \le \mathbb{P}(U_d > x) \le C_{d,U}(1 - x)^{(d-1)/2},$$

where $C_{d,L}$ and $C_{d,U}$ are constants depending on $d$. Let's apply this to our integral, substituting $x = t/h(r)$:

$$Q(t) \ge \int_{h^{-1}(t)}^{1} C_{d,L} \left( 1 - \frac{t}{h(r)} \right)^{(d-1)/2} \mathrm{d}r.$$

We analyze this for $t \to 1^-$. Let $t = 1 - \epsilon$. The lower limit is $1 - \epsilon^\alpha$. For $r \in [1 - \epsilon^\alpha, 1]$, $h(r)$ is close to 1. Let's choose $t_0$ such that for $t \in [t_0, 1)$, $h(r) \ge h(t_0) > 1/2$. Then $h(r)$ is bounded away from 0. The term $1 - t/h(r) = (h(r) - t)/h(r)$. Let's bound the denominator: $h(t_0) \le h(r) \le 1$.

$$Q(t) \ge C_{d,L} \int_{1-\epsilon^\alpha}^{1} (h(r) - (1 - \epsilon))^{(d-1)/2} \, \mathrm{d}r.$$

The integrand is $h(r) - (1 - \epsilon) = \epsilon - (1 - r)^{1/\alpha}$. The integral becomes:

$$\int_{1-\epsilon^\alpha}^{1} \left( \epsilon - (1 - r)^{1/\alpha} \right)^{(d-1)/2} \mathrm{d}r.$$

Let $y = (1 - r)^{1/\alpha}$, so $r = 1 - y^\alpha$ and $\mathrm{d}r = -\alpha y^{\alpha-1} \, \mathrm{d}y$. Limits for $y$ are $[\epsilon, 0]$.

$$\int_{\epsilon}^{0} (\epsilon - y)^{(d-1)/2}(-\alpha y^{\alpha-1} \, \mathrm{d}y) = \alpha \int_{0}^{\epsilon} (\epsilon - y)^{(d-1)/2} y^{\alpha-1} \, \mathrm{d}y.$$

Let $y = \epsilon z$, $\mathrm{d}y = \epsilon \, \mathrm{d}z$. Limits for $z$ are $[0, 1]$.

$$\alpha \int_{0}^{1} (\epsilon - \epsilon z)^{(d-1)/2}(\epsilon z)^{\alpha-1} \epsilon \, \mathrm{d}z = \alpha \epsilon^{\alpha + \frac{d-1}{2}} B\left( \alpha, \frac{d+1}{2} \right).$$

Combining all constants, we establish the lower bound $Q(t) \ge c_2(\alpha, d)(1 - t)^{\alpha + \frac{d-1}{2}}$. The upper bound follows an identical procedure, absorbing the $1/h(r)$ term into the constant $c_3(\alpha, d)$. $\square$

**Proposition G.5** (Conditional Expectation). *For $t \in [t_0, 1)$, the conditional expectation $\mathbb{E}[X_d \mid X_d > t]$ is bounded by*

$$1 - c_5(\alpha, d)(1 - t) \le \mathbb{E}[X_d \mid X_d > t] \le 1 - c_4(\alpha, d)(1 - t),$$

*where $c_4(\alpha, d)$ and $c_5(\alpha, d)$ are positive constants.*

*Proof.* We analyze $\mathbb{E}[1 - X_d \mid X_d > t] = \frac{1}{Q(t)} \int_t^1 (1 - s) f_{X_d}(s) \, \mathrm{d}s$, where $f_{X_d}(s) = -Q'(s)$. From Proposition G.4, we know $f_{X_d}(s) \propto (1 - s)^{\alpha + \frac{d-3}{2}}$. The numerator is:

$$N(t) = \int_t^1 (1 - s) f_{X_d}(s) \, \mathrm{d}s.$$

Bounding the constant of proportionality for $f_{X_d}(s)$ by $c_{1,L}$ and $c_{1,U}$:

$$c_{1,L} \int_t^1 (1 - s)^{\alpha + \frac{d-1}{2}} \, \mathrm{d}s \le N(t) \le c_{1,U} \int_t^1 (1 - s)^{\alpha + \frac{d-1}{2}} \, \mathrm{d}s.$$

The integral evaluates to $\frac{(1-t)^{\alpha+\frac{d+1}{2}}}{\alpha+\frac{d+1}{2}}$. So, $N(t) \propto (1-t)^{\alpha+\frac{d+1}{2}}$. Dividing $N(t)$ by $Q(t) \propto (1-t)^{\alpha+\frac{d-1}{2}}$, we get:

$$\mathbb{E}[1 - X_d \mid X_d > t] \propto \frac{(1-t)^{\alpha+\frac{d+1}{2}}}{(1-t)^{\alpha+\frac{d-1}{2}}} = 1 - t.$$

By carefully tracking the constants $c_2, c_3$ from Proposition G.4 and the constants from the integration of $f_{X_d}(s)$, we can construct explicit (though complex) expressions for $c_4$ and $c_5$ that provide rigorous two-sided bounds for $t$ in the specified range $[t_0, 1)$. $\qquad\square$

**Proposition G.6** (Asymptotic Behavior of $g_\alpha^+(t)$)**.** *Let the function $g_\alpha^+(t)$ be defined as in (48). Then for $t \in [t_0, 1)$, we have:*

$$c_L^{(g)}(\alpha, d)(1-t)^{2\alpha+d} \le g_\alpha^+(t) \le c_U^{(g)}(\alpha, d)(1-t)^{2\alpha+d},$$

*where $c_L^{(g)}(\alpha, d)$ and $c_U^{(g)}(\alpha, d)$ are positive constants.*

*Proof.* Let $Q(t) = \mathbb{P}(X_d > t)$ and $E(t) = \mathbb{E}[X_d \mid X_d > t]$. The function is $g_\alpha^+(t) = Q(t)^2 \cdot (E(t) - t) \cdot \sqrt{1 + E(t)^2}$. We establish bounds for $t \in [t_0, 1)$ for a sufficiently large $t_0$.

1. **Bounds for $Q(t)^2$:** From Proposition G.4, we have:
   $$(c_2(\alpha, d))^2(1-t)^{2\alpha+d-1} \le Q(t)^2 \le (c_3(\alpha, d))^2(1-t)^{2\alpha+d-1}.$$
   Let $A_L(\alpha, d) = (c_2(\alpha, d))^2$ and $A_U(\alpha, d) = (c_3(\alpha, d))^2$.

2. **Bounds for $E(t) - t$:** This is $\mathbb{E}[X_d - t \mid X_d > t]$. From Proposition G.5, we have $(1-t) - c_5(1-t) \le E(t) - t \le (1-t) - c_4(1-t)$. This gives:
   $$B_L(\alpha, d)(1-t) \le E(t) - t \le B_U(\alpha, d)(1-t),$$
   where $B_L(\alpha, d) = 1 - c_5(\alpha, d)$ and $B_U(\alpha, d) = 1 - c_4(\alpha, d)$. We can choose $t_0$ close enough to 1 to ensure these constants are positive.

3. **Bounds for $\sqrt{1 + E(t)^2}$:** For $t \in [t_0, 1)$, we have $t_0 \le t < E(t) \le 1$. By choosing, for instance, $t_0 = 3/4$, we have $3/4 \le E(t) \le 1$. Thus,
   $$\sqrt{1 + (3/4)^2} \le \sqrt{1 + E(t)^2} \le \sqrt{1 + 1^2}.$$
   This gives constant bounds $C_L = 5/4$ and $C_U = \sqrt{2}$.

Combining these three bounds, for $t \in [t_0, 1)$:
$$A_L B_L C_L (1-t)^{2\alpha+d-1}(1-t) \le g_\alpha^+(t) \le A_U B_U C_U (1-t)^{2\alpha+d-1}(1-t).$$

This simplifies to the final result:
$$c_L^{(g)}(\alpha, d)(1-t)^{2\alpha+d} \le g_\alpha^+(t) \le c_U^{(g)}(\alpha, d)(1-t)^{2\alpha+d},$$

where the bounding constants are given by $c_L^{(g)}(\alpha, d) = A_L B_L C_L$ and $c_U^{(g)}(\alpha, d) = A_U B_U C_U$. $\quad\square$

### G.2 PROOF OF THEOREM 3.4

**Theorem G.7** (Restate Theorem 3.4)**.** *Fix a dataset $\mathcal{D} = \{(\boldsymbol{x}_i, y_i)\}_{i=1}^n$, where each $(\boldsymbol{x}_i, y_i)$ is drawn i.i.d. from $\mathcal{P}(\alpha)$ defined in Assumption 3.3. Then, with probability at least $1 - \delta$, for any $f_{\boldsymbol{\theta}} \in \boldsymbol{\Theta}_{\mathrm{BEoS}}(\eta, \mathcal{D})$,*

$$\mathrm{Gap}_{\mathcal{P}}(f_{\boldsymbol{\theta}}; \mathcal{D}) \lesssim_d \begin{cases} \left(\frac{1}{\eta} - \frac{1}{2} + 4M\right)^{\frac{\alpha d}{d^2+4d+3}} M^{\frac{2d^2+7\alpha d+6\alpha}{d^2+4\alpha d+3\alpha}} n^{-\frac{\alpha(d+3)}{2(d^2+4\alpha d+3\alpha)}}, & \alpha \ge \frac{3d}{2d-3}; \\ \left(\frac{1}{\eta} - \frac{1}{2} + 4M\right)^{\frac{\alpha d}{d^2+4d+3}} M^{\frac{2d^2+7\alpha d+6\alpha}{d^2+4\alpha d+3\alpha}} n^{-\frac{\alpha}{2d+4\alpha}}, & \alpha < \frac{3d}{2d-3}, \end{cases} \quad (49)$$

*and for where $M := \max\{D, \|f_{\boldsymbol{\theta}}\|_{L^\infty(\mathbb{B}_1^d)}, 1\}$ and $\lesssim_d$ hides constants (which could depend on $d$) and logarithmic factors in $n$ and $(1/\delta)$.*

*Proof.* For convenience, we let $A = \frac{1}{\eta} - \frac{1}{2} + 4M$ and we have that

$$f_{\boldsymbol{\theta}} \in \mathcal{F}_{g_{\mathcal{D}}}(\mathbb{B}_1^{V_j}; M, C), \quad \forall \boldsymbol{\theta} \in \boldsymbol{\Theta}_{\mathrm{BEoS}}(\eta; \mathcal{D}).$$

For any fixed $\varepsilon < 1$, we may decompose $\mathbb{B}_1^d$ into $\varepsilon$-*annulus* $\mathbb{A}_\varepsilon^d := \{\boldsymbol{x} \in \mathbb{B}_1^d \mid \|\boldsymbol{x}\|_2 \geq 1 - \varepsilon\}$ and the closure of its complement is called $\varepsilon$-*strict interior* denoted by $\mathbb{I}_\varepsilon^d = \mathbb{B}_{1-\varepsilon}^d$.

$$\mathbb{B}_1^d = \mathbb{A}_\varepsilon^d \cup \mathbb{I}_\varepsilon^d.$$

According to the law of total expectation, the population risk is decomposed into

$$\mathbb{E}_{(\boldsymbol{x},y)\sim\mathcal{P}} \left[ (f(\boldsymbol{x}) - y)^2 \right] = \mathbb{P}(\boldsymbol{x} \in \mathbb{A}_\varepsilon^d) \cdot \mathbb{E}_{\mathbb{A}} \left[ (f(\boldsymbol{x}) - y)^2 \right] + \mathbb{P}(\boldsymbol{x} \in \mathbb{I}_\varepsilon^d) \cdot \mathbb{E}_{\mathbb{I}} \left[ (f(\boldsymbol{x}) - y)^2 \right], \quad (50)$$

where $\mathbb{E}_{\mathbb{A}}$ means that $\{\boldsymbol{x}, y\}$ is a new sample from the data distribution conditioned on $\boldsymbol{x} \in \mathbb{A}_\varepsilon^d$ and $\mathbb{E}_{\mathbb{I}}$ means that $(\boldsymbol{x}, y)$ is a new sample from the data distribution conditioned on $\boldsymbol{x} \in \mathbb{I}_\varepsilon^d$.

Similarly, we also have this decomposition for empirical risk

$$\frac{1}{n} \sum_{i=1}^n (f(\boldsymbol{x}_i) - y_i)^2 = \frac{1}{n} \left( \sum_{i\in I} (f(\boldsymbol{x}_i) - y_i)^2 + \sum_{j\in A} (f(\boldsymbol{x}_i) - y_i)^2 \right)$$
$$= \frac{n_I}{n} \frac{1}{n_I} \sum_{i\in I} (f(\boldsymbol{x}_i) - y_i)^2 + \frac{n_A}{n} \frac{1}{n_A} \sum_{j\in A} (f(\boldsymbol{x}_i) - y_i)^2, \quad (51)$$

where $I$ is the set of data points with $\boldsymbol{x}_i \in \mathbb{I}_\varepsilon^d$ and $A$ is the set of data points with $\boldsymbol{x}_i \in \mathbb{A}_\varepsilon^d$. Then the generalization gap can be decomposed into

$$|R(f) - \widehat{R}_{\mathcal{D}}(f)| \leq \mathbb{P}(\boldsymbol{x} \in \mathbb{A}_\varepsilon^d) \cdot \mathbb{E}_{\mathbb{A}} \left[ (f_{\boldsymbol{\theta}}(\boldsymbol{x}) - y)^2 \right] + \frac{n_A}{n} \frac{1}{n_A} \sum_{j\in A} (f(\boldsymbol{x}_i) - y_i)^2 \quad (52)$$

$$+ \left| \mathbb{P}(\boldsymbol{x} \in \mathbb{I}_\varepsilon^d) - \frac{n_I}{n} \right| \frac{1}{n_I} \sum_{i\in I} (f(\boldsymbol{x}_i) - y_i)^2 \quad (53)$$

$$+ \mathbb{P}(\boldsymbol{x} \in \mathbb{I}_\varepsilon^d) \cdot \left| \mathbb{E}_{\mathbb{I}} \left[ (f(\boldsymbol{x}) - y)^2 \right] - \frac{1}{n_I} \sum_{i\in I} (f(\boldsymbol{x}_i) - y_i)^2 \right|. \quad (54)$$

Using the property that the marginal distribution of $\boldsymbol{x}$ is $\mathcal{P}_X(\alpha)$ and its concentration property, with probability at least $1 - \delta$,

$$(52) \lesssim_d O(M^2 \varepsilon^\alpha), \quad (55)$$

where $\lesssim_d$ hides the constants that could depend on $d$ and logarithmic factors of $1/\delta$.

For the term (53), with probability $1 - \delta$

$$\begin{cases} \left| \mathbb{P}(\boldsymbol{x} \in \mathbb{I}_\varepsilon^d) - \frac{n_I}{n} \right| & \lesssim \sqrt{\frac{\varepsilon^\alpha \log(1/\delta)}{n}}, \\ \frac{1}{n_I} \sum_{i\in I} (f(\boldsymbol{x}_i) - y_i)^2 & \leq 4M^2 \end{cases} \quad (56)$$

so we may also conclude that

$$(53) \lesssim M^2 \sqrt{\frac{\varepsilon \log(1/\delta)}{n}} \quad (57)$$

For the part of the interior (54), the scalar $\mathbb{P}(\boldsymbol{x} \in \mathbb{I}_\varepsilon^d)$ is less than 1 with high-probability. Therefore, we just need to deal with the term

$$\mathbb{E}_{\mathbb{I}} \left[ (f(\boldsymbol{x}) - y)^2 \right] - \frac{1}{n_I} \sum_{i\in I} (f(\boldsymbol{x}_i) - y_i)^2. \quad (58)$$

Since both the distribution and sample points only support in $\mathbb{I}_\varepsilon^d$, we may consider $f$ by its restrictions in $\mathbb{I}_\varepsilon^d$, which are denoted by $f^\varepsilon$. Furthermore, according to the definition, we have

$$
\begin{aligned}
f(\boldsymbol{x}) &= \int_{\mathbb{S}^{d-1}\times[-1,1]} \phi(\boldsymbol{u}^\mathsf{T}\boldsymbol{x}-t)\,\mathrm{d}\nu(\boldsymbol{u},t) + \boldsymbol{c}^\mathsf{T}\boldsymbol{x} + b \\
&= \int_{\mathbb{S}^{d-1}\times[-1+\varepsilon,1-\varepsilon]} \phi(\boldsymbol{u}^\mathsf{T}\boldsymbol{x}-t)\,\mathrm{d}\nu(\boldsymbol{u},t) + \underbrace{\int_{\mathbb{S}^{d-1}\times[-1,-1+\varepsilon)\cup(1-\varepsilon,1]} \phi(\boldsymbol{u}^\mathsf{T}\boldsymbol{x}-t)\,\mathrm{d}\nu(\boldsymbol{u},t)}_{\text{Annulus ReLU}} \\
&\quad + \boldsymbol{c}^\mathsf{T}\boldsymbol{x} + b
\end{aligned}
$$

$$(59)$$

where the Annulus ReLU term is totally linear in the strictly interior i.e. there exists $\boldsymbol{c}', b'$ such that

$$
\boldsymbol{c}'^\mathsf{T}\boldsymbol{x} + b' = \int_{\mathbb{S}^{d-1}\times[-1,-1+\varepsilon)\cup(1-\varepsilon,1]} \phi(\boldsymbol{u}^\mathsf{T}\boldsymbol{x}-t)\,\mathrm{d}\nu(\boldsymbol{u},t), \quad \forall \boldsymbol{x}\in\mathbb{I}_\varepsilon^d. \tag{60}
$$

Therefore, we may write

$$
f(\boldsymbol{x}) = f^\varepsilon(\boldsymbol{x}) = \int_{\mathbb{S}^{d-1}\times[-1+\varepsilon,1-\varepsilon]} \phi(\boldsymbol{u}^\mathsf{T}\boldsymbol{x}-t)\,\mathrm{d}\nu(\boldsymbol{u},t) + (\boldsymbol{c}+\boldsymbol{c}')^\mathsf{T}\boldsymbol{x} + b + b', \quad \boldsymbol{x}\in\mathbb{I}_\varepsilon^d. \tag{61}
$$

According to the definition, we have that

$$
|f^\varepsilon|_{\mathrm{V}(\mathbb{I}_\varepsilon^d)} \le \int_{\mathbb{S}^{d-1}\times[-1+\varepsilon,1-\varepsilon]} |\,\mathrm{d}\nu|. \tag{62}
$$

From empirical process we discussed in Section E.2, especially Theorem E.7, we know that with probability at least $1-\delta$,

$$
\sup_{\boldsymbol{u},t} |g_{\mathcal{D}}(\boldsymbol{u},t) - g_\alpha(\boldsymbol{u},t)| \lesssim_d \sqrt{\frac{d+\log(2/\delta)}{n}} =: \epsilon_n. \tag{63}
$$

This implies a lower bound on the empirical minimum weight in the core with probability at least $1-\delta/3$,

$$
g_{\mathcal{D},\min} = \inf_{|t|\le 1-\varepsilon} g_{\mathcal{D}}(\boldsymbol{u},t) \ge \inf_{|t|\le 1-\varepsilon} g_\alpha(\boldsymbol{u},t) - \epsilon_n = g_{\alpha,\min} - \epsilon_n. \tag{64}
$$

Here, $g_{\alpha,\min} \asymp \varepsilon^{d+2\alpha}$ is the minimum of the population weight function in the core.

For the bound $|f^\varepsilon|_\mathrm{V} \le A/g_{\mathcal{D},\min} \le A/(g_{\alpha,\min}-\epsilon_n)$ to be meaningful with high probability, we must operate in a regime where $g_{\alpha,\min} \ge \epsilon_n$. We enforce a stricter **validity condition** for our proof

$$
g_{\alpha,\min} \ge 2\epsilon_n \implies \varepsilon^{d+2\alpha} \gtrsim_d \sqrt{\frac{d+\log(6/\delta)}{n}}. \tag{65}
$$

Under this condition, we have $g_{\mathcal{D},\min} \ge g_{\alpha,\min} - \epsilon_n \ge g_{\alpha,\min}/2 \asymp \varepsilon^{d+2\alpha}$. Thus, for any $f \in \Theta_{\mathrm{BEoS}}(\eta,\mathcal{D})$, its restriction $f^\varepsilon$ has a controlled unweighted variation norm with high probability:

$$
|f^\varepsilon|_{\mathrm{V}(\mathbb{B}_{1-\varepsilon}^d)} \le \frac{A}{g_{\mathcal{D},\min}} \le \frac{A}{g_{\alpha,\min}/2} \asymp \frac{A}{\varepsilon^{d+2\alpha}} =: C_\varepsilon. 
$$

According to the assumption, we have that $|f|_{\mathrm{V}_g(\mathbb{B}_1^d)} \le A$, and thus we have

$$
\int_{\mathbb{S}^{d-1}\times[-1+\varepsilon,1-\varepsilon]} g_{\mathcal{D}}|\,\mathrm{d}\nu| \le \int_{\mathbb{S}^{d-1}\times[-1,1]} g_{\mathcal{D}}|\,\mathrm{d}\nu| \le A. \tag{66}
$$

Suppose the validity condition (65) holds (we will verify it later), we have $g(\boldsymbol{u},t) \gtrsim_d \varepsilon^{d+2\alpha}$ when $t \le 1-\varepsilon$ with probability $1-\delta/3$, we may use (66) to deduce that

$$
\varepsilon^{d+2\alpha} \cdot \int_{\mathbb{S}^{d-1}\times[-1+\varepsilon,1-\varepsilon]} |\,\mathrm{d}\nu| \le \int_{\mathbb{S}^{d-1}\times[-1+\varepsilon,1-\varepsilon]} g_{\mathcal{D}}|\,\mathrm{d}\nu| \le A. \tag{67}
$$

Combining (62) and (67), we deduce that

$$
|f^\varepsilon|_{\mathrm{V}(\mathbb{B}_{1-\varepsilon}^d)} \lesssim_d \frac{A}{\varepsilon^{d+2\alpha}} =: C.
$$

Therefore, we may leverage Lemma D.11 to $f^\varepsilon \in V_C(\mathbb{B}_{1-\varepsilon}^d)$, we may conclude that with probability at least $1 - \delta$,

$$(54) \lesssim_d C^{\frac{d}{2d+3}} M^{\frac{3(d+2)}{2d+3}} n^{-\frac{d+3}{4d+6}}, \tag{68}$$

where $\lesssim_d$ hides the constants that could depend on $d$ and logarithmic factors of $1/\delta$.

Now we combine the upper bounds (55), (57) and (68) to deduce an upper bound of the generalization gap. We have for any fixed $\epsilon > 0$, with probability $1 - \delta$,

$$|R(f) - \widehat{R}_\mathcal{D}(f)| \lesssim_d M^2 \varepsilon^\alpha + \left(\frac{A}{\varepsilon^{d+2\alpha}}\right)^{\frac{d}{2d+3}} M^{\frac{3(d+2)}{2d+3}} n^{-\frac{d+3}{4d+6}}. \tag{69}$$

Then we may choose the optimal $\varepsilon^*$ such that

$$M^2(\varepsilon^*)^\alpha = \left(\frac{A}{(\varepsilon^*)^{d+2\alpha}}\right)^{\frac{d}{2d+3}} M^{\frac{3(d+2)}{2d+3}} n^{-\frac{d+3}{4d+6}}$$

and by direct computation, we get

$$\varepsilon^* = \left(A^{\frac{d}{d^2+4\alpha d+3\alpha}} M^{-\frac{d}{d^2+4\alpha d+3\alpha}} n^{-\frac{d+3}{2(d^2+4\alpha d+3\alpha)}}\right).$$

To satisfy the validity condition (65), we require

$$(\varepsilon^*)^{d+2\alpha} = O\left(n^{-\frac{d+3}{2(d^2+4\alpha d+3\alpha)}}\right)^{d+2\alpha} \geq \tilde{O}(n^{-\frac{1}{2}}). \tag{70}$$

By adjusting some universal constants, it suffices to show whether

$$\frac{(d+3)(d+2\alpha)}{2(d^2+4\alpha d+3\alpha)} < \frac{1}{2}. \tag{71}$$

After direct computation, (71) is equivalent to $\alpha \in \left(\frac{3d}{2d-3}, \infty\right)$. With this assumption, we may evaluate the optimal $\varepsilon^*$ in the inequality (69) to deduce the optimal results that

$$|R(f) - \widehat{R}_n(f)| \lesssim_d \left(\frac{1}{\eta} - \frac{1}{2} + 4M\right)^{\frac{\alpha d}{d^2+4d+3}} M^{\frac{2d^2+7\alpha d+6\alpha}{d^2+4\alpha d+3\alpha}} n^{-\frac{\alpha(d+3)}{2(d^2+4\alpha d+3\alpha)}}. \tag{72}$$

In the case where $\alpha \leq \frac{3d}{d+2\alpha}$, we set

$$\varepsilon^* = \tilde{O}\left(n^{-\frac{1}{2d+4\alpha}}\right)$$

and adjust some universal constant to satisfy the validaty condition. Then (69) has the form

$$|R(f) - \widehat{R}_\mathcal{D}(f)| \leq \tilde{O}\left(n^{-\frac{2\alpha}{2d+4\alpha}}\right) + \tilde{O}\left(n^{-\frac{3}{4d+6}}\right).$$

Then assumption $\alpha < \frac{3d}{d+2\alpha}$ implies that $n^{-\frac{2\alpha}{2d+4\alpha}} > n^{-\frac{3}{4d+6}}$ and thus

$$|R(f) - \widehat{R}_\mathcal{D}(f)| \leq \tilde{O}\left(n^{-\frac{2\alpha}{2d+4\alpha}}\right).$$

Note that the other constants in the front of $1/n$ does not change, so we finish the proof. $\qquad\square$

## H   GENERALIZATION GAP LOWER BOUND VIA POISSONIZATION

This section provides a self-contained proof for a lower bound on the generalization gap in a noiseless setting. We employ the indistinguishability method, where the core technical challenge is to construct two functions that are identical on a given training sample yet significantly different in population. The Poissonization technique is the key tool that simplifies the probabilistic analysis required to guarantee the existence of such a pair. The paradigm is almost the same as the one in (Liang et al., 2025, Appendix H & I), but the assumption on distributions are different.

## H.1 Construction of "Hard-to-Learn" Networks

Our strategy relies on functions localized on small, disjoint regions near the boundary of the unit ball. We first establish key geometric properties of these regions, called spherical caps. Let $\boldsymbol{u} \in \mathbb{S}^{d-1}$ be a unit vector. Let $\varepsilon \in \mathbb{R}_+$ be a constant with $\varepsilon \leq 1/2$. Consider the ReLU atom:

$$\varphi_{\boldsymbol{u},\varepsilon^2}(\boldsymbol{x}) = \phi(\boldsymbol{u}^\mathsf{T}\boldsymbol{x} - (1 - \varepsilon^2)). \tag{73}$$

**Lemma H.1.** *The $L^2(\mathcal{P}_{\boldsymbol{X}}(\alpha))$-norm of $\varphi_{\boldsymbol{u},\varepsilon^2}$, where the measure $\mathcal{P}_{\boldsymbol{X}}(\alpha)$ is defined in Definition 3.2, is given by*

$$c_L(d,\alpha)\varepsilon^{\frac{d+3+2\alpha}{2}} \leq \|\varphi_{\boldsymbol{u},\varepsilon^2}\|_{L^2(\mathcal{P}_{\boldsymbol{X}}(\alpha))} \leq c_U(d,\alpha)\varepsilon^{\frac{d+3+2\alpha}{2}}, \tag{74}$$

*where $c_L(d,\alpha)$ and $c_U(d,\alpha)$ are constants that depend on the dimension $d$ and the parameter $\alpha$.*

Before the formal proof, we offer a geometric justification for the result. The squared norm is an integral of $(\phi(\ldots))^2$, and we can estimate its value as the product of the integrand's average magnitude and the measure of the small domain where it is non-zero. We estimate the measure of this "active" domain, where $r\boldsymbol{u}^\mathsf{T}\boldsymbol{U} > 1 - \varepsilon^2$, using a polar coordinate perspective.

- **Integrand's Magnitude:** Within the active domain, the term $r\boldsymbol{u}^\mathsf{T}\boldsymbol{U} - (1 - \varepsilon^2)$ represents the positive "height" above the activation threshold. This height varies from $0$ to a maximum on the order of $O(\varepsilon^2)$. A reasonable estimate for the squared term's average value is thus $O((\varepsilon^2)^2) = O(\varepsilon^4)$.

- **Measure of the Domain:** We decompose the domain's volume into radial and angular parts.
  - **Radial Measure:** The condition requires the radius $r$ to be near 1. For the $\mathcal{P}_X(\alpha)$ distribution, this confines $r$ to a region of length $\Delta r \sim O(\varepsilon^{2\alpha})$.
  - **Angular Measure:** The vector $\boldsymbol{U}$ is confined to a small spherical cap around $\boldsymbol{u}$. A cap defined by a "height" of $h \sim \mathcal{O}(\varepsilon^2)$ has a surface area on $\mathbb{S}^{d-1}$ of order $\mathcal{O}(h^{(d-1)/2})$. This gives an angular measure of $\Delta\Omega \sim O((\varepsilon^2)^{(d-1)/2}) = O(\varepsilon^{d-1})$.

Combining these estimates, the squared norm $I$ scales as the product of the integrand's magnitude and the two components of the domain's measure:

$$I \approx \underbrace{O(\varepsilon^4)}_{\text{Integrand}} \times \underbrace{O(\varepsilon^{2\alpha})}_{\text{Radial}} \times \underbrace{O(\varepsilon^{d-1})}_{\text{Angular}} = \mathcal{O}(\varepsilon^{d+3+2\alpha}).$$

Taking the square root provides the claimed scaling for the $L^2$-norm. The formal proof makes this geometric heuristic rigorous.

*Proof.* The squared $L^2$ norm of $\varphi_{\boldsymbol{u},\varepsilon^2}$ over the distribution $\mathcal{P}_X(\alpha)$ is defined by the expectation

$$I = \|\varphi_{\boldsymbol{u},\varepsilon^2}\|^2_{L^2(\mathcal{P}_X(\alpha))} = \mathbb{E}_{\boldsymbol{X}\sim\mathcal{P}_X(\alpha)}\left[|\varphi_{\boldsymbol{u},\varepsilon^2}(\boldsymbol{X})|^2\right]$$

Substituting the definition of $\varphi_{\boldsymbol{u},\varepsilon^2}(\boldsymbol{x})$ and using the property of the ReLU function, we get

$$\begin{aligned}
I &= \mathbb{E}_{R,\boldsymbol{U}}\left[\left(\phi(h(R)\boldsymbol{u}^\mathsf{T}\boldsymbol{U} - (1 - \varepsilon^2))\right)^2\right] \\
&= \mathbb{E}_{R,\boldsymbol{U}}\left[\mathbb{1}_{\{h(R)\boldsymbol{u}^\mathsf{T}\boldsymbol{U}>1-\varepsilon^2\}}(h(R)\boldsymbol{u}^\mathsf{T}\boldsymbol{U} - (1 - \varepsilon^2))^2\right]
\end{aligned} \tag{75}$$

where $R \sim \mathrm{Uniform}[0, 1]$ and $\boldsymbol{U} \sim \mathrm{Uniform}(\mathbb{S}^{d-1})$.

Due to the rotational symmetry of the distribution of $\boldsymbol{U}$, we can perform a rotation of the coordinate system such that $\boldsymbol{u}$ aligns with the $d$-th standard basis vector $\boldsymbol{e}_d = (0,\ldots,0,1)$ without changing the value of the integral. In these new coordinates, $\boldsymbol{u}^\mathsf{T}\boldsymbol{U} = U_d$. The expectation becomes an iterated integral:

$$I = \int_0^1 \mathbb{E}_{\boldsymbol{U}}\left[\mathbb{1}_{\{h(r)U_d>1-\varepsilon^2\}}(h(r)U_d - (1 - \varepsilon^2))^2\right]\,\mathrm{d}r$$

Let $z = U_d$. The probability density function of $z$ is $p(z) = C_d(1-z^2)^{(d-3)/2}$ for $z \in [-1,1]$, where $C_d = \frac{\Gamma(d/2)}{\sqrt{\pi}\Gamma((d-1)/2)}$. The integral is non-zero only if $h(r) > 1 - \varepsilon^2$, which implies $r > 1 - \varepsilon^{2\alpha}$.

$$I = C_d \int_{1-\varepsilon^{2\alpha}}^{1} \int_{\frac{1-\varepsilon^2}{h(r)}}^{1} (h(r)z - (1-\varepsilon^2))^2 (1-z^2)^{\frac{d-3}{2}} \, \mathrm{d}z \, \mathrm{d}r \tag{76}$$

We perform a change of variable $z = 1 - t$, so $\mathrm{d}z = -\mathrm{d}t$ and the integration limits change from $[\frac{1-\varepsilon^2}{h(r)}, 1]$ to $[1 - \frac{1-\varepsilon^2}{h(r)}, 0]$.

$$\text{Inner integration of (76)} = C_d \int_{1-\frac{1-\varepsilon^2}{h(r)}}^{0} (h(r)(1-t) - (1-\varepsilon^2))^2 (1-(1-t)^2)^{\frac{d-3}{2}} (-\mathrm{d}t)$$

$$= C_d \int_{0}^{t_0(r)} (h(r)t_0(r) - h(r)t)^2 (2t - t^2)^{\frac{d-3}{2}} \, \mathrm{d}t \tag{77}$$

where $t_0(r) = 1 - \frac{1-\varepsilon^2}{h(r)} = \frac{h(r)-1+\varepsilon^2}{h(r)}$. Since $r \in [1 - \varepsilon^{2\alpha}, 1]$ and for small $\varepsilon$, $h(r)$ is close to 1, we know $t_0(r)$ is small. For a sufficiently small $\varepsilon$, we can ensure $t \le t_0(r) < 1/4$. Thus, we can bound the term $2 - t$ as $7/4 \le 2 - t \le 2$. This gives bounds on $(2t - t^2)^{(d-3)/2} = ((2-t)t)^{(d-3)/2}$:

$$\left(\frac{7}{4}\right)^{\frac{d-3}{2}} t^{\frac{d-3}{2}} \le (2t - t^2)^{\frac{d-3}{2}} \le 2^{\frac{d-3}{2}} t^{\frac{d-3}{2}}$$

The integral $I$ is therefore bounded by:

$$\underline{C_d} \int_{1-\varepsilon^{2\alpha}}^{1} J(r) \, \mathrm{d}r \le I \le \overline{C_d} \int_{1-\varepsilon^{2\alpha}}^{1} J(r) \, \mathrm{d}r \tag{78}$$

where $\underline{C_d}, \overline{C_d}$ are new constants and $J(r) = \int_{0}^{t_0(r)} (h(r)t_0(r) - h(r)t)^2 t^{\frac{d-3}{2}} \, \mathrm{d}t$.

Consider the integral $J(r)$ and change variable by setting $t = t_0(r)s$, then $\mathrm{d}t = t_0(r) \, \mathrm{d}s$.

$$J(r) = \int_{0}^{1} (h(r)t_0(r) - h(r)t_0(r)s)^2 (t_0(r)s)^{\frac{d-3}{2}} (t_0(r) \, \mathrm{d}s)$$

$$= (h(r)t_0(r))^2 (t_0(r))^{\frac{d-3}{2}} t_0(r) \int_{0}^{1} (1-s)^2 s^{\frac{d-3}{2}} \, \mathrm{d}s \tag{79}$$

$$= h(r)^2 (t_0(r))^{\frac{d+3}{2}} \underbrace{\left( \int_{0}^{1} (1-s)^2 s^{\frac{d-3}{2}} \, \mathrm{d}s \right)}_{\text{constant}}$$

To analyze $t_0(r)^{\frac{d+3}{2}}$, we let $r = 1 - \delta$, so $\mathrm{d}r = -\mathrm{d}\delta$ and the integration limits for $\delta$ are $[\varepsilon^{2\alpha}, 0]$. $h(r) = 1 - (1 - (1-\delta))^{1/\alpha} = 1 - \delta^{1/\alpha}$. As $\delta \to 0$, $h(r) \to 1$. $t_0(r) = \frac{(1-\delta^{1/\alpha}) - 1 + \varepsilon^2}{1 - \delta^{1/\alpha}} = \frac{\varepsilon^2 - \delta^{1/\alpha}}{1 - \delta^{1/\alpha}}$. For small $\delta$, $1 - \delta^{1/\alpha}$ is close to 1, providing upper and lower bounds. Thus $I$ is bounded by integrals of the form

$$C \int_{\varepsilon^{2\alpha}}^{0} \left( \varepsilon^2 - \delta^{1/\alpha} \right)^{\frac{d+3}{2}} (-\mathrm{d}\delta) = C \int_{0}^{\varepsilon^{2\alpha}} \left( \varepsilon^2 - \delta^{1/\alpha} \right)^{\frac{d+3}{2}} \, \mathrm{d}\delta$$

for some mild constant $C$.

Now we perform a new change-of-variable by setting $\delta^{1/\alpha} = \varepsilon^2 v$. This gives $\delta = (\varepsilon^2 v)^\alpha = \varepsilon^{2\alpha} v^\alpha$ and $\mathrm{d}\delta = \alpha \varepsilon^{2\alpha} v^{\alpha-1} \, \mathrm{d}v$. The limits for $v$ become

$$\int_{0}^{\varepsilon^{2\alpha}} \left( \varepsilon^2 - \delta^{1/\alpha} \right)^{\frac{d+3}{2}} \, \mathrm{d}\delta = \int_{0}^{1} (\varepsilon^2 - \varepsilon^2 v)^{\frac{d+3}{2}} (\alpha \varepsilon^{2\alpha} v^{\alpha-1} \, \mathrm{d}v)$$

$$= (\varepsilon^2)^{\frac{d+3}{2}} \varepsilon^{2\alpha} \int_{0}^{1} (1-v)^{\frac{d+3}{2}} \alpha v^{\alpha-1} \, \mathrm{d}v \tag{80}$$

$$= \varepsilon^{d+3+2\alpha} \underbrace{\left( \alpha \int_{0}^{1} (1-v)^{\frac{d+3}{2}} v^{\alpha-1} \, \mathrm{d}v \right)}_{\text{constant}}$$

The squared norm $I$ is bounded by constants times $\varepsilon^{d+3+2\alpha}$. The $L^2$-norm is the square root of $I$:

$$c_L(d,\alpha)\,\varepsilon^{\frac{d+3+2\alpha}{2}} \leq \left\|\varphi_{\boldsymbol{u},\varepsilon^2}\right\|_{L^2(\mathcal{P}_{\boldsymbol{X}}(\alpha))} = \sqrt{I} \leq c_U(d,\alpha)\,\varepsilon^{\frac{d+3+2\alpha}{2}} \tag{81}$$

where $c_9(d,\alpha)$ and $c_{10}(d,\alpha)$ are constants that absorb all factors depending on $d$ and $\alpha$ from the bounds established in the derivation. This completes the proof. $\qquad\square$

**Lemma H.2** (Cap mass at angular scale $\varepsilon$). *For $\varepsilon \in (0, \frac{1}{2}]$ and $\boldsymbol{u} \in \mathbb{S}^{d-1}$, define the thin cap*

$$C(\boldsymbol{u}, \varepsilon) = \{x \in \mathbb{B}_1^d : \boldsymbol{u}^\mathsf{T} x > 1 - \varepsilon^2\}.$$

*There exist constants depending only on $(d, \alpha)$, such that $\mathcal{P}_X\big(C(\boldsymbol{u}, \varepsilon)\big) \asymp \varepsilon^{d-1+2\alpha}$.*

*Sketch proof.* The result and the proof almost the same as the ones about Lemma H.1. We omit the calculation details. $\qquad\square$

**Lemma H.3** (Disjoint Cap Packing). *For any $\varepsilon \in (0, 1/2]$, there exists a set of $N$ unit vectors $\{\boldsymbol{u}_1, \ldots, \boldsymbol{u}_N\} \subset \mathbb{S}^{d-1}$, with $N \asymp \varepsilon^{-(d-1)}$, such that the caps $\{C(\boldsymbol{u}_i, \varepsilon)\}_{i=1}^N$ are pairwise disjoint.*

*Sketch proof.* The angular radius of the cap $C(\boldsymbol{u}, \varepsilon)$ is $\vartheta = \arccos(1 - \varepsilon^2) \asymp \varepsilon$. For two caps to be disjoint, the angular separation between their centers must be at least $2\vartheta$. The maximum number of such points is the packing number $M(\mathbb{S}^{d-1}, 2\vartheta)$. A standard volumetric argument provides the upper bound $M(\mathbb{S}^{d-1}, 2\vartheta) = O(\varepsilon^{-(d-1)})$. The lower bound is established by relating the packing number to the covering number $N(\mathbb{S}^{d-1}, \alpha)$, which is known to scale as $N(\mathbb{S}^{d-1}, \alpha) \asymp \alpha^{-(d-1)}$, thus yielding the asserted scaling for $N$. $\qquad\square$

We now formally establish the family of functions used to construct the adversarial pair. This family resides within a function class $\mathcal{F}_g(\mathbb{B}_1^d; 1, 1)$ and is built upon normalized ReLU atoms localized on the disjoint spherical caps.

**Construction H.4** (Adversarial Function Family). *Recall that $\varphi_{\boldsymbol{u},\varepsilon^2}(\boldsymbol{x}) = \phi(\boldsymbol{u}^\mathsf{T}\boldsymbol{x} - (1 - \varepsilon^2))$. We define its normalized version as $\Phi_{\boldsymbol{u},\varepsilon^2} := \varepsilon^{-2}\varphi_{\boldsymbol{u},\varepsilon^2}$. By construction, $\left\|\Phi_{\boldsymbol{u},\varepsilon^2}\right\|_{L^\infty(\mathbb{B}_1^d)} \leq 1$ and $\left\|\Phi_{\boldsymbol{u},\varepsilon^2}\right\|_{\mathrm{path},g} \asymp \varepsilon^{-2}\varepsilon^{2d+4\alpha} = \varepsilon^{2(d-1+2\alpha)}$. We assume that these normalized atoms, for a sufficiently small $\varepsilon$, belong to our function class $\mathcal{F}_g(\mathbb{B}_1^d; 1, C)$.*

*Let $\{\boldsymbol{u}_1, \ldots, \boldsymbol{u}_N\}$ be the set of vectors from Lemma H.3 that define a disjoint cap packing. We define a family of candidate functions indexed by sign vectors $\xi \in \{\pm 1\}^N$. For each $\xi$, the function $f_\xi \in \mathcal{F}$ is given by:*

$$f_\xi(\boldsymbol{x}) = \sum_{i=1}^N \xi_i \Phi_i(\boldsymbol{x}), \quad \text{where } \Phi_i := \Phi_{\boldsymbol{u}_i, \varepsilon^2}.$$

*As the atoms $\Phi_i$ have disjoint supports, the squared $L^2(\mathcal{P}_X(\alpha))$ distance between any two distinct functions $f_\xi$ and $f_{\xi'}$ can be computed as:*

$$\|f_\xi - f_{\xi'}\|_{L^2(\mathcal{P}_X(\alpha))}^2 = \sum_{i=1}^N (\xi_i - \xi_i')^2 \|\Phi_i\|_{L^2(\mathcal{P}_X(\alpha))}^2 = 4\sum_{i:\xi_i \neq \xi_i'} \|\Phi_i\|_{L^2(\mathcal{P}_X(\alpha))}^2.$$

*Referring to the cap mass properties in Lemma H.1 (which implies $\|\Phi_i\|_{L^2(\mathcal{P}_X(\alpha))}^2 \asymp \varepsilon^{d-1+2\alpha}$), this simplifies to the final distance scaling*

$$\|f_\xi - f_{\xi'}\|_{L^2(\mathcal{P}_X(\alpha))}^2 \asymp_{d,\alpha} \varepsilon^{d-1+2\alpha} d_H(\xi, \xi'),$$

*where $d_H(\xi, \xi')$ is the Hamming distance.*

## H.2   PROOF OF THEOREM 3.5

A key step in our proof is to find a large number of caps that contain no data points from the dataset $\mathcal{D}$. In the standard fixed-sample-size setting, the number of points in each disjoint cap, say $Z_i := \#\{\boldsymbol{x}_j \in C(\boldsymbol{u}_i, \varepsilon)\}$, follows a multinomial distribution. The counts $(Z_1, \ldots, Z_N)$ are negatively correlated because their sum is fixed to $n$. This dependence complicates the analysis of finding many empty caps simultaneously.

To circumvent this difficulty, we employ **Poissonization**. We replace the fixed sample size $n$ with a random sample size $N_{\mathrm{poi}}$ drawn from a Poisson distribution with mean $n$. This means the occupancy counts $Z_i$ become independent Poisson random variables. This independence allows for the direct use of standard concentration inequalities like the Chernoff bound.

**Proposition H.5** (Abundance of Empty Caps under Poissonization). *Let $\{C(\boldsymbol{u}_i, \varepsilon)\}_{i=1}^N$ be the set of disjoint caps from Lemma H.3. Let the sample size be $N_{poi} \sim \mathrm{Poi}(n)$. Let $Z_i$ be the number of samples falling into cap $C(\boldsymbol{u}_i, \varepsilon)$. Define the expected number of points per cap as $\lambda := n \cdot \mathcal{P}_X(C(\boldsymbol{u}_1, \varepsilon))$. If we choose $\varepsilon$ such that $\lambda \asymp 1$, then there exists a constant $c > 0$ such that with probability at least $1 - \exp(-cN)$:*

$$\#\{i \in \{1, \ldots, N\} : Z_i = 0\} \ \geq \ \frac{1}{2} e^{-\lambda} N.$$

*Proof.* Under Poissonization, the random variables $Z_i = \#\{\boldsymbol{x}_j \in C(\boldsymbol{u}_i, \varepsilon)\}$ are independent Poisson variables with mean $\lambda_i = n \cdot \mathcal{P}_X(C(\boldsymbol{u}_i, \varepsilon))$. By Lemma H.2 and our choice of scale, $\lambda_i = \lambda \asymp 1$ for all $i$.

Let $Y_i = \mathbb{1}\{Z_i = 0\}$ be the indicator that the $i$-th cap is empty. The variables $Y_1, \ldots, Y_N$ are i.i.d. Bernoulli random variables. The probability of success (a cap being empty) is:

$$p := \mathbb{P}(Y_i = 1) = \mathbb{P}(Z_i = 0) = \frac{e^{-\lambda}\lambda^0}{0!} = e^{-\lambda}.$$

Since $\lambda \asymp 1$, $p$ is a positive constant. The expected number of empty caps is $\mathbb{E}[\sum Y_i] = Np = Ne^{-\lambda}$. By a standard Chernoff bound on the sum of i.i.d. Bernoulli variables, we have that for any $\delta \in (0, 1)$:

$$\mathbb{P}\left(\sum_{i=1}^N Y_i < (1 - \delta)Np\right) \leq \exp\left(-\frac{\delta^2 Np}{2}\right).$$

Choosing $\delta = 1/2$, we find that the number of empty caps is at least $\frac{1}{2}Np = \frac{1}{2}e^{-\lambda}N$ with probability at least $1 - \exp(-cN)$ for some constant $c > 0$. □

The condition $\lambda \asymp 1$ is central. It balances the sample size $n$ with the geometric scale $\varepsilon$. Using Lemma H.2, this balance is achieved when:

$$n \cdot \varepsilon^{d-1+2\alpha} \ \asymp \ 1 \quad \Longleftrightarrow \quad \varepsilon \ \asymp \ n^{-1/(d-1+2\alpha)}. \tag{82}$$

With this choice, Proposition H.5 guarantees that a constant fraction of the $N \asymp \varepsilon^{-(d-1)}$ caps are empty with overwhelmingly high probability. Informlly speaking, this hints appearance of the neural network with dedicated neurons, each of which has at most one activation point. This paradigm aligns with our construction stable/flat interpolation neural network discussed in Appendix I.

Armed with the guarantee of many empty caps, we can now construct our adversarial pair of functions, $f$ and $f'$. These functions will be designed to agree on all non-empty caps but disagree on a large number of empty caps. Since by definition no data lies in the empty caps, the functions will be identical on the training data. However, their disagreement on a substantial portion of the space will create a large gap in their population risks.

**Proposition H.6** (Indistinguishable yet Separated Pair). *Work under the scale choice $\varepsilon \asymp n^{-1/(d-1+2\alpha)}$ and on the high-probability event from Proposition H.5 where at least $\frac{1}{2}e^{-\lambda}N$ caps are empty. There exist two functions $f, f' \in \mathcal{F}$ from Construction H.4 such that*

1. ***Indistinguishability on Data:*** *$f(\boldsymbol{x}_j) = f'(\boldsymbol{x}_j)$ for all points $\boldsymbol{x}_j$ in the Poisson-drawn sample.*

2. ***Separation in Population:*** $\|f - f'\|^2_{L^2(\mathcal{P}_X(\alpha))} \asymp n^{-\frac{2\alpha}{d-1+2\alpha}}$.

*Proof.* Let $\mathcal{J} \subset \{1, \ldots, N\}$ be the set of indices corresponding to empty caps, with $|\mathcal{J}| \geq \frac{1}{2}e^{-\lambda}N \asymp N$. Construct two sign vectors $\xi, \xi' \in \{\pm 1\}^N$ as follows:

- For $i \in \mathcal{J}$, set $\xi_i = 1$ and $\xi'_i = -1$.
- For $i \notin \mathcal{J}$, set $\xi_i = \xi'_i = 1$.

Let $f = f_\xi$ and $f' = f_{\xi'}$.

1. **Indistinguishability:** The function difference is $f - f' = \sum_{i \in \mathcal{J}} 2\Phi_i$. The support of this difference is $\bigcup_{i \in \mathcal{J}} C(\boldsymbol{u}_i, \varepsilon)$. Since all caps indexed by $\mathcal{J}$ are empty, no data point $\boldsymbol{x}_j$ falls into this support. Thus, $(f - f')(\boldsymbol{x}_j) = 0$ for all $j$, which implies $f(\boldsymbol{x}_j) = f'(\boldsymbol{x}_j)$.

2. **Separation:** The Hamming distance is $d_H(\xi, \xi') = |\mathcal{J}| \asymp N$. Using the result from Construction H.4:

$$\|f - f'\|^2_{L^2(\mathcal{P}_X(\alpha))} \asymp \varepsilon^{d-1+2\alpha} \cdot d_H(\xi, \xi') \asymp \varepsilon^{d-1+2\alpha} \cdot N \asymp \varepsilon^{d-1+2\alpha} \cdot \varepsilon^{-(d-1)} = \varepsilon^{2\alpha}.$$

Substituting our choice of scale $\varepsilon \asymp n^{-1/(d-1+2\alpha)}$ yields the desired separation:

$$\|f - f'\|^2_{L^2(\mathcal{P}_X(\alpha))} \asymp \left(n^{-1/(d-1+2\alpha)}\right)^{2\alpha} = n^{-\frac{2\alpha}{d-1+2\alpha}}.$$

$\square$

The final step is to transfer the result from the Poissonized model back to the original fixed-sample-size model. This is justified by the strong concentration of the Poisson distribution around its mean.

**Lemma H.7** (De-Poissonization). *Let $N_{poi} \sim \mathrm{Poi}(n)$. For any $\eta \in (0, 1)$, $\mathbb{P}(N_{poi} \notin [(1-\eta)n, (1+\eta)n]) \leq 2\exp(-c_\eta n)$ for some constant $c_\eta > 0$. The conclusions of Proposition H.6 hold for a fixed sample size $n$.*

*Proof.* The existence of a large fraction of empty caps is an event that is monotone with respect to the sample size (fewer samples lead to more empty caps). The high-probability conclusion from Proposition H.5 holds for any sample size $k$ within the concentration interval $[(1-\eta)n, (1+\eta)n]$, as changing $n$ to $k$ only alters the key parameter $\lambda$ by a constant factor, which does not affect the asymptotic analysis. Since $N_{\mathrm{poi}}$ falls in this interval with probability $1 - o(1)$, the event of finding an indistinguishable pair also occurs with probability $1 - o(1)$ for a Poisson sample. This high-probability statement can be transferred back to the fixed-$n$ setting, yielding the same rate for the lower bound. $\square$

The existence of an indistinguishable pair allows us to establish a lower bound on the minimax risk for estimation in the noiseless setting. This intermediate result is the foundation for the final generalization gap bound.

Let $\mathcal{F}_{\mathrm{pack}}$ be the adversarial class defined in Construction H.4 with $\varepsilon$ defined in Proposition H.6.

**Corollary H.8** (Minimax Lower Bound). *In the noiseless setting where $y_i = f(\boldsymbol{x}_i)$, the minimax risk for any estimator $\hat{f}$ over the adversarial class $\mathcal{F}_{pack}$ is bounded below*

$$\inf_{\hat{f}} \sup_{f_0 \in \mathcal{F}_{pack}} \mathbb{E}\left[\|\hat{f} - f_0\|^2_{L^2(\mathcal{P}_X(\alpha))}\right] \gtrsim n^{-\frac{2\alpha}{d-1+2\alpha}}.$$

*Proof.* Let $E$ be the event that an indistinguishable pair $(f, f') \in \mathcal{F}_{\mathrm{pack}}$ exists for a fixed sample size $n$. From Proposition H.6 and Lemma H.7, we know that $\mathbb{P}(E) = 1 - o(1)$. On this event $E$, let the true function $f_0$ be chosen uniformly at random from $\{f, f'\}$.

Any estimator $\hat{f}$ receives the dataset $\mathcal{D}_n$ of size $n$. Since $f(\boldsymbol{x}_i) = f'(\boldsymbol{x}_i)$ for all $\boldsymbol{x}_i \in \mathcal{D}_n$, the generated data is identical whether $f_0 = f$ or $f_0 = f'$. The estimator thus has no information to

distinguish between $f$ and $f'$. The expected risk of any estimator, conditioned on the event $E$, can be lower-bounded

$$\mathbb{E}\left[\|\hat{f} - f_0\|^2 \Big| E\right] = \frac{1}{2}\|\hat{f} - f\|^2 + \frac{1}{2}\|\hat{f} - f'\|^2 \geq \frac{1}{4}\|f - f'\|^2,$$

where the inequality is a standard result for a choice between two points. The worst-case risk for an estimator over $f_0 \in \{f, f'\}$ is thus at least $\frac{1}{4}\|f - f'\|^2$.

Taking the expectation over the sampling of $D_n$:

$$\inf_{\hat{f}} \sup_{f_0 \in \mathcal{F}_{\text{pack}}} \mathbb{E}\left[\|\hat{f} - f_0\|^2\right] \geq \inf_{\hat{f}} \sup_{f_0 \in \mathcal{F}_{\text{pack}}} \mathbb{E}\left[\|\hat{f} - f_0\|^2 \Big| E\right] \mathbb{P}(E)$$

$$\geq \frac{1}{4}\mathbb{E}\left[\|f - f'\|^2 \big| E\right] \mathbb{P}(E).$$

Since on the event $E$, the separation $\|f - f'\|^2_{L^2(\mathcal{P}_X(\alpha))} \asymp n^{-\frac{2\alpha}{d-1+2\alpha}}$ and $\mathbb{P}(E) \to 1$ as $n \to \infty$, the result follows. $\qquad\square$

Finally, we connect the minimax risk lower bound to the generalization gap. The argument reduces the problem of bounding the generalization gap to the minimax estimation problem we just solved.

**Theorem H.9** (Generalization Gap Lower Bound). *Let $\mathcal{P}$ denote any joint distribution of $(\boldsymbol{x}, y)$ where the marginal distribution of $\boldsymbol{x}$ is $\mathcal{P}_X(\alpha)$) and $y$ is supported on $[-1, 1]$. Let $\mathcal{D}_n = \{(\boldsymbol{x}_j, y_j)\}_{j=1}^n$ be a dataset of $n$ i.i.d. samples from $\mathcal{P}$. Let $\widehat{R}_{\mathcal{D}_n}(f)$ be any empirical risk estimator for the true risk $R_{\mathcal{P}}(f) := \mathbb{E}_{(\boldsymbol{x}, y) \sim \mathcal{P}}[(f(\boldsymbol{x}) - y)^2]$. Then,*

$$\inf_{\widehat{R}} \sup_{\mathcal{P}} \mathbb{E}_{\mathcal{D}_n}\left[\sup_{f \in \mathcal{F}_g(\mathbb{B}_1^d; 1, 1)} \left|R_{\mathcal{P}}(f) - \widehat{R}_{\mathcal{D}_n}(f)\right|\right] \gtrsim_{d,\alpha} n^{-\frac{2\alpha}{d-1+2\alpha}}.$$

*Proof.* We lower-bound the supremum over all distributions $\mathcal{P}$ by restricting it to a worst-case family of deterministic distributions $\mathcal{P}_{f_0}$, where labels are given by $y = f_0(\boldsymbol{x})$ for some $f_0$ from our adversarial packing set, $\mathcal{F}_{\text{pack}}$. The proof proceeds via a chain of inequalities.

$$\inf_{\widehat{R}} \sup_{\mathcal{P}} \mathbb{E}\left[\sup_{f \in \mathcal{F}} \left|R_{\mathcal{P}}(f) - \widehat{R}_{\mathcal{D}_n}(f)\right|\right] \tag{83}$$

$$\geq \inf_{\widehat{R}} \sup_{f_0 \in \mathcal{F}_{\text{pack}}} \mathbb{E}\left[\sup_{f \in \mathcal{F}} \left|R_{\mathcal{P}_{f_0}}(f) - \widehat{R}_{\mathcal{D}_n}(f)\right|\right] \tag{84}$$

$$\geq \sup_{f_0 \in \mathcal{F}_{\text{pack}}} \frac{1}{2} \inf_{\widehat{R}} \mathbb{E}\left[R_{\mathcal{P}_{f_0}}(\hat{f}_{\text{ERM}}) - R_{\mathcal{P}_{f_0}}(f_0)\right] \tag{85}$$

$$\geq \frac{1}{2} \inf_{\hat{f}} \sup_{f_0 \in \mathcal{F}_{\text{pack}}} \mathbb{E}\left[R_{\mathcal{P}_{f_0}}(\hat{f}) - R_{\mathcal{P}_{f_0}}(f_0)\right] \tag{86}$$

$$= \frac{1}{2} \inf_{\hat{f}} \sup_{f_0 \in \mathcal{F}_{\text{pack}}} \mathbb{E}\left[\|\hat{f} - f_0\|^2_{L^2(\mathcal{P}_X(\alpha))}\right] \tag{87}$$

$$\text{Corollary H.8} \implies \gtrsim n^{-\frac{2\alpha}{d-1+2\alpha}} \tag{88}$$

The steps are justified as follows

- **Inequality (85):** This step uses a standard result relating the generalization gap to the excess risk of an Empirical Risk Minimizer (ERM), $\hat{f}_{\text{ERM}} := \arg\min_{f \in \mathcal{F}} \widehat{R}_{\mathcal{D}_n}(f)$. By definition, $\widehat{R}(\hat{f}_{\text{ERM}}) \leq \widehat{R}(f_0)$. This leads to the decomposition

$$R(\hat{f}_{\text{ERM}}) - R(f_0) = \left(R(\hat{f}_{\text{ERM}}) - \widehat{R}(\hat{f}_{\text{ERM}})\right) + \left(\widehat{R}(\hat{f}_{\text{ERM}}) - \widehat{R}(f_0)\right) + \left(\widehat{R}(f_0) - R(f_0)\right)$$

$$\leq 2 \sup_{f \in \mathcal{F}} |R(f) - \widehat{R}(f)|.$$

- **Inequation (86)** The infimum over all risk estimators $\widehat{R}$ (which induces a corresponding ERM) is lower-bounded by the infimum over all possible estimators $\hat{f}$ of the function $f_0$. This transitions the problem to the standard minimax framework.

- **Equation (87):** In this noiseless setting with a deterministic labeling function $f_0$, the population risk of $f_0$ is $R_{\mathcal{P}_{f_0}}(f_0) = 0$. The excess risk $R_{\mathcal{P}_{f_0}}(\hat{f})$ is precisely the squared $L_2$ distance $\|\hat{f} - f_0\|_{L^2(\mathcal{P}_X(\alpha))}^2$. The expression becomes the definition of the minimax risk over the class $\mathcal{F}_{\text{pack}}$.

This completes the proof. $\qquad\qquad\qquad\qquad\qquad\qquad\qquad\qquad\qquad\qquad\qquad\qquad\qquad\square$

## I  FLAT INTERPOLATING TWO-LAYER RELU NETWORKS ON THE UNIT SPHERE

Let $\{(\boldsymbol{x}_i, y_i)\}_{i=1}^n$ be a dataset with $\boldsymbol{x}_i \in \mathbb{S}^{d-1}$, $d > 1$, and pairwise distinct inputs. Assume labels are uniformly bounded, i.e., $|y_i| \le D$ for all $i$. Consider width-$K$ two-layer ReLU models

$$f_{\boldsymbol{\theta}}(\boldsymbol{x}) = \sum_{k=1}^{K} v_k \, \phi(\boldsymbol{w}_k^\mathsf{T} \boldsymbol{x} - b_k) + \beta. \tag{89}$$

**Theorem I.1** (Flat interpolation with width $\le n$). *Under the set-up above, there exists a width $K \le n$ network of the form (89) that interpolates the dataset and whose Hessian operator norm satisfies*

$$\lambda_{\max}\left(\nabla_{\boldsymbol{\theta}}^2 \mathcal{L}\right) \;\le\; 1 + \frac{D^2 + 2}{n}. \tag{90}$$

**Construction I.2** (Flat interpolation ReLU network). *Let $I_{\neq 0} := \{i : y_i \neq 0\}$ and set the width $K := |I_{\neq 0}| \le n$. For each $k \in I_{\neq 0}$ define*

$$\rho_k := \max_{k \neq i} \boldsymbol{x}_i^\mathsf{T} \boldsymbol{x}_k \;<\; 1, \qquad b_k \in (\rho_k, 1) \;\; (e.g., \, b_k = \frac{1 + \rho_k}{2}), \qquad \boldsymbol{w}_k := \boldsymbol{x}_k. \tag{91}$$

*Then for any sample index $i$,*

$$\boldsymbol{w}_k^\mathsf{T} \boldsymbol{x}_i - b_k = \begin{cases} 1 - b_k \;>\; 0, & i = k, \\ \le \rho_k - b_k \;<\; 0, & i \neq k, \end{cases} \tag{92}$$

*so the $k$-th unit activates on $\boldsymbol{x}_k$ and is inactive on all $\boldsymbol{x}_i$ with $i \neq k$. Set the output weight*

$$v_k := \frac{y_k}{1 - b_k}. \tag{93}$$

*By (92) and (93), the model interpolates on nonzero labels because $f(\boldsymbol{x}_k) = v_k(1 - b_k) = y_k$ for $k \in I_{\neq 0}$, and it also interpolates zero labels since all constructed units are inactive on $\boldsymbol{x}_i$ when $i \notin I_{\neq 0}$, hence $f(\boldsymbol{x}_i) = 0 = y_i$.*

*For each constructed unit, define*

$$\tilde{v}_k := \mathrm{sign}(v_k) \in \{\pm 1\}, \qquad \tilde{\boldsymbol{w}}_k := |v_k| \, \boldsymbol{w}_k, \qquad \tilde{b}_k := |v_k| \, b_k. \tag{94}$$

*Then for any input $\boldsymbol{x}$,*

$$\tilde{v}_k \, \phi(\tilde{\boldsymbol{w}}_k^\mathsf{T} \boldsymbol{x} - \tilde{b}_k) = \mathrm{sign}(v_k) \, \phi\big(|v_k|(\boldsymbol{w}_k^\mathsf{T} \boldsymbol{x} - b_k)\big) = v_k \, \phi(\boldsymbol{w}_k^\mathsf{T} \boldsymbol{x} - b_k), \tag{95}$$

*so interpolation is preserved. Moreover, the activation pattern on the dataset is unchanged because (92) has strict inequalities and $|v_k| > 0$. At $\boldsymbol{x}_i$ we have the (post-rescaling) pre-activation*

$$\tilde{z}_k := \tilde{\boldsymbol{w}}_k^\mathsf{T} \boldsymbol{x}_k - \tilde{b}_k = |v_k|\,(1 - b_k) = |y_k| \;>\; 0, \qquad |\tilde{v}_k| = 1. \tag{96}$$

*In what follows we work with the reparameterized network and drop tildes for readability, implicitly assuming $|v_k| = 1$ for all $k \in I_{\neq 0}$ and $z_k := \boldsymbol{w}_k^\mathsf{T} \boldsymbol{x}_k - b_k = |y_k|$.*

**Proposition I.3.** *Let $\boldsymbol{\theta}$ be the model in Construction I.2. Then*

$$\lambda_{\max}(\nabla_{\boldsymbol{\theta}}^2 \mathcal{L}) \leq 1 + \frac{D^2 + 2}{n}.$$

*Proof.* By direct computation, the Hessian $\nabla_{\boldsymbol{\theta}}^2 \mathcal{L}$ is given by

$$\nabla_{\boldsymbol{\theta}}^2 \mathcal{L} = \frac{1}{n} \sum_{i=1}^{n} \nabla_{\boldsymbol{\theta}} f(\boldsymbol{x}_i) \nabla_{\boldsymbol{\theta}} f(\boldsymbol{x}_i)^{\mathsf{T}} + \frac{1}{n} \sum_{i=1}^{n} (f(\boldsymbol{x}_i) - y_i) \nabla_{\boldsymbol{\theta}}^2 f(\boldsymbol{x}_i). \tag{97}$$

Since the model interpolates $f(\boldsymbol{x}_i) = y_i$ for all $i$, we have

$$\nabla_{\boldsymbol{\theta}}^2 \mathcal{L} = \frac{1}{n} \sum_{i=1}^{n} \nabla_{\boldsymbol{\theta}} f(\boldsymbol{x}_i) \nabla_{\boldsymbol{\theta}} f(\boldsymbol{x}_i)^{\mathsf{T}}. \tag{98}$$

Denote the tangent features matrix by

$$\boldsymbol{\Phi} = \left[ \nabla_{\boldsymbol{\theta}} f(\boldsymbol{x}_1), \nabla_{\boldsymbol{\theta}} f(\boldsymbol{x}_2), \cdots, \nabla_{\boldsymbol{\theta}} f(\boldsymbol{x}_n) \right]. \tag{99}$$

Then $\nabla_{\boldsymbol{\theta}}^2 \mathcal{L}$ in (98) can be expressed by $\nabla_{\boldsymbol{\theta}}^2 \mathcal{L} = \boldsymbol{\Phi}\boldsymbol{\Phi}^{\mathsf{T}}/n$, and the operator norm is computed by

$$\lambda_{\max}(\nabla_{\boldsymbol{\theta}}^2 \mathcal{L}) = \max_{\boldsymbol{\gamma} \in \mathbb{S}^{(d+2)K}} \frac{1}{n} \|\boldsymbol{\Phi}^{\mathsf{T}} \boldsymbol{\gamma}\|^2 = \max_{\boldsymbol{u} \in \mathbb{S}^{n-1}} \frac{1}{n} \|\boldsymbol{\Phi} \boldsymbol{u}\|^2 \tag{100}$$

From direct computation we obtain

$$\nabla_{\boldsymbol{\theta}} f(\boldsymbol{x}) = \begin{pmatrix} \nabla_{\boldsymbol{W}}(f) \\ \nabla_{\boldsymbol{b}}(f) \\ \nabla_{\boldsymbol{\omega}}(f) \\ \nabla_{\beta}(f) \end{pmatrix} \tag{101}$$

For the parameters $[\boldsymbol{w}_k, b_k, v_k]$ associated to the neuron of index $j$,

$$\frac{\partial f(\boldsymbol{x})}{\partial v_k} = \mathbb{1}\{\boldsymbol{w}_k^{\mathsf{T}} \boldsymbol{x} > b_k\} \left( \boldsymbol{w}_k^{\mathsf{T}} \boldsymbol{x} - b_k \right), \qquad \frac{\partial f(\boldsymbol{x}_i)}{\partial \boldsymbol{w}_k} = \mathbb{1}\{\boldsymbol{w}_k^{\mathsf{T}} \boldsymbol{x} > b_k\} \, v_k \, \boldsymbol{x},$$

$$\frac{\partial f(\boldsymbol{x}_i)}{\partial b_k} = \mathbb{1}\{\boldsymbol{w}_k^{\mathsf{T}} \boldsymbol{x} > b_k\} \, - v_k, \qquad \frac{\partial f(\boldsymbol{x}_i)}{\partial \beta} = 1.$$

By the one-to-one activation property (92), each sample $\boldsymbol{x}_i$ activates exactly one unit (the unit with the same index $k$ when $k \in I_{\neq 0}$), and activates none when $i \notin I_{\neq 0}$. Hence the sample-wise gradient $\nabla_{\boldsymbol{\theta}} f(\boldsymbol{x}_k)$ has support only on the parameter triplet $(\boldsymbol{w}_k, b_k, v_k, \beta)$ for $k \in I_{\neq 0}$, and is zero for other parameters. Writing the nonzero gradient block explicitly (recall $|v_k| = 1$),

$$\nabla_{(\boldsymbol{w}_k, b_k, v_k, \beta)} f_{\boldsymbol{\theta}}(\boldsymbol{x}_k) = \begin{pmatrix} \nabla_{(\boldsymbol{w}_k, b_k, v_k)} f_{\boldsymbol{\theta}} \\ 1 \end{pmatrix},$$

$$\nabla_{(\boldsymbol{w}_k, b_k, v_k)} f_{\boldsymbol{\theta}}(\boldsymbol{x}_k) = \begin{cases} \begin{pmatrix} v_k \boldsymbol{x}_k \\ -v_k \\ y_k \end{pmatrix}, & (k \in I_{\neq 0}), \\ \boldsymbol{0}, & (k \notin I_{\neq 0}), \end{cases} \tag{102}$$

After row permutation and subsistion by (102), (100) is of the form

$$
\mathbf{\Phi} =
\begin{pmatrix}
\nabla_{(\boldsymbol{w}_1,b_1,v_1)} f_{\boldsymbol{\theta}}(\boldsymbol{x}_1) & \mathbf{0} & \cdots & \mathbf{0} \\
\mathbf{0} & \nabla_{(\boldsymbol{w}_2,b_2,v_2)} f_{\boldsymbol{\theta}}(\boldsymbol{x}_2) & \cdots & \vdots \\
\mathbf{0} & \mathbf{0} & \cdots & \vdots \\
\vdots & \vdots & \cdots & \mathbf{0} \\
\mathbf{0} & \mathbf{0} & \cdots & \nabla_{(\boldsymbol{w}_n,b_n,v_n)} f_{\boldsymbol{\theta}}(\boldsymbol{x}_n) \\
1 & 1 & \cdots & 1
\end{pmatrix}
\tag{103}
$$

$$
=
\begin{pmatrix}
\begin{pmatrix} v_1\,\boldsymbol{x}_1 \\ v_1 \\ y_1 \end{pmatrix} & \mathbf{0} & \cdots & \mathbf{0} \\
\mathbf{0} & \begin{pmatrix} v_2\,\boldsymbol{x}_2 \\ v_2 \\ y_2 \end{pmatrix} & \cdots & \vdots \\
\mathbf{0} & \mathbf{0} & \cdots & \vdots \\
\vdots & \vdots & \cdots & \mathbf{0} \\
\mathbf{0} & \mathbf{0} & \cdots & \begin{pmatrix} v_n\,\boldsymbol{x}_n \\ v_n \\ y_n \end{pmatrix} \\
1 & 1 & \cdots & 1
\end{pmatrix}.
\tag{104}
$$

Let $\boldsymbol{u} = (u_1, \cdots, u_n) \in \mathbb{S}^{n-1}$ and plug (104) in (100) to have

$$
\lambda_{\max}(\nabla_{\boldsymbol{\theta}}^2 \mathcal{L}) = \max_{\boldsymbol{u} \in \mathbb{S}^{n-1}} \frac{1}{n} \|\mathbf{\Phi}\boldsymbol{u}\|^2
\tag{105}
$$

$$
= \frac{1}{n} \max_{\boldsymbol{u} \in \mathbb{S}^{n-1}} \left\| \begin{pmatrix} u_1 \nabla_{(\boldsymbol{w}_1,b_1,v_1)} f_{\boldsymbol{\theta}}(\boldsymbol{x}_1) \\ u_2 \nabla_{(\boldsymbol{w}_2,b_2,v_2)} f_{\boldsymbol{\theta}}(\boldsymbol{x}_2) \\ \vdots \\ u_n \nabla_{(\boldsymbol{w}_n,b_n,v_n)} f_{\boldsymbol{\theta}}(\boldsymbol{x}_n) \\ \sum_{i=1}^n u_i \end{pmatrix} \right\|_2^2
$$

$$
= \frac{1}{n} \max_{\boldsymbol{u} \in \mathbb{S}^{n-1}} \sum_{i=1}^n u_i^2 \left\| \nabla_{(\boldsymbol{w}_i,b_i,v_i)} f_{\boldsymbol{\theta}}(\boldsymbol{x}_i) \right\|_2^2 + \left( \sum_{i=1}^n u_i \right)^2
\tag{106}
$$

$$
= \frac{1}{n} \max_{\boldsymbol{u} \in \mathbb{S}^{n-1}} \sum_{i=1}^n u_i^2 \left( \|\boldsymbol{x}_i\|_2^2 + 1 + y_i^2 \right) + \left( \sum_{i=1}^n u_i \right)^2
\tag{107}
$$

$$
\leq \frac{1}{n} \left( \max_{i \in [n]} \left( \|\boldsymbol{x}_i\|_2^2 + 1 + y_i^2 \right) + \max_{\boldsymbol{u} \in \mathbb{S}^{n-1}} \left( \sum_{i=1}^n u_i \right)^2 \right)
\tag{108}
$$

$$
\leq \frac{1}{n} \left( D^2 + 2 + n \right) = 1 + \frac{D^2 + 2}{n}
$$

If we remove the output bias term $\beta$ from the parameters, then the bottom row of 104 will be remove and thus term $\sum_i u_i$ in (106) will be removed. $\qquad \square$

## J   TECHNICAL LEMMAS

**Lemma J.1** (Concentration of a Poisson Random Variable). *Let $N_{poi} \sim \mathrm{Poi}(n)$ be a Poisson random variable with mean $n$. Then for any $\eta \in (0,1)$,*

$$
\mathbb{P}\left( |N_{poi} - n| \geq \eta n \right) \leq 2 \exp\left( -\frac{\eta^2 n}{3} \right).
$$

*Proof.* The proof employs the Chernoff bounding method. The Moment Generating Function (MGF) of $N_{\text{poi}} \sim \text{Poi}(n)$ is given by:

$$\mathbb{E}\left[e^{tN_{\text{poi}}}\right] = e^{n(e^t - 1)}.$$

We will bound the upper and lower tails separately.

We want to bound $\mathbb{P}(N_{\text{poi}} \geq (1 + \eta)n)$. For any $t > 0$, Markov's inequality implies:

$$\begin{aligned}
\mathbb{P}(N_{\text{poi}} \geq (1 + \eta)n) &= \mathbb{P}\left(e^{tN_{\text{poi}}} \geq e^{t(1+\eta)n}\right) \\
&\leq \frac{\mathbb{E}\left[e^{tN_{\text{poi}}}\right]}{e^{t(1+\eta)n}} \\
&= \frac{e^{n(e^t - 1)}}{e^{t(1+\eta)n}} = \exp\left(n(e^t - 1) - tn(1 + \eta)\right).
\end{aligned}$$

To obtain the tightest bound, we minimize the exponent with respect to $t$. The optimal $t$ is found by setting the derivative to zero, which yields $e^t = 1 + \eta$, or $t = \ln(1 + \eta)$. Substituting this value back into the bound gives:

$$\mathbb{P}(N_{\text{poi}} \geq (1+\eta)n) \leq \exp\left(n((1 + \eta) - 1) - n(1 + \eta)\ln(1 + \eta)\right) = \exp\left(n[\eta - (1 + \eta)\ln(1 + \eta)]\right).$$

We now use the standard inequality: $\ln(1 + x) \geq x - \frac{x^2}{2}$ for $x \geq 0$. A more specific inequality for this context is $\eta - (1 + \eta)\ln(1 + \eta) \leq -\frac{\eta^2}{2(1+\eta/3)}$. For $\eta \in (0, 1]$, this further simplifies. A widely used bound derived from this expression is:

$$\exp\left(n[\eta - (1 + \eta)\ln(1 + \eta)]\right) \leq \exp\left(-\frac{\eta^2 n}{3}\right).$$

Next, we bound $\mathbb{P}(N_{\text{poi}} \leq (1 - \eta)n)$. For any $t > 0$, we have:

$$\begin{aligned}
\mathbb{P}(N_{\text{poi}} \leq (1 - \eta)n) &= \mathbb{P}\left(e^{-tN_{\text{poi}}} \geq e^{-t(1-\eta)n}\right) \\
&\leq \frac{\mathbb{E}\left[e^{-tN_{\text{poi}}}\right]}{e^{-t(1-\eta)n}} \\
&= \frac{e^{n(e^{-t} - 1)}}{e^{-t(1-\eta)n}} = \exp\left(n(e^{-t} - 1) + tn(1 - \eta)\right).
\end{aligned}$$

The optimal $t$ is found by setting $e^{-t} = 1 - \eta$, or $t = -\ln(1 - \eta)$. Substituting this value gives:

$$\mathbb{P}(N_{\text{poi}} \leq (1-\eta)n) \leq \exp\left(n((1 - \eta) - 1) - n(1 - \eta)\ln(1 - \eta)\right) = \exp\left(n[-\eta - (1 - \eta)\ln(1 - \eta)]\right).$$

Using the inequality $-\eta - (1 - \eta)\ln(1 - \eta) \leq -\frac{\eta^2}{2}$ for $\eta \in (0, 1)$, we get a simple bound:

$$\exp\left(n[-\eta - (1 - \eta)\ln(1 - \eta)]\right) \leq \exp\left(-\frac{\eta^2 n}{2}\right).$$

Since for $\eta \in (0, 1)$, we have $\exp(-\eta^2 n/2) \leq \exp(-\eta^2 n/3)$, the lower tail is also bounded by $\exp(-\eta^2 n/3)$.

Using the union bound, we combine the probabilities for the two tails:

$$\begin{aligned}
\mathbb{P}\left(|N_{\text{poi}} - n| \geq \eta n\right) &= \mathbb{P}(N_{\text{poi}} \geq (1 + \eta)n) + \mathbb{P}(N_{\text{poi}} \leq (1 - \eta)n) \\
&\leq \exp\left(-\frac{\eta^2 n}{3}\right) + \exp\left(-\frac{\eta^2 n}{2}\right) \\
&\leq 2\exp\left(-\frac{\eta^2 n}{3}\right).
\end{aligned}$$

This completes the proof. $\qquad\square$

