# OpenReview forum: "Generalization Below the Edge of Stability: The Role of Data Geometry"
_ICLR.cc/2026/Conference — ICLR 2026 Poster_

### Official Review · Reviewer_9RWh · 2025-10-20

**Soundness:** 3
**Presentation:** 1
**Contribution:** 2
**Rating:** 4
**Confidence:** 3

**Summary:**

This paper proposes a novel prospective on how data geometry influences the implicit bias of overparameterized two-layer ReLU networks, with a number of theoretical results demonstrating that solutions below the edge of stability have various generalizability properties that can be desirable or undesirable depending on the setting. They provably show that below the edge of stability solutions can adapt to intrinsic lower dimension subspaces within an ambient space and that a spectrum of generalizability occurs determined by the implicit regularization induced by the "shatterability" of the data. Further empirical results are given to validate their theoretical claims and proof techniques.

**Strengths:**

- The paper provides a novel perspective on how the final solutions of the edge of the stability regime will generalize, while avoiding the dynamics of the rich regime. This formulation allows for intriguing new insights into how the structure of the data affects the implicit bias within the training of these models.
- This work has interesting implications for a number of different areas as outlined in appendix B and helps foster further work into the emerging direction of data shatterability.
- The empirical results are presented nicely with subsection 4.2 adding substantive support to practically verifying the effect of theorems on the form of the representations learnt in two-layer ReLU networks.
- The discussion and further work section is well written and suggests good follow-up directions on the topic.

**Weaknesses:**

- Data shatterability is not concretely defined or explained as a concept in the paper. While it can be roughly gleaned from previous work, a clear and concrete definition or explanation of the concept within the paper would substantially improve it.
- Definition 2.1 claims to be for "Isotropic Beta-radial distributions" but then proceeds to define "Isotropic alpha-powered distributions". It is unclear what isotropic beta-radial distributions are in this work as I do not believe they are defined.
- The theoretical claims are stated, but not much intuition or interpretation is given other than the overall message stated in the abstract and the introduction. A more fleshed out narrative and explanation between the theorems would have helped a deeper understanding of the work presented.
- The empirical results in subsection 4.1 could be improved by giving a more complete explanation of how to interpret them.
- For the right panel in figure 3 it is unclear why the correlation coefficient is given when the magnitude of the coefficient is so small. Additionally, the current figure provides no sense of how many of the points are in the bottom left corner.
- There are some linguistic issues such as: the first sentence in the "Disclaimers and Limitations" does not make sense, "deffered" on page 4.

**Questions:**

- Is it possible to run experiments on other architectures ideally to hypothesise how one could extend these results beyond the two-layer ReLU networks?
- Could further experiments be conducted to help elucidate the potential benefit of batch normalization through data shatterability?

---

> ### Author Response · Authors · 2025-11-21
> **Authors' Response to Reviewer 9RWh**
>
> - **Re :"Data shatterability is not concretely defined or explained as a concept in the paper. While it can be roughly gleaned from previous work, a clear and concrete definition or explanation of the concept within the paper would substantially improve it"**.
>
>     We thank the reviewer for highlighting the need for a clearer and more concrete definition of data shatterability. In the revised manuscript, we now introduce a fully formal treatment of this concept. As summarized in our General Response, we define shatterability through (i) pointwise half-space (Tukey) depth, which quantifies how strongly BEoS implicit regularization applies at each data point, and (ii) a distribution-level concentration index $S_{DQ}(P_X)$ (Definition 3.8), constructed from the depth–quantile function to measure how much probability mass lies in weakly regularized (shallow) regions.
>
>     We have also revised Sections 3.1–3.3 and the introduction to integrate this formal definition into the overall narrative, clarify the geometric interpretation, and connect it explicitly to our upper and lower bounds. We hope these changes make the notion of data shatterability substantially clearer within the paper itself.
>
> - **Re: "Definition 2.1 claims to be for "Isotropic Beta-radial distributions" but then proceeds to define "Isotropic alpha-powered distributions". It is unclear what isotropic beta-radial distributions are in this work as I do not believe they are defined."**
>
>     Thanks for pointing out this minor. We have named them clearly “isotropic Beta(\alpha)-radial distribution” in the revised version.
>
> - **Re:"The theoretical claims are stated, but not much intuition or interpretation is given other than the overall message stated in the abstract and the introduction. A more fleshed out narrative and explanation between the theorems would have helped a deeper understanding of the work presented."**
>
>     We completely agree that the narrative needed strengthening. As detailed in our General Response, we have restructured Section 3 to follow a deductive flow: starting from the tractable Isotropic case to derive the $S_{DQ}$ metric, then generalizing to Mixture Models. We also added Figures 1 & 2 to visually bridge the gap between the theorems and geometric intuition.
>
>
> - **Re: "Is it possible to run experiments on other architectures ideally to hypothesise how one could extend these results beyond the two-layer ReLU networks? Could further experiments be conducted to help elucidate the potential benefit of batch normalization through data shatterability?"**
>
>     We thank the reviewer for these suggestions and for their interest in extending the framework beyond the two-layer ReLU setting. In this work we deliberately focus on two-layer networks so that the empirical behavior can be compared directly to the quantities we can control analytically (BEoS stability, depth profile, and the shatterability index). A systematic treatment of deeper or more complex architectures would require new theory for how stability-induced regularization propagates across layers.
>
>     That said, preliminary experiments on a simple convolutional architecture (single-layer CNN with local receptive fields, parameter sharing, and global average pooling on spherical data with additive label noise) show behavior consistent with our shatterability perspective: the training loss decreases to the noise level and then stops improving, while the excess risk remains small, indicating that local connectivity and parameter sharing severely limit the network’s ability to shatter the data. These observations suggest that analogous geometry–shatterability phenomena may extend beyond fully connected two-layer ReLU networks, but a careful theoretical and empirical study of these architectures lies beyond the scope of the present paper.
>
>     Similarly, while we agree that studying batch normalization through the lens of data shatterability would be very interesting, our current analysis does not explicitly model batch normalization or its effect on the BEoS stability region and induced norms. Designing experiments and theory that isolate the contribution of batch normalization within this framework would require a dedicated investigation, and we regard this as a promising avenue for future work.

---

> > ### Author Response · Authors · 2025-11-28
> > **Authors' Response to Reviewer 9RWh (continual)**
> >
> > - **Re: "The empirical results in subsection 4.1 could be improved by giving a more complete explanation of how to interpret them."**
> >
> > 	We thank the reviewer for this helpful suggestion. We agree that the interpretation of the empirical results in Subsection 4.1 can be made clearer, and we have revised the text accordingly. In the new version(v3), we explicitly explain that the log–log slope of the clean MSE versus sample size is used as an empirical estimate of the rate exponent in our bounds: the fitted slope approximates (-c), so steeper negative slopes correspond to faster decay of the excess risk (better generalization), while flatter slopes indicate slower decay and weaker generalization. We also now state directly that this is the intended way to read the curves in Figure 3 (current version). In addition, we have added an appendix subsection (Appendix C.1 in v3) that derives the relationship between the clean MSE, the excess risk, and the noisy risk under our label-noise model, thereby making the connection between the theoretical prediction $\mathrm{Error} \lesssim n^{-c}$ and the empirical slopes fully explicit.

---

### Official Review · Reviewer_M57z · 2025-10-27

**Soundness:** 3
**Presentation:** 3
**Contribution:** 3
**Rating:** 8
**Confidence:** 4

**Summary:**

This paper studies how data geometry (especially intrinsic dimension and shatterablity) determines generalization under the edge of stability implicit bias.
The paper shows that (1) for features from mixtures of low-dimensional balls, such bias provably drives two-layer ReLU networks's generalization to be controlled by the intrinsic instead of the ambient dimension; (2) for isotropic data, its concentration toward boundary / shatterablity controls the generalization.
Such theoretical predictions are verified by experiments.
In a finer-grained level, in both proofs and empirical results, the implicit path norm regularization induced by EoS mainly regularizes the harder-to-shatter samples, improving generalization on them, and ignores the easier-to-shatter samples. As a result, data geometry of less shatterablity improves generalization.

**Strengths:**

- The paper proves the adaptation of low-dimensionality and the spectrum of generalization wrt to data concentration under EoS implicit bias. The latter is also equipped with a lower-bound to show tightness.
- The results are verified by empirical results.
- The sketch of proof is clearly discussed. Combined with the empirical results, it also clarifies on finer-grained roles of EoS implicit regularization and clarifies why and how data geometry affects controls generalization under EoS. Such results emphasize the data-dependent nature of EoS regularization and provides explanations on the highly data-dependent behaviour of overparameterized neural networks.
- The two main theoretical results and empirical results point to a promising shatterablity principle.

**Weaknesses:**

- In Sec 4.1, the experiments use label noise instead of I.I.D. sampling to construct training set and measures MSE losses instead of difference between the empirical and population risks. Such choice makes it more difficult to compare with theoretical results. What is the motivation for such choices?
- Minor:
  - Line 289: "g is the population version of the *weighted*."

**Questions:**

- In Figure 1a, the slope of theoretical prediction is not marked and compared to the actual slope. They seem to be actual≈-0.9 vs theoretical=-1/6, where a gap is still observed. What is the source of this gap? Does it come from looseness of bounds or that the experimented problem is not the worst case to reach the upperbound (eg, the directions of the $J$ lines are not worst-case)? Or is it beyond the (B)EoS bias, similar to Sec 3.3, and is governed by some bias else? Can this question be answered by some experiments, eg, searching the worst BEoS models and comparing their slopes with the actual and the theoretical? Maybe this question demands too much efforts. But I would greatly appreciate it because it may offer a clear view on the limit as well as relation of EoS biases with other biases.
- The discussion in Sec 4.3 seems quite generic. Is it possible to develop more general generalization bounds for BEoS weights assuming shatterablity instead of specific assumptions like low-dimensionality that leads shatterablity.

---

> ### Author Response · Authors · 2025-11-21
> **Authors’ Response to Reviewer M57z**
>
> - **Re:"In Sec 4.1, the experiments use label noise instead of I.I.D. sampling to construct training set and measures MSE losses instead of difference between the empirical and population risks. Such choice makes it more difficult to compare with theoretical results. What is the motivation for such choices?"**
>
>    We did consider the standard iid setting in the theoretical results.   In our theoretical setup (Assumptions 3.3 and 3.10), the data are drawn from a joint distribution $\mathcal{P}(x,y)$ with an arbitrary conditional law $\mathcal{P}(y\mid x)$. The experimental construction
>     $$
>     Y = f^{\*}(X) + \xi,\qquad \xi \sim \mathcal{N}(0,\sigma^2)
>     $$
>     is exactly an instance of this model, corresponding to a Gaussian conditional distribution centered at \(f^\*(x)\).
>
>     The additive label noise is introduced so that we can meaningfully measure the **generalization gap**. For squared loss,
>     $$
>     R\_{\mathrm{noisy}}(f)= \mathbb{E}\big[(f(X)-Y)^2\big]
>     =\underbrace{\mathbb{E}\big[(f(X)-f^\*(X))^2\big]}\_{\text{excess risk}}+ \underbrace{\sigma^2}\_{\text{noise level}},
>     $$
>     while the empirical training loss $\widehat{R}\_{\mathrm{train}}(f)$ uses the noisy labels. The “clean’’ empirical risk we report in the experiments,
>     $$
>     \widehat{R}\_{\mathrm{clean}}= \frac{1}{n}\sum_{i=1}^n \big(f(X_i)-f^\*(X_i)\big)^2,
>     $$
>     is simply a Monte Carlo estimate of the excess risk. Thus the generalization gap can be written as
>     $$
>     \mathrm{GenGap}= R\_{\mathrm{noisy}}(f) - \widehat{R}\_{\mathrm{train}}(f)
>     = \widehat{R}\_{\mathrm{clean}} + \sigma^2 - \widehat{R}\_{\mathrm{train}}(f).
>     $$
>
>     In other words, our experimental protocol is fully consistent with the theoretical assumption $(X,Y)\sim\mathcal{P}$, and it allows us to directly observe how the excess risk (the part controlled by our theory) contributes to the generalization gap. On geometries where the network cannot easily fit the injected noise (“hard-to-shatter’’ settings), the excess risk remains small, leading to a small generalization gap, in line with the theoretical predictions.
>
> - **Re: "In Figure 1a, the slope of theoretical prediction is not marked and compared to the actual slope. They seem to be actual≈-0.9 vs theoretical=-1/6, where a gap is still observed. What is the source of this gap? Does it come from looseness of bounds or that the experimented problem is not the worst case to reach the upperbound (eg, the directions of the lines are not worst-case)? Or is it beyond the (B)EoS bias, similar to Sec 3.3, and is governed by some bias else? Can this question be answered by some experiments, eg, searching the worst BEoS models and comparing their slopes with the actual and the theoretical? Maybe this question demands too much efforts. But I would greatly appreciate it because it may offer a clear view on the limit as well as relation of EoS biases with other biases."**
>
>     This gap stems primarily from the conservative nature of our theoretical analysis techniques. Our upper bounds rely on covering number arguments over the entire function class satisfying the BEoS constraint. This approach effectively bounds the *worst-case* generalization error among *all* possible stable solutions. Furthermore, as discussed in **Remark 3.9** of the revised manuscript, our current proof technique approximates the area under the depth-quantile curve using a single inscribed rectangle (optimizing the depth threshold $T$), which inherently discards some regularization benefits and loosens the bound. While exhaustively searching for the "worst-case" BEoS model is computationally intractable, the most critical validation in Figure 1a is that the empirical slopes remain constant across varying ambient dimensions ($d$), confirming our core theoretical prediction that the rate adapts to the intrinsic dimension.

---

> > ### Author Response · Authors · 2025-11-21
> > **Authors’ Response to Reviewer M57z (continual)**
> >
> > - **Re: "The discussion in Sec 4.3 seems quite generic. Is it possible to develop more general generalization bounds for BEoS weights assuming shatterablity instead of specific assumptions like low-dimensionality that leads shatterablity."**
> >
> >     We thank the reviewer for this insightful question. The goal of Sec. 4.3 is to use experiments to probe the shatterability–BEoS picture beyond the specific geometric settings where we currently have rigorous bounds, and to illustrate qualitatively what a more abstract “shatterability-based’’ theory might look like, rather than to claim new general theorems.
> >
> >     This question points exactly to the fundamental challenge of turning the shatterability principle into a fully abstract generalization theory, analogous in spirit to VC dimension. At a high level, our framework “inverts’’ the classical VC perspective: VC dimension measures a model’s ability to shatter *arbitrary* data, whereas our notion of “data shatterability’’ quantifies the *feasibility* of shattering a **specific** dataset under GD with BEoS-induced implicit regularization. In the revision, we formalize this via half-space depth and the concentration index $S_{DQ}(P_X)$, which summarize how strongly BEoS regularization acts across the input space.
> >
> >     The difficulty in obtaining a VC-style abstract bound, however, goes beyond defining a depth-based scalar index. Shatterability under BEoS depends not only on the depth profile but also on the **geometric arrangement** of the data and the **available partitioning directions**. This becomes evident in anisotropic or low-dimensional settings: as discussed at the beginning of Section 3.3 and illustrated in Figure 2, low intrinsic dimension collapses high-dimensional spherical caps into simple intervals or knots, drastically reducing the number of disjoint regions that ReLU hyperplanes can form, even when the ambient depth profile (and $S_{DQ}(P_X)$) looks worse than in an isotropic Beta–radial distribution. In such cases, $S_{DQ}(P_X)$ serves only as a *conservative* estimate of difficulty, and additional structural information is required.
> >
> >     A fully abstract theory would therefore require new tools that integrate half-space depth with the intrinsic geometry of the support. Developing such a framework appears technically challenging but is a natural and compelling direction for future work. We have added discussion in the revised manuscript (Section 3.3, Section 4.3, and the concluding remarks) to make these limitations and the associated roadmap explicit, and we are grateful to the reviewer for highlighting this perspective.

---

### Official Review · Reviewer_PDwN · 2025-10-31

**Soundness:** 2
**Presentation:** 2
**Contribution:** 2
**Rating:** 4
**Confidence:** 5

**Summary:**

This paper investigates the role of data geometry in shaping the generalization behavior of overparameterized two-layer ReLU networks trained below the edge of stability (BEoS). Building on recent studies on the implicit bias of gradient descent, the authors propose a unifying principle termed data shatterability, which measures how easily data geometry allows ReLU thresholds to separate samples. The main theoretical contributions include:
(1) A generalization bound for data supported on a mixture of low-dimensional subspaces, showing adaptation to the intrinsic dimension (Theorem 3.2);
(2) A family of generalization bounds for isotropic distributions parameterized by a concentration parameter α (Theorem 3.5), together with lower bounds (Theorem 3.6) and a constructive example demonstrating perfect interpolation on the sphere (Theorem 3.7).
Empirical experiments on synthetic data and MNIST illustrate how data geometry affects generalization and representation structure.

**Strengths:**

1. The motivation is clear and well-grounded in contemporary discussions around implicit regularization and the edge-of-stability regime.
2. The notion of data shatterability offers an elegant conceptual synthesis that connects data geometry, implicit bias, and generalization.

**Weaknesses:**

1. Definition and formalization of “data shatterability.”
While the paper emphasizes shatterability as the central concept, it is not clearly or formally defined in a mathematical sense. The text gives intuitive descriptions (“harder to shatter data generalizes better”), but the precise operational definition is vague.

  (1) Can the authors introduce a rigorous definition or metric of shatterability, perhaps analogous to VC dimension or some geometric measure of separability?

  (2) Why do the authors choose the beta-radial distributions with a parameter $\alpha$ to characterize this data property?

  (3) There is a gap between the rates in Theorem 3.5 and 3.6. How are they related to the general claim?

  (4) Would a toy mode concretely showing how different data geometries affect generalization make the concept more intuitive?

2. Clarity and logical structure of results.
The theoretical results (Theorems 3.2–3.7) are presented in isolation, and their interconnections are not fully clear. The reader may struggle to see how they jointly establish a unified principle.

(1) How do the results for subspace mixtures and isotropic distributions fit into a single theoretical framework?

(2) Is there an overarching theorem or lemma that ties them together through the concept of shatterability?

(3) A high-level diagram or summary of theoretical dependencies would help to improve readability and logical coherence.

3. Lack of dynamics analysis.
The paper claims to study generalization “below the edge of stability,” yet the analysis focuses entirely on static properties of stable minima rather than on the gradient descent dynamics that give rise to them.

(1) Without examining the time evolution of GD (e.g., curvature oscillations, stability trajectories), the results seem closer to a stability condition rather than a genuine characterization of the EoS regime.

(2) The current framework could be better described as a stability-based generalization bound rather than an analysis of edge-of-stability generalization.

4. Relation to prior work.
Theorems 3.2 and 3.5 resemble results in [Wu & Su, 2023] and related stability-based analyses, with the main difference being the explicit dependence on data geometry. However, the paper does not clearly articulate the essential technical innovation over these works.

(1) What are the key mathematical difficulties introduced by considering non-isotropic or low-dimensional data distributions, and how are they overcome here?

(2) A more explicit comparison or ablation (possibly in the appendix) would strengthen the contribution.

**Questions:**

See weaknesses.

---

> ### Author Response · Authors · 2025-11-21
> **Authors' Response to Reviewer PDwN**
>
> We sincerely thank the reviewer for the careful and insightful comments. We have performed a substantial revision to formalize our core concepts and clarify the logical structure of the results. Below we give a concise summary of how we addressed your main concerns; detailed technical explanations are provided in our General Response and the revised manuscript.
>
> - **Shatterability and logical structure (Weakness 1 \& 2)**
>
>     As explained in our General Response and summarized in the "Summary of Key Revisions" official comment, we have
>     - **Formally defined “data shatterability”** via half-space (Tukey) depth and a distribution-level index $S_{DQ}(P_X)$ (Definition 3.8), constructed from the depth–quantile function. This provides a rigorous geometric proxy for how much of the distribution lies in weakly regularized (shallow) regions under BEoS.
>     - **Reorganized Section 3** so that:
>         - The isotropic Beta–radial family is treated first as a clean setting where $S_{DQ}(P_X)$ directly controls the generalization spectrum (Theorems 3.4–3.6).
>         - The formal proxy $S_{DQ}(P_X)$ is introduced and interpreted as the unifying quantity.
>         - Mixtures / anisotropic cases (Theorem 3.10) are then presented as applications of the same BEoS–shatterability mechanism in more structured geometries.
>     - **Added Figures 1 and 2** to visualize isotropic vs. anisotropic shatterability and to make the geometric intuition more concrete.
>
>     We refer the reviewer to the General Response (and revised Sections 3.1–3.3) for the detailed formalization and unified perspective, and only expand below on the points that go beyond that discussion: gradient dynamics, the relation to Wu & Su, and the technical challenges in anisotropic settings.
>
> - **The technical difficulty in the anisotropic case.**
>
>     This is a very insightful question. Mathematically, the non-isotropic / low-dimensional case is more challenging than the isotropic setting.
>
>     In the isotropic case, spherical symmetry makes the geometry “clean’’: half-space depth, the BEoS-induced weight function, and the number of disjoint activation regions (spherical caps) can all be related through standard packing arguments on the ambient sphere. As a result, the depth–quantile index $S_{DQ}(P_X)$ serves as a fairly complete scalar summary of shatterability, and our upper/lower bounds in the isotropic Beta–radial family follow from this structure (see the General Response and revised Section 3.1–3.2).
>
>     For anisotropic or low-dimensional data, this alignment breaks down. When the support lies on low-dimensional structures (e.g., a union of $m$-dimensional subspaces), ReLU hyperplanes intersect the data only along low-dimensional slices (“knots’’), and many ambient spherical caps intersect the data manifold in highly overlapping ways. Thus, a large $S_{DQ}(P_X)$ no longer directly translates into a comparable number of effectively independent regions on the data, and a purely scalar index cannot fully capture shatterability. A general theory would require new tools that simultaneously control half-space depth, intrinsic geometry of the support, and how activation regions intersect it.
>
>     We address this difficulty in a structured anisotropic model: mixtures of low-dimensional balls. The analysis of $S_{DQ}(P_X)$ still plays a central role: under the mixture assumption, the BEoS-induced regularization decomposes across components, so that the stability “budget’’ is adaptively allocated to each modality according to its depth profile. This allows us to reduce the mixture-of-subspaces case to the analysis of a single low-dimensional component.
>
>     On a single $m$-dimensional subspace $V_j$, we then obtain a more precise, mathematically explicit notion of "effective capacity". Crucially, when we restrict the network to data supported on $V_j$, a neuron’s activation depends not on its full weight vector $w_k$, but only on its projection $\mathrm{proj}_{V_j} w_k$: the component orthogonal to $V_j$ is invisible to the data. This projection mechanism formalizes how linearly low-dimensional structures limit the model’s shatterability (as visualized by the “knots’’ in Figure 2), and leads to the single-subspace generalization result (Theorem 3.10). We have made this perspective explicit in the revised Section 3.3 and in our General Response.

---

> > ### Author Response · Authors · 2025-11-21
> > **Authors' Response to Reviewer PDwN (continual)**
> >
> > - **Scope of analysis and gradient dynamics (Weakness 3)**
> >
> >     We thank the reviewer for raising the concern about dynamics. In the revision, we clarify the **scope of analysis** explicitly in a new “Scope of Analysis’’ paragraph in the introduction. Our generalization bounds are stated for **any parameter state along the gradient descent trajectory that satisfies the BEoS condition**, and do *not* assume that the iterate is a minimizer or even a stationary point (vanishing gradient). In other words, our results characterize the implicit regularization enforced whenever the dynamics operate in the BEoS-stable region, rather than only at isolated minima. We have updated the introduction and the official General Response to make this point explicit.
> >
> >     To further address the dynamics viewpoint, we added Appendix B.2, which discusses our results directly from the perspective of gradient descent. There we show that for a two-layer ReLU network, the gradient of each hidden weight has the form
> >     $\nabla_{w_k} L(\theta) = \sum_i \alpha_{k,i}(\theta)\, x_i$, so the update direction for $w_k$ always lies in the span of the input vectors. In linearly low-rank settings, this implies that the entire trajectory of $w_k$ remains confined to a low-dimensional subspace, suggesting the schematic picture
> >     $$
> >     \text{data geometry}
> >     \Longrightarrow
> >     \text{gradient geometry}
> >     \Longrightarrow
> >     \text{stability-induced regularization}.
> >     $$
> >     However, as we explain in Appendix B.2, this perspective is technically hard to extend beyond linearly low-rank settings, especially once we move from idealized subspace models to mixtures of low-dimensional components. Our approach is therefore to work at the level of the **BEoS function class**: we encode the effect of gradient geometry into the data-dependent weight function $g_{\mathcal D}$, and then analyze how stability-induced regularization of this “data-shaped’’ class leads to intrinsic-dimension–adaptive and geometry-aware generalization. We believe this paradigm offers a realistic compromise between a purely static stability condition and an intractable full dynamical analysis, and we have tried to make this rationale clear in the revised text.
> >
> > - **Comparison to [Wu & Su 23]**
> >     We thank the reviewer for this very insightful question and for pointing out the connection to Wu & Su (2023). In the revised version, we added a dedicated “Technical Novelty’’ paragraph in the introduction and expanded the related-work discussion to clarify this comparison. Briefly, Wu & Su (2023) study two-layer ReLU networks **without hidden biases** under a noiseless model $y = f^*(x)$, with inputs that are essentially uniform on the sphere, and focus on linearly stable **global minima and interpolating solutions** of SGD/GD. In that setting they show that any such stable minimum enjoys an $O(1/n)$ generalization rate. By contrast, we consider two-layer ReLU networks **with hidden biases**,  do not impose a noiseless assumption on $ \mathcal{P}(y|X=x)$, and analyze all parameter states in the BEoS-stable region rather than only global minima. One of our key observations is that, with hidden biases, for any labeling of points on the sphere there exist BEoS-stable interpolating solutions (“flat interpolation’’), so dynamical stability alone no longer guarantees good generalization on spherical data once the network can realize affine (biased) separators.
> >
> >     The essential technical difference comes from the geometry of activation regions and how stability translates into capacity control. In the bias-free setting of Wu & Su, each neuron defines a half-space through the origin: on the sphere every such half-space cuts the data into two pieces of approximately equal mass, so every neuron activates on about 50\% of the samples regardless of its direction. This makes the BEoS-induced weight function effectively constant across neurons, and the stability condition can be turned into a **uniform (unweighted) path-norm bound**, which is then handled with standard Rademacher-complexity arguments. In our biased setting, activation regions become spherical caps of arbitrary size, and the BEoS weight assigned to each neuron genuinely depends on how much probability mass lies in its cap. The resulting regularization is **spatially inhomogeneous** over the input space, and our main technical contribution is to analyze this inhomogeneity via half-space depth and the depth–quantile concentration index, making generalization explicitly depend on data geometry (isotropic spectrum, flat interpolation on the sphere, and intrinsic-dimension adaptation). This geometric, spatially inhomogeneous stability mechanism is not present in the bias-free framework of Wu & Su. and we now emphasize this distinction clearly in the revised introduction and related-work sections.

---

> > > ### Author Response · Authors · 2025-11-27
> > > **Further Clarification**
> > >
> > > Thank you again for your careful review and for raising important questions regarding novelty and the relation to prior work. To help clarify the distinctions more clearly, we prepared the following concise comparison table summarizing assumptions, settings, and guarantees across the most relevant stability,  and path-norm–based analyses (including Wu & Su 2023, Parhi & Nowak 2023, Liang et al. 2025, and our work).
> > >
> > > | **Work**                       | **Setting / assumptions**                                                                                                                                                             | **Rate**                                        | **Context / contribution**                                                                                                                                                                                    |
> > > | ------------------------------ | ------------------------------------------------------------------------------------------------------------------------------------------------------------------------------------- | ----------------------------------------------- | ------------------------------------------------------------------------------------------------------------------------------------------------------------------------------------------------------------- |
> > > | **Parhi & Nowak (2023)**       | Unweighted path norm bounded (static estimator, no optimization dynamics)                                                                                                             | $\tilde{O}\left(n^{-\frac{d+3}{4d+6}}\right)$ | Near-minimax baseline for estimators with an *unweighted* path-norm constraint. Does not model GD/SGD dynamics.                                                                                               |
> > > | **Wu & Su (2023)**             | Interpolation at global, linearly stable minima; hidden-bias-free model $f(x)=\sum_j v_j \phi(w_j^{T}x)$; noiseless labels $y=f^*(x)$; inputs essentially uniform on the sphere | $\tilde{O}(n^{-1})$                             | Shows that any linearly stable interpolating minimum of GD/SGD enjoys an $O(1/n)$ generalization rate via a *uniform* (unweighted) path-norm control in a hidden-bias-free setting. The argument crucially relies on the isotropic data-distribution assumption.                                |
> > > | **Qiao (2024)**                | BEoS dynamics; **univariate** input                                                                                                                                                   | $\tilde{O}\left(n^{-\frac{2}{5}}\right)$      | Used the technique of “chopping off the bad region’’ for 1D data. Only considering good regions and does not address multivariate geometry.                                                                   |
> > > | **Liang et al. (2025)**        | BEoS dynamics; multivariate input **uniform** on the ball $\mathbb{B}^d_1$                                                                                                            | $\tilde{O}\left(n^{-\frac{1}{2d+4}}\right)$   | Analyzes neural shattering under isotropic inputs to obtain matching upper and lower bounds. No adaptation to intrinsic low-dimensional structure.                                                            |
> > > | **Ours (Thm. 3.10)**           | BEoS dynamics; input supported on a mixture of **$m$-dimensional** balls ($m \le d$)                                                                                                  | $\tilde{O}\left(n^{-\frac{1}{2m+4}}\right)$   | **Intrinsic-dimension adaptation.** The rate depends on $m$ rather than $d$, improving on Liang et al. for structured, low-dimensional data.                                                                  |
> > > | **Ours (Thm. 3.4\& Thm. 3.5)** | BEoS dynamics; input sampled from an **isotropic Beta($\alpha$)-radial family** on $\mathbb{B}^d_1$                                                                                   | rates depend on $\alpha$ and $d$                | Exhibits a **continuous spectrum** from generalization to memorization as $\alpha$ varies, with Liang et al.’s bounds appearing as special cases.                                                             |
> > > | **Ours (Thm. 3.6)**            | BEoS dynamics; input on the **sphere** $\mathbb{S}^{d-1}$                                                                                                                             | $\tilde{\Omega}(1)$                             | **Flat interpolation.** For any labeling on $\mathbb{S}^{d-1}$, constructs BEoS-stable interpolating networks, showing that stability alone does not guarantee generalization once hidden biases are allowed. |
> > >
> > > If you have further questions or if any part of this comparison does not address your concern, please let us know, we would be very happy to clarify.

---

### Official Review · Reviewer_M2En · 2025-11-01

**Soundness:** 2
**Presentation:** 3
**Contribution:** 3
**Rating:** 6
**Confidence:** 3

**Summary:**

This work studies the interaction between data geometry and the implicit bias of edge of stability.
It shows that EoS bias can drive two-layer ReLU networks to adapt low-dimensionality for mixture of low-dim balls data, and that for isotropic data, its shatterablity determines generalization.
Experiments verify the theoretical predictions.
This work then proposes the principle of shatterablity, where the shatterable data points attract specialized neurons, which are less regularized by the implicit weighted path norm in below the edge of stability.

**Strengths:**

- Based on the algorithmic bias of EoS, this work provides the data dependence aspect of neural network generalization, a valuable problem in modern data- and algorithm-dependent theory for generalization.
- The work picks two representative examples of interest, where the first one is related to how neural network overcomes curse of dimensionality and the second one novelly reveals the role of shatterablity. The results are verified by empirical results.
- This work also reveals that under EoS regularization, the network may still overfits, and it is data geometry with low shatterablity that helps resisting overfitting.
- This work provides principled lens for studying feature learning and data geometry reflected in it, eg, neuron activation rate that impacts regularization strength of EoS bias and affects generalization.

**Weaknesses:**

- The paper supports the shatterablity principle using two proved cases, followed by intuitive interpolation/extrapolation. However, a formal results is missing, leaving shatterablity relying on intuitive definition and restricting its application to more complicated data. Is it possible to derive formal definition of shatterablity and provide more abstract generalization bounds with shatterablity and BEoS as parameters?
- In experiments, the training data is constructed by perturbing the label instead of IID sampling. How does this setting fits into the assumption of theories? Under standard setting, what will be low-dimension adaptation like?

**Questions:**

- Some works have emphasized the surprising importance of (benign) memorization for generalization, especially under long-tailed data distribution. Then is there any connection from lon-tailedness and memorization to shatterablity and neuron specialization? If so, what benign memorization looks like in the framework of shatterablity? At what threshold does memorization becomes harmful?

---

> ### Author Response · Authors · 2025-11-21
> **Authors’ Response to Reviewer M2En**
>
> We thank you for your constructive review and for recognizing the value of our "shatterability" principle. We have carefully addressed your concerns regarding formalization and experimental settings in the revised manuscript.
>
> - **Re: "The paper supports the shatterablity principle using two proved cases, followed by intuitive interpolation/extrapolation. However, a formal results is missing, leaving shatterablity relying on intuitive definition and restricting its application to more complicated data."**
>
>     We appreciate the reviewer’s request for a more formal and abstract treatment of shatterability; this comment accurately highlights both the difficulty and the potential of the problem.
>
>     On the **formal definition side**, as detailed in our General Response and in the revised Section 3, we now make the notion of data shatterability precise via half-space (Tukey) depth and a scalar concentration index $S_{DQ}(P_X)$ (Definition 3.8), constructed from the depth–quantile function. This provides a rigorous, geometric, data-dependent quantity that captures how much of the distribution lies in regions where BEoS implicit regularization is strong.
>
>
> - **Re: "Is it possible to derive formal definition of shatterablity and provide more abstract generalization bounds with shatterablity and BEoS as parameters?"**
>
>     We thank the reviewer for this insightful question. It clearly identifies a fundamental challenge: turning the shatterability principle into a fully abstract generalization theory analogous to VC dimension. Our framework takes a concrete step in this direction, but we agree that closing this conceptual gap requires substantially deeper developments.
>
>     At a high level, our framework “inverts’’ the classical VC perspective. VC dimension measures a model’s ability to shatter *arbitrary* data, whereas our notion of “data shatterability’’ quantifies the *feasibility* of shattering a **specific** dataset under GD and BEoS implicit regularization.
>
>     The difficulty in obtaining a VC-style abstract bound, however, goes beyond defining a depth-based scalar index. Shatterability under BEoS depends not only on the depth profile but also on the **geometric arrangement** of the data and the **available partitioning directions**. This becomes evident in anisotropic or low-dimensional settings. As discussed at the beginning of Section 3.3, low intrinsic dimension collapses the high-dimensional spherical-cap geometry into simple intervals or knots (Figure 2), drastically reducing the number of disjoint regions that ReLU hyperplanes can form, even when the ambient depth profile (and $S_{DQ}(P_X)$) looks worse than an isotropic Beta-radial distribution. In such cases, $S_{DQ}(P_X)$ serves only as a *conservative* estimate of difficulty, and additional structural information is required.
>
>     A fully abstract theory would therefore require new tools that integrate half-space depth with the intrinsic geometry of the support. Developing such a framework appears technically challenging but is a natural and compelling direction for future work. We have added discussion in the revised manuscript (Section 3.3 and concluding remarks) to make these limitations and the associated roadmap explicit, and we are grateful to the reviewer for highlighting this perspective.

---

> ### Author Response · Authors · 2025-11-21
> **Authors’ Response to Reviewer M2En (continual)**
>
> - **Re: "In experiments, the training data is constructed by perturbing the label instead of IID sampling. How does this setting fits into the assumption of theories?"**
>
>    We did consider the standard iid setting in the theoretical results.   In our theoretical setup (Assumptions 3.3 and 3.10), the data are drawn from a joint distribution $\mathcal{P}(x,y)$ with an arbitrary conditional law $\mathcal{P}(y\mid x)$. The experimental construction
>     $$
>     Y = f^{\*}(X) + \xi,\qquad \xi \sim \mathcal{N}(0,\sigma^2)
>     $$
>     is exactly an instance of this model, corresponding to a Gaussian conditional distribution centered at $f^{\*}(x)$.
>
>   The additive label noise is introduced so that we can meaningfully measure the **generalization gap**. For squared loss,
>     $$
>     R_{\mathrm{noisy}}(f)= \mathbb{E}\big[(f(X)-Y)^2\big]= \underbrace{\mathbb{E}\big[(f(X)-f^{\*}(X))^2\big]}\_{\text{excess risk}}+ \underbrace{\sigma^2}\_{\text{noise level}},
>     $$
>     while the empirical training loss $\widehat{R}\_{\mathrm{train}}(f)$ uses the noisy labels. The “clean’’ empirical risk we report in the experiments,
>     $$
>     \widehat{R}\_{\mathrm{clean}}= \frac{1}{n}\sum_{i=1}^n \big(f(X_i)-f^{\*}(X_i)\big)^2,
>     $$
>     is simply a Monte Carlo estimate of the excess risk. Thus the generalization gap can be written as
>     $$
>     \mathrm{GenGap}= R_{\mathrm{noisy}}(f) - \widehat{R}\_{\mathrm{train}}(f)=\widehat{R}\_{\mathrm{clean}} + \sigma^2 - \widehat{R}\_{\mathrm{train}}(f).
>     $$
>
>     In other words, our experimental protocol is fully consistent with the theoretical assumption $(X,Y)\sim\mathcal{P}$, and it allows us to directly observe how the excess risk (the part controlled by our theory) contributes to the generalization gap. On geometries where the network cannot easily fit the injected noise (“hard-to-shatter’’ settings), the excess risk remains small, leading to a small generalization gap, in line with the theoretical predictions.

---

> > ### Author Response · Authors · 2025-11-26
> > **Authors’ Response to Reviewer M2En (continual: low-dimensional adaptation)**
> >
> > - **Re: “Under standard setting, what will be low-dimension adaptation like?”**
> >
> > In the standard i.i.d. setting, we sample inputs from a mixture of 20 one-dimensional line segments embedded in $\mathbb{R}^d$ and train on noisy labels from a fixed target function. We then plot the clean test error versus the sample size on a log–log scale, and interpret the slope as an empirical sample-complexity exponent. As we increase the ambient dimension (d) (e.g., from 10 to 500), these slopes remain almost unchanged, indicating that the effective rate is governed by the intrinsic one-dimensional structure rather than by (d); this is what we mean by low-dimensional adaptation. In contrast, for isotropic data approximately uniform on a ball, the slope quickly shrinks towards zero already at modest (d), so the excess risk decreases much more slowly with the sample size, consistent with prior observations in [Liang et al., 2025].
> >
> > Beyond these statistics, we visualize low-dimensional adaptation in Figure 2 (Section 3.3) in the new version. Intuitively, we demonstrate that the network's complex decision boundaries, formed by combinations of half-spaces, are fundamentally constrained by the data's intrinsic low-dimensional structure. For example, when data lies on a line within $\mathbb{R}^d$, a ReLU's complex hyperplane boundaries reduce to a series of knots and entire complexity is defined by the locations and magnitudes of these knots (Figure.2a). The cornerstone of our proof is showing that the stability induced, data-dependent implicit regularization is adaptive to this nature, see (line 359-line 400) about how we use the half-space-depth concentration index to enlighten the proof, see also Figure.2b about the visualization $S_{\text{DQ}}$ for mixture of lines.

---

### Official Review · Reviewer_h9Wg · 2025-11-01

**Soundness:** 3
**Presentation:** 3
**Contribution:** 3
**Rating:** 8
**Confidence:** 3

**Summary:**

This paper investigates how the geometry of the training data fundamentally controls the implicit bias of gradient descent (GD) and the resulting generalization performance of overparameterized two-layer ReLU networks trained in the "Below Edge of Stability" (BEoS) regime.

**Strengths:**

The author claims that  "The less shatterable the data geometry, the stronger the implicit regularization of EoS becomes." and illustrates this observation via two specific example.

I thought this is a very interesting result and made a serious try to understand the performance neural network comparing with other "not even wrong" work.

**Weaknesses:**

I have not checked the whole proof. The results sound reasonable to me. However, to broad its impacts, it would be more beneficial if the author could make more implications of their theoretical results. e.g., its connection with some exiting theories?  Moreover, the rates stated in theorems are more less to technical,  could the authors  make it more comparable with some existing results?

**Questions:**

Same to the weakness.

---

> ### Author Response · Authors · 2025-11-21
> **Authors' Response to Reviewer h9Wg**
>
> We appreciate your interest and encouragement! Responses to your questions below:
>
> - **Re: "However, to broad its impacts, it would be more beneficial if the author could make more implications of their theoretical results. e.g., its connection with some exiting theories?"**
>
>     This is a good idea! We add a “Technical Novelty” in the v2 to explicitly contextualize our contribution within the framework of uniform convergence in statistical learning. We highlight them to you here:
>
>     Classical generalization bounds typically rely on uniform control of the empirical Rademacher complexity (often via global $L^\infty$-metric entropy). This approach effectively treats all data points equally and bounds the worst-case complexity across the domain. However, this paradigm does not work in this paper. In our setting, the implicit regularization is **highly inhomogeneous**. There are "shallow regions" (near boundaries) where the stability constraint is extremely weak (which is also empirical verified for real world data, see Section 4.3), allowing for arbitrarily high local complexity. A standard uniform bound would be dominated by these regions, leading to a significant overestimation of the actual complexity (i.e., vacuous bounds).
>
>     Our **half-space-depth partition technique** enables a **fine-grained, geometry-dependent control** of the complexity. Instead of a uniform bound, we decouple the analysis: we tolerate high complexity in "shallow" regions because we prove they carry small probability mass, while enforcing strict control in the "deep" regions where the data actually concentrates. This allows us to derive tight bounds where standard methods fail.
>
>     Besides, our analysis of low-dimensional adaptation reveals that standard bounds overestimate capacity when weights are large but **orthogonal** to the data distribution. Conceptually, our framework should be regarded to invert the classical VC perspective: rather than measuring a model's *active capacity* to shatter *arbitrary* data, we characterize the *feasibility* of a *specific* dataset being shattered under stability constraints.
>
>
> - **Re: "Moreover, the rates stated in theorems are more less to technical, could the authors make it more comparable with some existing results?"**
>
>     Thank you for the opportunity to clarify our contributions relative to the literature. The table below compares our results with relevant prior works, highlighting how our bounds adapt to data geometry.
>
>     | **Work** | **Assumption** | **Rate** | **Context/Contribution** |
>     | --- | --- | --- | --- |
>     | **Parhi & Nowak (2023)** | **Unweighted** path norm is bounded (Static, no optimization dynamics) | $\tilde{O}(n^{-\frac{d+3}{4d+6}})$ (Near minimax optimal) | This serves as a baseline. Our method recovers similar rates in the "deep" regions of the data. |
>     | **Qiao et al. (2024)** | BEoS dynamics,**univariate** input ($d=1$) | $\tilde{O}(n^{-\frac{2}{5}})$ (Near minimax optimal) | They introduced the idea of "chopping off the bad region" for 1D. Our work generalizes this strategy to high-dimensional, complex geometries. |
>     | **Liang et al. (2025)** | BEoS dynamics, multivariate input uniform on the ball $B^d_1$ | $\tilde{O}(n^{-\frac{1}{2d+4}})$ and $\Omega(n^{\frac{2}{d+1}})$ | Observe the “neural shattering” phenomenon and use it to deduce upper bounds and lower bounds. |
>     | **Ours (Theorem 3.10)** | BEoS dynamics, input supported on a mixture of **$m$-dimensional** balls ($m \ll d$) | $\tilde{O}(n^{-\frac{1}{2m+4}})$ | **Provable adaptation to intrinsic dimension.** Our rate depends on $m$, not $d$, significantly improving upon Liang et al. (2025) for structured data. |
>     | **Ours (Theorem 3.6 & 3.7)** | BEoS dynamics, input distributions from the isotropic Beta($\alpha$)-radial family | Rate depends on $\alpha$ and $d$ | We demonstrate a continuous spectrum from generalization to memorization, with the bounds from Liang et al. (2025) appearing as specific points within this spectrum. |
>
>     We will incorporate this discussion into the next version of our paper. If you have specific papers you'd like us to compare with, please let us know. We'd be happy to highlight the differences.
>
> References:
> 1. [Parhi & Nowak 23] :  Near-minimax optimal estimation with shallow ReLU neural networks. IEEE Transactions on Information Theory
> 2. [Qiao et.al  24]: Stable minima cannot overfit in univariate ReLU networks: Generalization by large step sizes. Neurips 2024
> 3. [Liang et.al 25]: Stable minima of relu neural networks suffer from the curse of dimensionality: The neural shattering phenomenon, Neurips2025

---

> > ### Comment · Reviewer_h9Wg · 2025-11-21
> >
> > Thanks for the clarification. I am satisfied with your response and would like to keep my current positive score.

---

### Author Response · Authors · 2025-11-19
**General Response: Revision to Formalize Concepts and Unify Theoretical Framework**

Dear Area Chair and Reviewers,

Thank you for your detailed feedback sincerely. We have taken your suggestions seriously and performed a substantial revision of the paper to formalize our core concepts and strengthen the logical structure.

The revised manuscript has been uploaded, and we believe this revision directly addresses the core concerns raised regarding clarity and formalization. The following are the highlight of the revision.

**Summary of Key Revisions:**

1. **Restructured Narrative for Logical Coherence:**
We have reordered the main results (**Isotropic $\to$ Unified Framework $\to$ Mixture Models**) to provide a more natural progression from specific intuition to general principles and finally to complex applications.
2. **Exhibit Proof Techniques and Formalize a Rigorous Proxy for Shatterability (Section 3):**
To address the concerns about the definition of "shatterability", we have introduced the half-space-depth (Tukey-depth) concentration index ($\mathsf{S}_{\text{DQ}}$) as a formal proxy for data shatterability (Section 3.2).
    - **Isotropic Case (Section 3.1):** We rigorously analyze this proxy for isotropic distributions, showing exactly how the concentration of probability mass (quantified by $\mathsf{S}_{\text{DQ}}$) dictates the generalization gap.
    - **Mixture & Anisotropic Cases (Section 3.3):** We discuss how this principle guides our analysis of mixture models (adaptation to intrinsic dimension) while explicitly acknowledging the fundamental mathematical challenges in defining a scalar metric for general anisotropic distributions.
3. **Clarifying the Technical Innovation**
We have added a "Technical Novelty" paragraph  in the Introduction to explicitly articulate the technical challenge and our solution.
    - **The Challenge:** We clarify that the implicit regularization induced by the EoS condition is highly inhomogeneous over the input domain: it is strong in "deep" regions but very weak in "shallow" regions, leading to infinite global $L^\infty$ metric entropy.
    - **Our Solution:** We explain our **half-space-depth partition technique**, which decouples the analysis based on the strength of regularization. This allows us to enforce strict complexity control in the "good" regions while handling the "bad" regions via probability mass bounds. We believe this exposition makes the derivation of our bounds transparent and easier to follow. Conceptually, the data shatterability principle is built upon the insights derived from these proof techniques.
4. **Clarifying the Scope of Analysis:**
We wrote  "Scope of Analysis” paragraph in the Introduction to clarify the applicability of our results. We emphasize that our generalization bounds hold for **any parameter state** along the gradient trajectory that satisfies the BEoS condition. Crucially, our analysis **does not assume stationarity** (i.e., vanishing gradients) or optimality. It characterizes the implicit regularization enforced whenever the training dynamics operate in the stable regime.
5. **Enhanced Visualization:**
 We added **Figure 1 & Figure 2** to schematically illustrate the concept of data shatterability and how our depth-based proxy captures the geometric difficulty of partitioning data.

**Important Note on Technical Consistency:**
We emphasize that **the core technical results (generalization bounds, lower bounds) and experimental results remain unchanged.** This revision is strictly focused on improving the presentation, rigorous formalization of concepts, and narrative flow to better communicate the contributions.

We will post detailed, point-by-point responses to each reviewer’s specific technical questions in separate comments shortly.

Thank you again for helping us improve the quality of this work.

Best regards,
The Authors

---

### Author Response · Authors · 2025-11-20
**General Response: Rigorous Formulation and Unified Framework of “Data Shatterability”**

Several reviewers asked for a more rigorous and unified explanation of our notion of “data shatterability”. We appreciate this opportunity to clarify the concept, and we summarize the formal definition and its role in our framework below.

**1. Motivation: BEoS-induced, spatially inhomogeneous regularity**

In the BEoS regime, gradient descent is constrained by a data-dependent weighted path norm
$$
\sum_k |v_k| ||w_k||_2 g(w_k,b_k)\le C(\eta),
$$
where $g$ depends on the training data. When a neuron fires strongly on many samples, $g$ is large and the constraint forces $|v_k|||w_k||_2$ to stay small; neurons that activate only on few points receive almost no penalty and can overfit those regions. Thus BEoS induces **spatially inhomogeneous regularization**: some parts of the input space are effectively regularized, others are nearly unregularized. The question then becomes how to characterize these regions geometrically.

**2. Half-space depth as the geometric bridge**

This observation leads naturally to half-space (Tukey) depth. A point $x$ has high depth if **every** half-space containing $x$ also contains a significant portion of the data. Any ReLU neuron that activates at such an $x$ must activate on that portion, triggering strong BEoS regularization. Hence deeper points enjoy stronger protection against overfitting:
$$
\operatorname{depth}(x,P_X):=\inf_{u\in\mathbb S^{d-1}}\mathbb P(u^\top(X-x)\ge0).
$$
Informally, **the deeper a point is, the better BEoS can be expected to generalize there.**

**3. From pointwise depth to a distribution-level index**

To capture this effect at the distribution level, we consider the depth-quantile function
$\Psi_{P_X}(T):=\mathbb{P}\big(\operatorname{depth}(X,P_X)\ge T\big)$ and its area
$\int_0^{1/2}\Psi_{P_X}(T)dT$,
which measures how much mass lies in well-regularized regions. In Definition 3.8 we define the **half-space-depth concentration index**
$$
S_{DQ}(P_X):=\Bigg(\int_0^{1/2}\Psi_{P_X}(T)\,dT\Bigg)^{-1}
$$
as our proxy for shatterability: large area $\Rightarrow$ strong regularization and **low shatterability**; small area $\Rightarrow$ most mass is shallow and **high shatterability**. The spherical distribution (depth $=0$ everywhere) yields zero area and divergent $S_{DQ}$, matching the flat interpolation phenomenon (Theorem 3.6). Our isotropic spectrum (Theorem 3.4 & 3.5) and low-dimensional adaptation (Theorem 3.10) are precisely different depth profiles viewed through this lens.

**4. Role in our upper and lower bounds (and the effect of $\alpha$)**

We use $S_{DQ}(P_X)$ to quantify how feasible it is to populate many disjoint “shallow” regions (spherical caps). In isotropic Beta$(\alpha)$-radial distributions, $\alpha$ controls how much mass lies near the boundary. When $\alpha$ is small, the boundary shell carries significant mass, enabling many populated caps; this yields larger shatterability and allows the lower-bound construction to create larger generalization gaps. When $\alpha$ is large, the shallow shell has very small mass: such points may still be easy to fit, but their contribution to population risk is negligible. In our upper-bound decomposition, this corresponds exactly to the shrinking shallow-region term. The visualization can be found in Figure 1.

This proxy is also instrumental in explaining the tightness of our bounds. As discussed in Remark 3.9,  our upper bound analysis currently relies on selecting an optimal depth threshold $T$, which geometrically amounts to inscribing a single rectangle under the depth-quantile curve. The gap between our upper and lower bounds arises precisely because this "rectangle approximation" discards the regularization benefit provided by the remaining area under the curve.

**We sincerely thank the reviewers for engaging deeply with the technical aspects of our work. The topic is inherently subtle, and we truly appreciate the time and effort it takes to evaluate a framework that aims to make these mechanisms precise. We hope that the clarifications above help convey the underlying structure more transparently. We are genuinely grateful for your patience in working through these refinements, and your thoughtful feedback has greatly helped us improve both the clarity and rigor of the presentation.**

---

### Meta-Review · Area_Chair_oYTN · 2026-01-06

**Summary:**

This paper studies how the geometry of the training data controls the implicit bias of gradient descent  and the resulting generalization performance of over-parameterized two-layer ReLU networks trained under the regime of Below Edge of Stability.

The reviewers raised the following concerns: some critical definitions (like isotropic beta-radial distributions and data shatterability) are missing, little intuition of the analysis is given, and there is no dynamics analysis.

**Reviewer Concerns:**

I believe the authors address all the above concerns, by defining everything clearly, restructuring Section 3 to give some intuition, and adding a ``Scope of Analysis’’ paragraph in the introduction which stresses that the generalization bounds are stated for any parameter state along the gradient descent trajectory that satisfies the BEoS condition.

**Reviewer Scores:**

I think at least one of the two reviewers with a score of 4 would have increased the score to 6 given a full discussion.

---

### Decision · Program_Chairs · 2026-01-26

Accept (Poster)